# Learning and Generalization in Overparameterized Neural Networks, Going Beyond Two Layers[*]

**Zeyuan Allen-Zhu**
Microsoft Research AI
zeyuan@csail.mit.edu

**Yuanzhi Li**
Carnegie Mellon University
yuanzhil@andrew.cmu.edu

**Yingyu Liang**
University of Wisconsin-Madison
yliang@cs.wisc.edu

## Abstract

The fundamental learning theory behind neural networks remains largely open. What classes of functions can neural networks actually learn? Why doesn't the trained network overfit when it is overparameterized?

In this work, we prove that overparameterized neural networks can learn some notable concept classes, including two and three-layer networks with fewer parameters and smooth activations. Moreover, the learning can be simply done by SGD (stochastic gradient descent) or its variants in polynomial time using polynomially many samples. The sample complexity can also be almost independent of the number of parameters in the network.

On the technique side, our analysis goes beyond the so-called NTK (neural tangent kernel) linearization of neural networks in prior works. We establish a new notion of quadratic approximation of the neural network, and connect it to the SGD theory of escaping saddle points.

## 1   Introduction

Neural network learning has become a key machine learning approach and has achieved remarkable success in a wide range of real-world domains, such as computer vision, speech recognition, and game playing [25, 26, 30, 41]. In contrast to the widely accepted empirical success, much less theory is known. Despite a recent boost of theoretical studies, many questions remain largely open, including fundamental ones about the optimization and generalization in learning neural networks.

One key challenge in analyzing neural networks is that the corresponding optimization is non-convex and is theoretically hard in the general case [40, 55]. This is in sharp contrast to the fact that simple optimization algorithms like stochastic gradient descent (SGD) and its variants usually produce good solutions in practice even on both training and test data. Therefore,

*what functions can neural networks provably learn?*

Another key challenge is that, in practice, neural networks are heavily overparameterized (e.g., [53]): the number of learnable parameters is much larger than the number of the training samples. It is observed that overparameterization empirically improves both optimization and generalization, appearing to contradict traditional learning theory.[2] Therefore,

*why do overparameterized networks (found by those training algorithms) generalize?*

---

[*]Full version and future updates can be found on `https://arxiv.org/abs/1811.04918`.

[2]For example, Livni et al. [36] observed that on synthetic data generated from a target network, SGD converges faster when the learned network has more parameters than the target. Perhaps more interestingly, Arora et al. [6] found that overparameterized networks learned in practice can often be compressed to simpler ones with much fewer parameters, without hurting their ability to generalize; however, directly learning such simpler networks runs into worse results due to the optimization difficulty. We also have experiments in Figure 1(a).

## 1.1 What Can Neural Networks Provably Learn?

Most existing works analyzing the learnability of neural networks [9, 12, 13, 20, 21, 28, 33, 34, 42, 43, 47, 49, 50, 56] make unrealistic assumptions about the data distribution (such as being random Gaussian), and/or make strong assumptions about the network (such as using linear activations). Li and Liang [32] show that two-layer ReLU networks can learn classification tasks when the data come from mixtures of arbitrary but well-separated distributions.

A theorem *without distributional assumptions* on data is often more desirable. Indeed, how to obtain a result that does not depend on the data distribution, but only on the concept class itself, lies in the center of PAC-learning which is one of the foundations of machine learning theory [48]. Also, studying *non-linear activations* is critical because otherwise one can only learn linear functions, which can also be easily learned via linear models without neural networks.

Brutzkus et al. [14] prove that two-layer networks with ReLU activations can learn linearly-separable data (and thus the class of linear functions) using just SGD. This is an (improper) *PAC-learning* type of result because it makes no assumption on the data distribution. Andoni et al. [5] proves that two-layer networks can learn polynomial functions of degree $r$ over $d$-dimensional inputs in sample complexity $O(d^r)$. Their learner networks use exponential activation functions, where in practice the rectified linear unit (ReLU) activation has been the dominant choice across vastly different domains.

On a separate note, if one treats all but the last layer of neural networks as generating a random feature map, then training only the last layer is a convex task, so one can learn the class of linear functions in this implicit feature space [15, 16]. This result implies low-degree polynomials and compositional kernels can be learned by neural networks in polynomial time. Empirically, training last layer greatly weakens the power of neural networks (see Figure 1).

**Our Result.**   We prove that an important concept class that contains three-layer (resp. two-layer) neural networks equipped with smooth activations can be efficiently learned by three-layer (resp. two-layer) ReLU neural networks via SGD or its variants.

Specifically, suppose in aforementioned class the best network (called the target function or target network) achieves a population risk $\mathsf{OPT}$ with respect to some convex loss function. We show that one can learn up to population risk $\mathsf{OPT} + \varepsilon$, using three-layer (resp. two-layer) ReLU networks of size greater than a fixed *polynomial* in the size of the target network, in $1/\varepsilon$, and in the "complexity" of the activation function used in the target network. Furthermore, the sample complexity is also *polynomial* in these parameters, and only poly-logarithmic in the size of the learner ReLU network.

We stress here that this is *agnostic* PAC-learning because we allow the target function to have error (e.g., $\mathsf{OPT}$ can be positive for regression), and is *improper* learning because the concept class consists of smaller neural networks comparing to the networks being trained.

**Our Contributions.**   We believe our result gives further insights to the fundamental questions about the learning theory of neural networks.

- To the best of our knowledge, this is the first result showing that using *hidden layers* of neural networks one can provably learn the concept class containing two (or even three) layer neural networks with non-trivial activation functions.[3]

- Our three-layer result gives the first theoretical proof that learning neural networks, even with *non-convex* interactions across layers, can still be plausible. In contrast, in the two-layer case the optimization landscape with overparameterization is almost convex [17, 32]; and in previous studies on the multi-layer case, researchers have weakened the network by applying the so-called NTK (neural tangent kernel) linearization to remove all non-convex interactions [4, 27].

- To some extent we explain the reason why overparameterization improves testing accuracy: with larger overparameterization, one can hope to learn better target functions with possibly larger size, more complex activations, smaller risk $\mathsf{OPT}$, and to a smaller error $\varepsilon$.

- We establish new tools to tackle the learning process of neural networks in general, which can be useful for studying other network architectures and learning tasks. (E.g., the new tools here

have allowed researchers to study also the learning of recurrent neural networks [2].)

**Other Related Works.** We acknowledge a different line of research using *kernels* as improper learners to learn the concept class of neural networks [22, 23, 36, 54]. This is very different from us because we use "neural networks" as learners. In other words, we study the question of "what can neural networks learn" but they study "what alternative methods can replace neural networks."

There is also a line of work studying the relationship between neural networks and NTKs (neural tangent kernels) [3, 4, 7, 27, 31, 51]. These works study neural networks by considering their "linearized approximations." There is a known performance gap between the power of real neural networks and the power of their linearized approximations. For instance, ResNet achieves 96% test error on the CIFAR-10 data set but NTK (even with infinite width) achieves 77% [7]. We also illustrate this in Figure 1.

## 1.2 Why Do Overparameterized Networks Generalize?

Our result above assumes that the learner network is sufficiently overparameterized. So, why does it generalize to the population risk and give small test error? More importantly, why does it generalize with a number of samples that is (almost) independent of the number of parameters?

This question cannot be studied under the traditional VC-dimension learning theory since the VC dimension grows with the number of parameters. Several works [6, 11, 24, 39] explain generalization by studying some other "complexity" of the learned networks. Most related to the discussion here is [11] where the authors prove a generalization bound in the norms (of weight matrices) of each layer, as opposed to the number of parameters. There are two main concerns with those results.

- Learnability = Trainability + Generalization. It is not clear from those results how a network with *both* low "complexity" *and* small training loss can be found by the training method. Therefore, they do not directly imply PAC-learnability for non-trivial concept classes (at least for those concept classes studied by this paper).

- Their norms are "sparsity induced norms": for the norm not to scale with the number of hidden neurons $m$, essentially, it requires the number of neurons with non-zero weights *not* to scale with $m$. This more or less reduces the problem to the non-overparameterized case.

At a high level, our generalization is made possible with the following sequence of conceptual steps.

- Good networks with small risks are *plentiful*: thanks to overparameterization, with high probability over random initialization, there exists a good network in the close neighborhood of *any* point on the SGD training trajectory. (This corresponds to Section 6.2 and 6.3.)

- The optimization in overparameterized neural networks has benign properties: essentially along the training trajectory, there is no second-order critical points for learning three-layer networks, and no first-order critical points for two-layer. (This corresponds to Section 6.4.)

- In the learned networks, information is also *evenly distributed* among neurons, by utilizing either implicit or explicit regularization. This structure allows a new generalization bound that is (almost) independent of the number of neurons. (This corresponds to Section 6.5 and 6.6, and we also empirically verify it in Section 7.1.)

Since practical neural networks are typically overparameterized, we genuinely hope that our results can provide theoretical insights to networks used in various applications.

## 1.3 Roadmap

In the main body of this paper, we introduce notations in Section 2, present our main results and contributions for two and three-layer networks in Section 3 and 4, and conclude in Section 5.

For readers interested in our novel techniques, we present in Section 6 an 8-paged proof sketch of our three-layer result. For readers more interested in the practical relevance, we give more experiments in Section 7. In the appendix, we begin with mathematical preliminaries in Appendix A. Our full three-layer proof is in Appendix C. Our two-layer proof is much easier and in Appendix B.

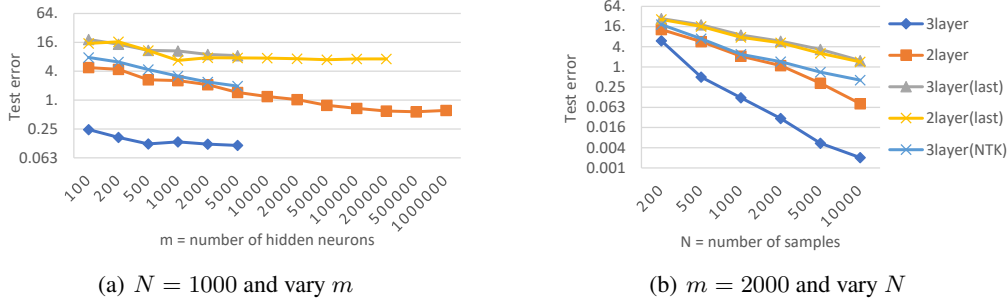

(a) $N = 1000$ and vary $m$            (b) $m = 2000$ and vary $N$

Figure 1: Performance comparison. `3layer/2layer` stands for training (hidden weights) in three and two-layer neural networks. `(last)` stands for *conjugate kernel* [15], meaning training only the output layer. `(NTK)` stands for *neural tangent kernel* [27] with finite width. We also implemented other direct kernels such as [54] but they perform much worse.

---

**Setup.** We consider $\ell_2$ regression task on synthetic data where feature vectors $x \in \mathbb{R}^4$ are generated as normalized random Gaussian, and label is generated by target function $F^*(x) = (\sin(3x_1) + \sin(3x_2) + \sin(3x_3) - 2)^2 \cdot \cos(7x_4)$. We use $N$ training samples, and SGD with mini-batch size 50 and best tune learning rates and weight decay parameters. See Appendix 7 for our experiment setup, how we choose such target function, and more experiments.

## 2 Notations

$\sigma(\cdot)$ denotes the ReLU function $\sigma(x) = \max\{x, 0\}$. Given $f: \mathbb{R} \to \mathbb{R}$ and a vector $x \in \mathbb{R}^m$, $f(x)$ denotes $f(x) = (f(x_1), \dots, f(x_m))$. For a vector $w$, $\|w\|_p$ denote its $p$-th norm, and when clear from the context, abbreviate $\|w\| = \|w\|_2$. For a matrix $W \in \mathbb{R}^{m \times d}$, use $W_i$ or sometimes $w_i$ to denote the $i$-th row of $W$. The row $\ell_p$ norm is $\|W\|_{2,p} := \left( \sum_{i \in [m]} \|W_i\|_2^p \right)^{1/p}$, the spectral norm is $\|W\|_2$, and the Frobenius norm is $\|W\|_F = \|W\|_{2,2}$. We say $f: \mathbb{R}^d \to \mathbb{R}$ is $L$-Lipschitz continuous if $|f(x) - f(y)| \leq L\|x - y\|_2$; is $L$-Lipschitz smooth if $\|\nabla f(x) - \nabla f(y)\|_2 \leq L\|x - y\|_2$.

**Function complexity.** The following notion measures the complexity of any smooth activation function $\phi(z)$. Suppose $\phi(z) = \sum_{i=0}^{\infty} c_i z^i$. Given a non-negative $R$, the complexity

$$\mathfrak{C}_\varepsilon(\phi, R) := \sum_{i=0}^{\infty} \left( (C^*R)^i + \left( \frac{\sqrt{\log(1/\varepsilon)}}{\sqrt{i}} C^*R \right)^i \right) |c_i|, \quad \mathfrak{C}_\mathfrak{s}(\phi, R) := C^* \sum_{i=0}^{\infty} (i+1)^{1.75} R^i |c_i|$$

where $C^*$ is a sufficiently large constant (e.g., $10^4$). Intuitively, $\mathfrak{C}_\mathfrak{s}$ measures the *sample* complexity: how many samples are required to learn $\phi$ correctly; while $\mathfrak{C}_\varepsilon$ bounds the *network* size: how much over-parameterization is needed for the algorithm to efficiently learn $\phi$ up to $\varepsilon$ error. It is always true that $\mathfrak{C}_\mathfrak{s}(\phi, R) \leq \mathfrak{C}_\varepsilon(\phi, R) \leq \mathfrak{C}_\mathfrak{s}(\phi, O(R)) \times \mathsf{poly}(1/\varepsilon)$.[4] While for $\sin z, \exp(z)$ or low degree polynomials, $\mathfrak{C}_\mathfrak{s}(\phi, O(R))$ and $\mathfrak{C}_\varepsilon(\phi, R)$ only differ by $o(1/\varepsilon)$.

**Example 2.1.** If $\phi(z) = e^{c \cdot z} - 1$, $\phi(z) = \sin(c \cdot z)$, $\phi(z) = \cos(c \cdot z)$ for constant $c$ or $\phi(z)$ is low degree polynomial, then $\mathfrak{C}_\varepsilon(\phi, 1) = o(1/\varepsilon)$ and $\mathfrak{C}_\mathfrak{s}(\phi, 1) = O(1)$. If $\phi(z) = \mathrm{sigmoid}(z)$ or $\tanh(z)$, we can truncate their Taylor series at degree $\Theta(\log \frac{1}{\varepsilon})$ to get $\varepsilon$ approximation. One can verify this gives $\mathfrak{C}_\varepsilon(\phi, 1) \leq \mathsf{poly}(1/\varepsilon)$ and $\mathfrak{C}_\mathfrak{s}(\phi, 1) \leq O(1)$.

## 3 Result for Two-Layer Networks

We consider learning some unknown distribution $\mathcal{D}$ of data points $z = (x, y) \in \mathbb{R}^d \times \mathcal{Y}$, where $x$ is the input point and $y$ is the label. Without loss of generality, assume $\|x\|_2 = 1$ and $x_d = \frac{1}{2}$.[5] Consider a loss function $L: \mathbb{R}^k \times \mathbb{R} \to \mathcal{Y}$ such that for every $y \in \mathcal{Y}$, the function $L(\cdot, y)$ is non-negative, convex, 1-Lipschitz continuous and 1-Lipschitz smooth and $L(0, y) \in [0, 1]$. This includes both the cross-entropy loss and the $\ell_2$-regression loss (for bounded $\mathcal{Y}$).

**Concept class and target function $F^*(x)$.** Consider target functions $F^*\colon \mathbb{R}^d \to \mathbb{R}^k$ of

$$F^* = (f_1^*, \dots, f_k^*) \quad \text{and} \quad f_r^*(x) = \sum_{i=1}^p a_{r,i}^* \phi_i(\langle w_{1,i}^*, x\rangle)\langle w_{2,i}^*, x\rangle \tag{3.1}$$

where each $\phi_i\colon \mathbb{R} \to \mathbb{R}$ is infinite-order smooth and the weights $w_{1,i}^* \in \mathbb{R}^d$, $w_{2,i}^* \in \mathbb{R}^d$ and $a_{r,i}^* \in \mathbb{R}$. We assume for simplicity $\|w_{1,i}^*\|_2 = \|w_{2,i}^*\|_2 = 1$ and $|a_{r,i}^*| \le 1$.[6] We denote by

$$\mathfrak{C}_\varepsilon(\phi, R) := \max_{j \in [p]}\{\mathfrak{C}_\varepsilon(\phi_j, R)\} \quad \text{and} \quad \mathfrak{C}_\mathfrak{s}(\phi, R) := \max_{j \in [p]}\{\mathfrak{C}_\mathfrak{s}(\phi_j, R)\}$$

the complexity of $F^*$ and assume they are bounded.

In the *agnostic PAC-learning language*, our concept class consists of all functions $F^*$ in the form of (3.1) with complexity bounded by threshold $C$ and parameter $p$ bounded by threshold $p_0$. Let $\mathsf{OPT} = \mathbb{E}[L(F^*(x), y)]$ be the population risk achieved by the *best* target function in this concept class. Then, our goal is to learn this concept class with population risk $\mathsf{OPT} + \varepsilon$ using sample and time complexity *polynomial* in $C$, $p_0$ and $1/\varepsilon$. In the remainder of this paper, to simplify notations, we do not explicitly define this concept class parameterized by $C$ and $p$. Instead, we equivalently state our theorem with respect to any (unknown) target function $F^*$ with specific parameters $C$ and $p$ satisfying $\mathsf{OPT} = \mathbb{E}[L(F^*(x), y)]$. We assume $\mathsf{OPT} \in [0, 1]$ for simplicity.

*Remark.* Standard two-layer networks $f_r^*(x) = \sum_{i=1}^p a_{r,i}^* \phi(\langle w_{1,i}^*, x\rangle)$ are special cases of (3.1) (by setting $w_{2,i}^* = (0, \dots, 0, 1)$ and $\phi_i = \phi$). Our formulation (3.1) additionally captures combinations of correlations between *non-linear* and linear measurements of different directions of $x$.

**Learner network $F(x; W)$.** Using a data set $\mathcal{Z} = \{z_1, \dots, z_N\}$ of $N$ i.i.d. samples from $\mathcal{D}$, we train a network $F = (f_1, \cdots, f_k)\colon \mathbb{R}^d \to \mathbb{R}^k$ with

$$f_r(x) := \sum_{i=1}^m a_{r,i}\sigma(\langle w_i, x\rangle + b_i) = a_r^\top \sigma(Wx + b) \tag{3.2}$$

where $\sigma$ is the ReLU activation, $W = (w_1, \dots, w_m) \in \mathbb{R}^{m \times d}$ is the hidden weight matrix, $b \in \mathbb{R}^m$ is the bias vector, and $a_r \in \mathbb{R}^m$ is the output weight vector. To simplify analysis, we only update $W$ and keep $b$ and $a_r$ at initialization values. For such reason, we write the learner network as $f_r(x; W)$ and $F(x; W)$. We sometimes use $b^{(0)} = b$ and $a_r^{(0)} = a_r$ to emphasize they are randomly initialized. Our goal is to learn a weight matrix $W$ with population risk $\mathbb{E}[L(F(x; W), y)] \le \mathsf{OPT} + \varepsilon$.

**Learning Process.** Let $W^{(0)}$ denote the initial value of the hidden weight matrix, and let $W^{(0)} + W_t$ denote the value at time $t$. (Note that $W_t$ is the matrix of *increments*.) The weights are initialized with Gaussians and then $W$ is updated by the vanilla SGD. More precisely,

- entries of $W^{(0)}$ and $b^{(0)}$ are i.i.d. random Gaussians from $\mathcal{N}(0, 1/m)$,
- entries of each $a_r^{(0)}$ are i.i.d. random Gaussians from $\mathcal{N}(0, \varepsilon_a^2)$ for some fixed $\varepsilon_a \in (0, 1]$.[7]

At time $t$, SGD samples $z = (x, y) \sim \mathcal{Z}$ and updates $W_{t+1} = W_t - \eta\nabla L(F(x; W^{(0)} + W_t), y)$.

## 3.1 Main Theorem

For notation simplicity, with high probability (or w.h.p.) means with probability $1 - e^{-c\log^2 m}$ for a sufficiently large constant $c$, and $\widetilde{O}$ hides factors of $\mathsf{polylog}(m)$.

**Theorem 1** (two-layer). *For every $\varepsilon \in \left(0, \frac{1}{pk\mathfrak{C}_\mathfrak{s}(\phi, 1)}\right)$, there exists*

$$M_0 = \mathsf{poly}(\mathfrak{C}_\varepsilon(\phi, 1), 1/\varepsilon) \quad \text{and} \quad N_0 = \mathsf{poly}(\mathfrak{C}_\mathfrak{s}(\phi, 1), 1/\varepsilon)$$

*such that for every $m \ge M_0$ and every $N \ge \widetilde{\Omega}(N_0)$, choosing $\varepsilon_a = \varepsilon/\widetilde{\Theta}(1)$ for the initialization, choosing learning rate $\eta = \widetilde{\Theta}\left(\frac{1}{\varepsilon km}\right)$ and*

$$T = \widetilde{\Theta}\left(\frac{(\mathfrak{C}_\mathfrak{s}(\phi, 1))^2 \cdot k^3 p^2}{\varepsilon^2}\right) ,$$

*with high probability over the random initialization, SGD after $T$ iteration satisfies*

$$\mathbb{E}_{sgd}\left[\frac{1}{T}\sum_{t=0}^{T-1}\mathbb{E}_{(x,y)\sim\mathcal{D}}L(F(x;W^{(0)}+W_t),y)\right] \leq \mathsf{OPT}+\varepsilon.$$

**Example 3.1.** For functions such as $\phi(z)=e^z,\sin z,\mathrm{sigmoid}(z),\tanh(z)$ or low degree polynomials, using Example 2.1, our theorem indicates that for target networks with such activation functions, we can learn them using two-layer ReLU networks with

$$\text{size } m = \frac{\mathsf{poly}(k,p)}{\mathsf{poly}(\varepsilon)} \text{ and sample complexity } \min\{N,T\} = \frac{\mathsf{poly}(k,p,\log m)}{\varepsilon^2}$$

We note sample complexity $T$ is (almost) independent of $m$, the amount of overparametrization.

### 3.2   Our Interpretations

**Overparameterization improves generalization.**   By increasing $m$, Theorem 1 supports more target functions with possibly larger size, more complex activations, and smaller population risk OPT. In other words, when $m$ is fixed, among the class of target functions whose complexities are captured by $m$, SGD can learn the best function approximator of the data, with the smallest population risk. This gives intuition how overparameterization improves test error, see Figure 1(a).

**Large margin non-linear classifier.**   Theorem 1 is a nonlinear analogue of the margin theory for linear classifiers. The target function with a small population risk (and of bounded norm) can be viewed as a "large margin non-linear classifier." In this view, Theorem 1 shows that assuming the existence of such large-margin classifier, SGD finds a good solution with sample complexity mostly determined by the margin, instead of the *dimension* of the data.

**Inductive bias.**   Recent works (e.g., [4, 32]) show that when the network is heavily overparameterized (that is, $m$ is polynomial in the number of training samples) and no two training samples are identical, then SGD can find a global optimum with $0$ classification error (or find a solution with $\varepsilon$ training loss) in polynomial time. This does not come with generalization, since it can even fit random labels. Our theorem, combined with [4], confirms the inductive bias of SGD for two-layer networks: when the labels are random, SGD finds a network that memorizes the training data; when the labels are (even only approximately) realizable by some target network, then SGD learns and generalizes. This gives an explanation towards the well-known empirical observations of such inductive bias (e.g., [53]) in the two-layer setting, and is more general than Brutzkus et al. [14] in which the target network is only linear.

## 4   Result for Three-Layer Networks

**Concept class and target function $F^*(x)$.**   This time we consider more powerful target functions $F^* = (f_1^*, \cdots, f_k^*)$ of the form

$$f_r^*(x) := \sum_{i\in[p_1]} a_{r,i}^* \Phi_i\left(\sum_{j\in[p_2]} v_{1,i,j}^* \phi_{1,j}(\langle w_{1,j}^*, x\rangle)\right)\left(\sum_{j\in[p_2]} v_{2,i,j}^* \phi_{2,j}(\langle w_{2,j}^*, x\rangle)\right) \qquad (4.1)$$

where each $\phi_{1,j},\phi_{2,j},\Phi_i\colon \mathbb{R}\to\mathbb{R}$ is infinite-order smooth, and the weights $w_{1,i}^*,w_{2,i}^* \in \mathbb{R}^d$, $v_{1,i}^*,v_{2,i}^* \in \mathbb{R}^{p_2}$ and $a_{r,i}^* \in \mathbb{R}$ satisfy $\|w_{1,j}^*\|_2 = \|w_{2,j}^*\|_2 = \|v_{1,i}^*\|_2 = \|v_{2,i}^*\|_2 = 1$ and $|a_{r,i}^*| \leq 1$. Let

$$\mathfrak{C}_\varepsilon(\phi,R) = \max_{j\in[p_2],s\in[1,2]}\{\mathfrak{C}_\varepsilon(\phi_{s,j},R)\}, \qquad \mathfrak{C}_\varepsilon(\Phi,R) = \max_{j\in[p_1]}\{\mathfrak{C}_\varepsilon(\Phi_j,R)\}$$
$$\mathfrak{C}_\mathfrak{s}(\phi,R) = \max_{j\in[p_2],s\in[1,2]}\{\mathfrak{C}_\mathfrak{s}(\phi_{s,j},R)\}, \qquad \mathfrak{C}_\mathfrak{s}(\Phi,R) = \max_{j\in[p_1]}\{\mathfrak{C}_\mathfrak{s}(\Phi_j,R)\}$$

to denote the complexity of the two layers, and assume they are bounded.

Our concept class contains measures of correlations between *composite non-linear* functions and *non-linear* functions of the input, there are plenty of functions in this new concept class that may not necessarily have small-complexity representation in the previous formulation (3.1), and as we shall see in Figure 1(a), this is the **critical advantage** of using three-layer networks compared to two-layer ones or their NTKs. The learnability of this correlation is due to the *non-convex* interactions between hidden layers. As a comparison, [15] studies the regime where the changes in hidden layers are negligible thus *can not* show how to learn this concept class with a three-layer network.

*Remark* 4.1. Standard three-layer networks

$$f_r^*(x) = \sum_{i \in [p_1]} a_{r,i}^* \Phi_i \left( \sum_{j \in [p_2]} v_{i,j}^* \phi_j(\langle w_j^*, x \rangle) \right)$$

are only special cases of (4.1). Also, even in the special case of $\Phi_i(z) = z$, the target

$$f_r^*(x) = \sum_{i \in [p_1]} a_{r,i}^* \left( \sum_{j \in [p_2]} v_{1,i,j}^* \phi_1(\langle w_{1,j}^*, x \rangle) \right) \left( \sum_{j \in [p_2]} v_{2,i,j}^* \phi_2(\langle w_{2,j}^*, x \rangle) \right)$$

captures combinations of correlations of *non-linear* measurements in different directions of $x$.

**Learner network $F(x; W, V)$.** Our learners are three-layer networks $F = (f_1, \ldots, f_k)$ with

$$f_r(x) = \sum_{i \in [m_2]} a_{r,i} \sigma(n_i(x) + b_{2,i}) \quad \text{where each } n_i(x) = \sum_{j \in [m_1]} v_{i,j} \sigma(\langle w_j, x \rangle + b_{1,j})$$

The first and second layers have $m_1$ and $m_2$ hidden neurons. Let $W \in \mathbb{R}^{m_1 \times d}$ and $V \in \mathbb{R}^{m_2 \times m_1}$ represent the weights of the first and second hidden layers respectively, and $b_1 \in \mathbb{R}^{m_1}$ and $b_2 \in \mathbb{R}^{m_2}$ represent the corresponding bias vectors, $a_r \in \mathbb{R}^{m_2}$ represent the output weight vector.

## 4.1 Learning Process

Again for simplicity, we only update $W$ and $V$. The weights are randomly initialized as:

- entries of $W^{(0)}$ and $b_1 = b_1^{(0)}$ are i.i.d. from $\mathcal{N}(0, 1/m_1)$,
- entries of $V^{(0)}$ and $b_2 = b_2^{(0)}$ are i.i.d. from $\mathcal{N}(0, 1/m_2)$,
- entries of each $a_r = a_r^{(0)}$ are i.i.d. from $\mathcal{N}(0, \varepsilon_a^2)$ for $\varepsilon_a = 1$.

As for the optimization algorithm, we use SGD with weight decay and an explicit regularizer.

For some $\lambda \in (0, 1]$, we will use $\lambda F(x; W, V)$ as the learner network, i.e., linearly scale $F$ down by $\lambda$. This is equivalent to replacing $W$, $V$ with $\sqrt{\lambda} W$, $\sqrt{\lambda} V$, since a ReLU network is positive homogenous. The SGD will start with $\lambda = 1$ and slowly decrease it, similar to weight decay.[8]

We also use an explicit regularizer for some $\lambda_w, \lambda_v > 0$ with[9]

$$R(\sqrt{\lambda} W, \sqrt{\lambda} V) := \lambda_v \|\sqrt{\lambda} V\|_F^2 + \lambda_w \|\sqrt{\lambda} W\|_{2,4}^4 \ .$$

Now, in each round $t = 1, 2, \ldots, T$, we use (noisy) SGD to minimize the following stochastic objective for some fixed $\lambda_{t-1}$:

$$L_2(\lambda_{t-1}; W', V') := L\left( \lambda_{t-1} F(x; W^{(0)} + W^\rho + \mathbf{\Sigma} W', V^{(0)} + V^\rho + V' \mathbf{\Sigma}) \right)$$
$$+ R(\sqrt{\lambda_{t-1}} W', \sqrt{\lambda_{t-1}} V') \tag{4.2}$$

Above, the objective is stochastic because (1) $z \sim \mathcal{Z}$ is a random sample from the training set, (2) $W^\rho$ and $V^\rho$ are two small perturbation random matrices with entries i.i.d. drawn from $\mathcal{N}(0, \sigma_w^2)$ and $\mathcal{N}(0, \sigma_v^2)$ respectively, and (3) $\mathbf{\Sigma} \in \mathbb{R}^{m_1 \times m_1}$ is a random diagonal matrix with diagonals i.i.d. uniformly drawn from $\{+1, -1\}$. We note that the use of $W^\rho$ and $V^\rho$ is standard for Gaussian smoothing on the objective (and not needed in practice).[10] The use of $\Sigma$ may be reminiscent of the Dropout technique [46] in practice which randomly masks out neurons, and can also be removed.[11]

**Algorithm 1** SGD for three-layer networks (second variant (4.2))

---

**Input:** Data set $\mathcal{Z}$, initialization $W^{(0)}, V^{(0)}$, step size $\eta$, number of inner steps $T_w, \sigma_w, \sigma_v, \lambda_w, \lambda_v$.

1: $W_0 = 0, V_0 = 0, \lambda_1 = 1, T = \Theta\big(\eta^{-1} \log \frac{\log(m_1 m_2)}{\varepsilon_0}\big)$.

2: **for** $t = 1, 2, \ldots, T$ **do**

3:     Apply noisy SGD with step size $\eta$ on the stochastic objective $L_2(\lambda_{t-1}; W, V)$ for $T_w$ steps; the starting point is $W = W_{t-1}, V = V_{t-1}$ and suppose it reaches $W_t, V_t$.          $\diamond$ *see Lemma A.9*

4:     $\lambda_{t+1} = (1 - \eta)\lambda_t$.          $\diamond$ *weight decay*

5: **end for**

6: Randomly sample $\widehat{\Sigma}$ with diagonal entries i.i.d. uniform on $\{1, -1\}$

7: Randomly sample $\widetilde{\Theta}(1/\varepsilon_0^2)$ many noise matrices $\{W^{\rho,j}, V^{\rho,j}\}$. Let

$$j^* = \arg\min_j \left\{ \mathbb{E}_{z \in \mathcal{Z}} L\left(\lambda_T F\big(x; W^{(0)} + W^{\rho,j} + \widehat{\Sigma}W_T, V^{(0)} + V^{\rho,j} + V_T\widehat{\Sigma}\big)\right)\right\}$$

8: Output $W_T^{(out)} = W^{(0)} + W^{\rho,j^*} + \widehat{\Sigma}W_T, V_T^{(out)} = V^{(0)} + V^{\rho,j^*} + V_T\widehat{\Sigma}$.

---

Algorithm 1 presents the details. Specifically, in each round $t$, Algorithm 1 starts with weight matrices $W_{t-1}, V_{t-1}$ and performs $T_w$ iterations. In each iteration it goes in the negative direction of the stochastic gradient $\nabla_{W',V'} L_2(\lambda_t; W', V')$. Let the final matrices be $W_t, V_t$. At the end of this round $t$, Algorithm 1 performs weight decay by setting $\lambda_t = (1 - \eta)\lambda_{t-1}$ for some $\eta > 0$.

### 4.2 Main Theorems

For notation simplicity, with high probability (or w.h.p.) means with probability $1 - e^{-c \log^2(m_1 m_2)}$ and $\widetilde{O}$ hides factors of $\mathsf{polylog}(m_1, m_2)$.

**Theorem 2** (three-layer, second variant). *Consider Algorithm 1. For every* constant $\gamma \in (0, 1/4]$, *every* $\varepsilon_0 \in (0, 1/100]$, *every* $\varepsilon = \frac{\varepsilon_0}{kp_1 p_2^2 \mathfrak{C}_\mathfrak{s}(\Phi, p_2 \mathfrak{C}_\mathfrak{s}(\phi,1))\mathfrak{C}_\mathfrak{s}(\phi,1)^2}$, *there exists*

$$M = \mathsf{poly}\left(\mathfrak{C}_\varepsilon(\Phi, \sqrt{p_2}\mathfrak{C}_\varepsilon(\phi, 1)), \frac{1}{\varepsilon}\right)$$

*such that for every* $m_2 = m_1 = m \geq M$, *and properly set* $\lambda_w, \lambda_v, \sigma_w, \sigma_v$ *in Table 1, as long as*

$$N \geq \widetilde{\Omega}\left(\left(\frac{\mathfrak{C}_\varepsilon(\Phi, \sqrt{p_2}\mathfrak{C}_\varepsilon(\phi, 1)) \cdot \mathfrak{C}_\varepsilon(\phi, 1) \cdot \sqrt{p_2}p_1 k^2}{\varepsilon_0}\right)^2\right)$$

*there is a choice* $\eta = 1/\mathsf{poly}(m_1, m_2)$ *and* $T = \mathsf{poly}(m_1, m_2)$ *such that with probability* $\geq 99/100$,

$$\mathbb{E}_{(x,y) \sim \mathcal{D}} L(\lambda_T F(x; W_T^{(out)}, V_T^{(out)}), y) \leq (1 + \gamma)\mathsf{OPT} + \varepsilon_0.$$

### 4.3 Our Contributions

Our sample complexity $N$ scales polynomially with the complexity of the target network, and is *(almost) independent* of $m$, the amount of overparameterization. This itself can be quite surprising, because recent results on neural network generalization [6, 11, 24, 39] require $N$ to be polynomial in $m$. Furthermore, Theorem 2 shows three-layer networks can efficiently learn a *bigger* concept class (4.1) comparing to what we know about two-layer networks (3.1).

From a practical standpoint, one can construct target functions of the form (4.1) that cannot be (efficiently) approximated by any two-layer target function in (3.1). If data is generated according to such functions, then it may be necessary to three-layer networks as learners (see Figure 1).

From a theoretical standpoint, even in the special case of $\Phi(z) = z$, our target function can capture *correlations* between non-linear measurements of the data (recall Remark 4.1). This means $\mathfrak{C}_\varepsilon(\Phi, \mathfrak{C}_\varepsilon(\phi, 1)\sqrt{p_2}) \approx O(\sqrt{p_2}\mathfrak{C}_\varepsilon(\phi, 1))$, so learning it is essentially in the *same complexity* as learning each $\phi_{s,j}$. For example, a three-layer network can learn $\cos(100\langle w_1^*, x\rangle) \cdot e^{100\langle w_2^*, x\rangle}$ up to accuracy $\varepsilon$ in complexity $\mathsf{poly}(1/\varepsilon)$, while it is unclear how to do so using two-layer networks.

**Technical Contributions.**    We highlight some technical contributions in the proof of Theorem 2.

In recent results on the training convergence of neural networks for more than two layers [3, 4], the optimization process stays in a close neighborhood of the initialization so that, with heavy overparameterization, the network becomes "linearized" and the interactions across layers are negligible. In our three-layer case, this means that the matrix $W$ never interacts with $V$. They then argue that

SGD simulates a neural tangent kernel so the learning process is almost convex [27]. In our analysis, we directly tackle *non-convex* interactions between $W$ and $V$, by studying a "quadratic approximation" of the network. (See Remark 6.1 for a mathematical comparison.) Our new proofs techniques that could be useful for future theoretical applications.

Also, for the results [3, 4] and our two-layer Theorem 1 to hold, it suffices to analyze a regime where the "sign pattern" of ReLUs can be replaced with that of the random initialization. (Recall $\sigma(x) = \mathbb{I}_{x \geq 0} \cdot x$ and we call $\mathbb{I}_{x \geq 0}$ the "sign pattern.") In our three-layer analysis, the optimization process has moved *sufficiently away from initialization*, so that the sign pattern change can significantly affect output. This brings in additional technical challenge because we have to tackle non-convex interactions between $W$ and $V$ together with changing sign patterns.[12]

**Comparison to Daniely [15].** Daniely [15] studies the learnability of multi-layer networks when (essentially) only the *output layer* is trained, which reduces to a convex task. He shows that multi-layer networks can learn a compositional kernel space, which implies two/three-layer networks can efficiently learn low-degree polynomials. He did not derive the general sample/time complexity bounds for more complex functions such as those in our concept classes (3.1) and (4.1), but showed that they are finite.

In contrast, our learnability result of concept class (4.1) is due to the *non-convex* interaction between hidden layers. Since Daniely [15] studies the regime when the changes in hidden layers are negligible, if three layer networks are used, to the best of our knowledge, their theorem *cannot* lead to similar sample complexity bounds comparing to Theorem 2 by only training the last layer of a three-layer network. Empirically, one can also observe that training hidden layers is better than training the last layer (see Figure 1).

## 5   Conclusion and Discussion

We show by training the *hidden layers* of two-layer (resp. three-layer) overparameterized neural networks, one can efficiently learn some important concept classes including two-layer (resp. three-layer) networks equipped with smooth activation functions. Our result is in the agnostic PAC-learning language thus is *distribution-free*. We believe our work *opens up* a new direction in both algorithmic and generalization perspectives of overparameterized neural networks, and pushing forward can possibly lead to more understanding about deep learning.

Our results apply to other more structured neural networks. As a concrete example, consider convolutional neural networks (CNN). Suppose the input is a two dimensional matrix $x \in \mathbb{R}^{d \times s}$ which can be viewed as $d$-dimensional vectors in $s$ *channels*, then a convolutional layer on top of $x$ is defined as follows. There are $d'$ fixed subsets $\{S_1, S_2, \ldots, S_{d'}\}$ of $[d]$ each of size $k'$. The output of the convolution layer is a matrix of size $d' \times m$, whose $(i, j)$-th entry is $\phi(\langle w_j, x_{S_i} \rangle)$, where $x_{S_i} \in \mathbb{R}^{k' \times s}$ is the submatrix of $x$ with rows indexed by $S_i$; $w_j \in \mathbb{R}^{k' \times s}$ is the weight matrix of the $j$-th channel; and $\phi$ is the activation function. Overparameterization then means a larger number of channels $m$ in our learned network comparing to the target. Our analysis can be adapted to show a similar result for this type of networks.

One can also combine this paper with properties of recurrent neural networks (RNNs) [3] to derive PAC-learning results for RNNs [2], or use the existential tools of this paper to derive PAC-learning results for three-layer residual networks (ResNet) [1]. The latter gives a provable separation between neural networks and kernels in the efficient PAC-learning regime.

### Acknowledgements

This work was supported in part by FA9550-18-1-0166. Y. Liang would also like to acknowledge that support for this research was provided by the Office of the Vice Chancellor for Research and Graduate Education at the University of Wisconsin-Madison with funding from the Wisconsin Alumni Research Foundation.

## Footnotes

[3]In contrast, Daniely [15] focuses on training essentially only the last layer (and the hidden-layer movement is negligible). After this paper has appeared online, Arora et al. [8] showed that neural networks can provably learn two-layer networks with a slightly weaker class of smooth activation functions. Namely, the activation functions that are either linear functions or even functions.

[4]Recall $\left( \frac{\sqrt{\log(1/\varepsilon)}}{\sqrt{i}} C^* \right)^i \leq e^{O(\log(1/\varepsilon))} = \frac{1}{\mathsf{poly}(\varepsilon)}$ for every $i \geq 1$.

[5]$\frac{1}{2}$ can always be padded to the last coordinate, and $\|x\|_2 = 1$ can always be ensured from $\|x\|_2 \leq 1$ by padding $\sqrt{1 - \|x\|_2^2}$. This assumption is for simplifying the presentation.

[6]For general $\|w_{1,i}^*\|_2 \le B, \|w_{2,i}^*\|_2 \le B, |a_{r,i}^*| \le B$, the scaling factor $B$ can be absorbed into the activation function $\phi'(x) = \phi(Bx)$. Our results then hold by replacing the complexity of $\phi$ with $\phi'$.

[7]We shall choose $\varepsilon_a = \widetilde{\Theta}(\varepsilon)$ in the proof due to technical reason. As we shall see in the three-layer case, if weight decay is used, one can relax this to $\varepsilon_a = 1$.

[8]We illustrate the technical necessity of adding weight decay. During training, it is *easy* to add new information to the current network, but *hard* to forget "false" information that is already in the network. Such false information can be accumulated from randomness of SGD, non-convex landscapes, and so on. Thus, by scaling down the network we can effectively forget false information.

[9]This $\| \cdot \|_{2,4}$ norm on $W$ encourages weights to be more evenly distributed across neurons. It can be replaced with $\|\sqrt{\lambda_{t-1}} W_{t-1}\|_{2,2+\alpha}^{2+\alpha}$ for any constant $\alpha > 0$ for our theoretical purpose. We choose $\alpha = 2$ for simplicity, and observe that in practice, weights are automatically spread out due to data randomness, so this explicit regularization may not be needed. See Section 7.1 for an experiment.

[10]Similar to known non-convex literature [19] or smooth analysis, we introduce Gaussian perturbation $W^\rho$ and $V^\rho$ for theoretical purpose and it is not needed in practice. Also, we apply noisy SGD which is the vanilla SGD plus Gaussian perturbation, which again is needed in theory but believed unnecessary for practice [19].

[11]In the full paper we study two variants of SGD. This present version is the "second variant," and the first variant $L_1(\lambda_{t-1}; W', V')$ is the same as (4.2) by removing $\Sigma$. Due to technical difficulty, the best sample complexity we can prove for $L_1$ is a bit higher.

[12]For instance, the number of sign changes can be $m^{0.999}$ for the second hidden layer (see Lemma 6.5). In this region, the network output can be affected by $m^{0.499}$ since each neuron is of value roughly $m^{-1/2}$. Therefore, if after training we replace the sign pattern with random initialization, the output will be meaningless.

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
