[Supplementary Material]

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

[13]One technical issue is the following. When considering multiple neurons of the second layer, those Gaussian variables $\rho$ are not independent because they all depend on $W^{(0)}$. The notion of $\widetilde{\rho}$ in Lemma 6.4 removes such dependency across multiple neurons, at the expense of small Wasserstein distance.

[14]Recall that our sample complexity in Theorem 2 has the form $\mathfrak{C}(\Phi, \mathfrak{C}(\phi))$ ignoring other parameters. But in fact, one can be more careful on this target function and derive a sample complexity of the form $\mathfrak{C}(\Phi, \mathfrak{C}(\phi_1)) \cdot \mathfrak{C}(\phi_2)$, where $\Phi(x) = (x-2)^2$, $\phi_1(x) = \sin(3x)$ and $\phi_2(x) = \cos(7x)$. Because of this, to show the best contrast between two and three-layer networks, we choose constant 7 on $\phi_2$ so that $\mathfrak{C}(\phi_2) \approx \mathfrak{C}(\phi_1)^2$.

[15]The original proof of [19] was for constant probability but it is easy to change it to $1 - \delta$ at the expense of paying a polynomial factor in $1/\delta$. The original proof did not state the objective "non-increasing" guarantee but it is easy to show it for mini-batch SGD. Recall $\eta = \mathsf{poly}(d, B, 1/\varepsilon, 1/\delta)$ in [19] so in each iteration, the original SGD may increase the objective by no more than $\frac{\eta^2 B}{b}$ if a batch size $b$ is used. If $b$ is polynomially large, this goal is easily achievable. In practice, however, using a very small batch size suffices.

[16]There are slightly different versions of the contraction lemma in the literature. For the scalar case without absolute value, see [35, Section 3.8]; for the scalar case with absolute value, see [10, Theorem 12]; and for the

[17]We note that [35, Theorem 43] did not have $W^{(0)}$ but the same result holds with the introduction of $W^{(0)}$.

[18] Indeed, with high probability $|g_r(x; W)| \le \widetilde{O}(\varepsilon_a)$ and since $\|W_t\|_{2,\infty} \le \tau_{w,\infty}$ we have $|g_r^{(b)}(x; W_t)| \le \widetilde{O}(\varepsilon_a \tau_{w,\infty} m)$. Together we have $|g_r(x; W + W_t)| \le \widetilde{O}(\varepsilon_a \tau_{w,\infty} m)$. By the coupling Lemma B.3b, this implies $|f_r(x; W + W_t)| \le \widetilde{O}(\varepsilon_a \tau_{w,\infty} m)$ as well. Using $L(0, y) \in [0, 1]$ and the 1-Lipschitz continuity finishes the proof.

[19] In some literature this issue was simply ignored or an absolute bound on $L$ is imposed; however, the only globally absolutely bounded convex function is constant.

[20]This is possible for the following reason. Let $x^\perp = (\sqrt{1-x_1^2}, -x_1)$ be unit vector orthogonal to $x$. We can write $w_0 = \alpha x + \beta x^\perp$ where $\alpha, \beta \sim \mathcal{N}(0,1)$ are two independent Gaussians.

[21]More specifically, $\alpha_{i,j} = \alpha_{i,j}(v_i^{(0)}, W^{(0)}, b_1^{(0)})$ and $\beta_i = \beta_i(x, v_i^{(0)}, W^{(0)}, b_1^{(0)})$ depend on the randomness of $v_i^{(0)}, W^{(0)}$ and $b_1^{(0)}$.

[22] All the variables considered in this section is not absolutely bounded, but only with high probability with a Gaussian tail. Strictly speaking, when apply this Theorem we should be first replaced $B_j^{\mathsf{s}}$ by $B_j^{\mathsf{s}} \mathbb{1}_{\text{all } B_j \leq \widetilde{O}(1/\sqrt{m_2 p_2S})}$. We choose to avoid writing this truncation in the paper to simply the presentation.

[23] More specifically, we can choose $w$ from Lemma 6.3 as $\left(\alpha_{i,1}, \ldots, \alpha_{i,p_2}, \beta_i/\sqrt{1 - \sum_{j\in[p_2]} \phi_{1,j,\varepsilon}^2(x)}\right)$, choose $x$ from Lemma 6.3 as $\left(\phi_{1,1,\varepsilon}(x), \ldots, \phi_{1,p_2,\varepsilon}(x), \sqrt{1 - \sum_{j\in[p_2]} \phi_{1,j,\varepsilon}^2(x)}\right)$, choose $w^*$ from Lemma 6.3 as $(v_{1,1}^*, \ldots, v_{1,p_2}^*, 0)$, and choose $b_0$ from Lemma 6.3 as $b_{2,i}^{(0)}$.

[24]Recall from the proof of Claim C.3 that, we need to replace $m_1$ with $\frac{m_1}{p_2 S}$ and scale up $W^{(0)}$ and $b_1^{(0)}$ by $\sqrt{p_2 S}$ before applying Lemma 6.4.

[25]We have skipped the details since it is analogous to (C.17).

[26]We note that the derivation of $\|a_r D_{v,x}(V^{(0)} + V^\rho)\|_\infty \leq \widetilde{O}(1)$ may be non-trivial for some readers, because $D_{v,x}$ is dependent of the randomness of $V^{(0)} + V^{(\rho)}$. In fact, for every fixed basis vector $e_j$, we have w.h.p. $\|D_{v,x}(V^{(0)} + V^\rho)e_j\|_2 \leq \widetilde{O}(1)$, and thus by the randomness of $a_r$ it satisfies $|a_r D_{v,x}(V^{(0)} + V^\rho)e_j| \leq \widetilde{O}(1)$. Taking a union bound over $j = 1, 2, \ldots, m_2$ gives the bound. Such proof idea was repeatedly used in [4].

[27] Namely, to first show that each coordinate $i \in [m_2]$ satisfies $(D_{v,x,\rho,\eta} - D_{v,x,\rho})_{i,i} \ne 0$ with probability $\widetilde{O}\left( \eta \frac{(\frac{1}{\sqrt{m_1}} \tau_{v,\infty} + \tau_{w,\infty})}{\sigma_v} \right)$. Then, since we can ignoring terms of magnitude $O_p(\eta^3)$, it suffices to consider the case of $\|D_{v,x,\rho,\eta} - D_{v,x,\rho}\|_0 = 1$, which occurs with probability at most $m_2 \times \widetilde{O}\left( \eta \frac{(\frac{1}{\sqrt{m_1}} \tau_{v,\infty} + \tau_{w,\infty})}{\sigma_v} \right)$ by union bound. Finally, each coordinate changes by at most $\widetilde{O}\left( \eta \left( \frac{1}{\sqrt{m_1}} \tau_{v,\infty} + \tau_{w,\infty} \right) \right)$ by the argument above.

[28]When convoluted with Gaussian noise $\nabla(f * g) = f * \nabla g$, every bounded function $f$ becomes infinite-order differentiable with parameter $B$ inversely-polynomially dependent on the noise level $g$.

[29]Indeed, with our parameter choices in Table 1, the spectral norms $\sqrt{\lambda_T}\|W^{(0)} + W^{\rho,j} + W_T\|_2 \leq O(1) + \|\sqrt{\lambda_T}W_T\|_F \leq O(1 + m_1^{1/4}\tau_w') \leq O(1)$ and $\sqrt{\lambda_T}\|V^{(0)} + V^{\rho,j} + V_T\|_2 \leq O(1) + \|\sqrt{\lambda_T}V_T\|_F \leq O(1 + \tau_v') \leq O(1)$. Therefore, the network output $\lambda_T F(x; W^{(0)} + W^{\rho,j} + W_T, V^{(0)} + V^{\rho,j} + V_T)$ must be bounded by $\widetilde{O}(\sqrt{km_2})$ in Euclidean norm. By the assumption that $L(0, y) \in [0, 1]$ and $L(\cdot, y)$ is 1-Lipschitz continuous in the first variable, we have that $L_F$ is bounded as stated.

[30]Indeed, $|f_r(x; W^{(0)} + W^{\rho,j}, V^{(0)} + V^{\rho,j})| \le \widetilde{O}(1)$ with high probability, and $\lambda_T |g_r^{(b,b)}(x; W_T, V_T)| \le \widetilde{O}(\sqrt{m_2} \|\sqrt{\lambda_T} V_T\|_2 \|\sqrt{\lambda_T} W_T\|_2) \le \widetilde{O}(\sqrt{m_2} \|\sqrt{\lambda_T} V_T\|_F \|\sqrt{\lambda_T} W_T\|_F) \le \widetilde{O}(\varepsilon_0^{3/4} \sqrt{m_2} m_1^{1/4} \tau_w' \tau_v') \le \widetilde{O}(C_0)$ by spectral norm bounds.

[31]Strictly speaking, Corollary A.11 requires an absolute value bound $b$ as opposed to a high probability bound. It is a simple exercise to deal with this issue, see for instance Remark B.6 in our two-layer proof.

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

# PROOF SKETCH

We present key technical lemmas we used for proving the three-layer network theorems in Section 6 so that interested readers do not need to go through the appendix. Many of them can be of independent interests and have found further applications beyond this paper (such as for residual networks [1] and for recurrent networks [2]). The full proof is in Appendix C.

The two-layer result is based on similar ideas but simpler, and the full proof is in Appendix B.

We give more experiments in Section 7 on Page 21. Our appendix starts on page 23.

## 6  Proof Overview for Three Layer Networks

In Section 6.1, we state the main theorem for the first variant of the SGD, which we excluded from the main body due to space limitation. In Section 6.2, we show the existence of some good "pseudo network" that can approximate the target. In Section 6.3, we present our coupling technique between a real network and a pseudo network. In Section 6.4, we present the key lemma about the optimization procedure. In Section 6.5, we state a simple generalization bound that is compatible with our algorithm. These techniques together give rise to the proof of Theorem 3. In Section 6.6, we present additional techniques needed to show Theorem 2.

### 6.1  First Variant of SGD

In the first variant of SGD, in each round $t = 1, 2, \ldots, T$, we use (noisy) SGD to minimize the following stochastic objective for some fixed $\lambda_{t-1}$:

$$L_1(\lambda_{t-1}; W', V') := L\left(\lambda_{t-1}F(x; W^{(0)} + W^\rho + W', V^{(0)} + V^\rho + V')\right) + R(\sqrt{\lambda_{t-1}}W', \sqrt{\lambda_{t-1}}V') \tag{6.1}$$

Above, the objective is stochastic because (1) $z \sim \mathcal{Z}$ is a random sample from the training set, and (2) $W^\rho$ and $V^\rho$ are two small perturbation matrices. This is only different from (4.2) by removing $\Sigma$. We include the pseudocode in Algorithm 2.

---

**Algorithm 2** SGD for three-layer networks (first variant (6.1))

---

**Input:** Data set $\mathcal{Z}$, initialization $W^{(0)}, V^{(0)}$, step size $\eta$, number of inner steps $T_w$, $\sigma_w, \sigma_v, \lambda_w, \lambda_v$.

1: $W_0 = 0, V_0 = 0, \lambda_1 = 1, T = \Theta\left(\eta^{-1}\log\frac{\log(m_1 m_2)}{\varepsilon_0}\right)$.
2: **for** $t = 1, 2, \ldots, T$ **do**
3:     Apply noisy SGD with step size $\eta$ on the stochastic objective $L_1(\lambda_{t-1}; W, V)$ for $T_w$ steps; the starting point is $W = W_{t-1}, V = V_{t-1}$ and suppose it reaches $W_t, V_t$.  $\diamond$ *see Lemma A.9*
4:     $\lambda_{t+1} = (1 - \eta)\lambda_t$.                                                               $\diamond$ *weight decay*
5: **end for**
6: Randomly sample $\widetilde{\Theta}(1/\varepsilon_0^2)$ many noise matrices $\{W^{\rho,j}, V^{\rho,j}\}$. Let

$$j^* = \arg\min_j \left\{\mathbb{E}_{z\in\mathcal{Z}} L\left(\lambda_T F(x; W^{(0)} + W^{\rho,j} + W_T, V^{(0)} + V^{\rho,j} + V_T)\right)\right\}$$

7: Output $W_T^{(out)} = W^{(0)} + W^{\rho,j^*} + W_T, V_T^{(out)} = V^{(0)} + V^{\rho,j^*} + V_T$.

---

Below is the main theorem for using the first variant of SGD to train three-layer networks.

**Theorem 3** (three-layer, first variant). *Consider Algorithm 2. In the same setting as Theorem 2, for every $m_2 = m_1 = m \geq M$, as long as*

$$N \geq \widetilde{\Omega}(Mm^{3/2})$$

*there is choice $\eta = 1/\mathsf{poly}(m_1, m_2)$, $T = \mathsf{poly}(m_1, m_2)$ such that with probability at least 99/100,*

$$\mathbb{E}_{(x,y)\sim\mathcal{D}} L(\lambda_T F(x; W_T^{(out)}, V_T^{(out)}), y) \le (1+\gamma)\mathsf{OPT} + \varepsilon_0.$$

As $m$ goes large, this sample complexity $N \approx m^{3/2}$ polynomially scales with $m$ so may not be very efficient (we did not try hard to improve the exponent $3/2$). Perhaps interesting enough, this is already *non-trivial*, because $N$ can be much smaller than $m^2$, the number of parameters of the network or equivalently the naive VC dimension bound. Recall in our second variant of SGD, the sample complexity $N$ only grows polylogarithmically in $m$.

## 6.2 Existence

We wish to show the existence of some good "pseudo network" that can approximate the target network. In a pseudo network, each ReLU activation $\sigma(x) = \mathbb{I}_{x\ge 0}x$ is replaced with $\mathbb{I}_{x^{(0)}\ge 0}x$ where $x^{(0)}$ is the value at *random initialization*. Formally, let

- $D_{w,x} \in \{0,1\}^{m_1 \times m_1}$ denote a diagonal sign matrix indicating the sign of the ReLU's for the first layer at *random initialization*, that is, $[D_{w,x}]_{i,i} = \mathbb{I}[\langle w_i^{(0)}, x\rangle + b_{1,i}^{(0)} \ge 0]$, and

- $D_{v,x} \in \{0,1\}^{m_2 \times m_2}$ denote the diagonal sign matrix of the second layer at random initialization.

Consider the output of a three-layer network at randomly initialized sign without bias as

$$g_r^{(0)}(x; W, V) := a_r D_{v,x} V D_{w,x} W x$$
$$G^{(0)}(x; W, V) := \left(g_1^{(0)}, \cdots, g_k^{(0)}\right)$$

*Remark* 6.1. The above pseudo network can be reminiscent of the simpler linearized NTK approximation of a network used in prior work [4, 27], which in our language means

$$a_r D_{v,x} V D_{w,x} W^{(0)} x + a_r D_{v,x} V^{(0)} D_{w,x} W x \ .$$

In such linearization it is clear that the weight matrices $W$ and $V$ do not interact with each other. In contrast, in our quadratic formula $a_r D_{v,x} V D_{w,x} W x$, the matrices $W$ and $V$ are multiplied together, resulting in a non-convex interaction after putting into the loss function. We shall see in Section 6.3 that the pseudo network can be made close to the real network in some sense.

**Lemma 6.2** (existence). *For every $\varepsilon \in \left(0, \frac{1}{kp_1 p_2^2 \mathfrak{C}_{\mathfrak{s}}(\Phi, p_2 \mathfrak{C}_{\mathfrak{s}}(\phi,1))\mathfrak{C}_{\mathfrak{s}}(\phi,1)^2}\right)$, there exists*

$$M = \mathsf{poly}\left(\mathfrak{C}_\varepsilon\big(\Phi, \sqrt{p_2}\mathfrak{C}_\varepsilon(\phi,1)\big), \frac{1}{\varepsilon}\right)$$
$$C_0 = \mathfrak{C}_\varepsilon(\Phi, \sqrt{p_2}\mathfrak{C}_\varepsilon(\phi,1)) \cdot \mathfrak{C}_\varepsilon(\phi,1) \cdot \widetilde{O}(p_1\sqrt{p_2}k)$$

*such that if $m_1, m_2 \ge M$, then with high probability, there exists weights $W^*, V^*$ with*

$$\|W^*\|_{2,\infty} = \max_i \|w_i^*\|_2 \le \frac{C_0}{m_1}, \quad \|V^*\|_{2,\infty} = \max_i \|v_i^*\|_2 \le \frac{\sqrt{m_1}}{m_2}$$

*such that*

$$\mathbb{E}_{(x,y)\sim\mathcal{D}}\left[\sum_{r=1}^k \left|f_r^*(x) - g_r^{(0)}(x; W^*, V^*)\right|\right] \le \varepsilon,$$

*and hence,*

$$\mathbb{E}_{(x,y)\sim\mathcal{D}}\left[L(G^{(0)}(x; W^*, V^*), y)\right] \le \mathsf{OPT} + \varepsilon.$$

In other words, at randomly initialized signs, there exist choices of $W^*$ and $V^*$ with small norms so that $G^{(0)}(x; W^*, V^*)$ approximates the target. Later, we will combine this with the coupling lemma Section 6.3 to show a main structural property of overparameterized networks: solutions with good population risks are *dense* in the parameter space, in the sense that with high probability over the random initialization, there exists a good solution in the "close" neighborhood of the initialized weights.

### 6.2.1 Technical Ideas

We begin with a simple task to illustrate the main idea. Let $w^* \in \mathbb{R}^d$ be a given vector and suppose we want to approximate function $\phi(\langle w^*, x \rangle)$ (over $x$) by designing a random function $\mathbb{I}_{\langle w, x \rangle + b_0 \geq 0} h(\langle w, w^* \rangle, b_0)$ (over $x$) where $w$ is a random Gaussian, $b_0$ is a random bias, and $h(\cdot, \cdot)$ is a function at our choice. The next lemma says that, we can design $h$ with good property so that the expectation of the random function $\mathbb{I}_{\langle w, x \rangle + b_0 \geq 0} h(\langle w, w^* \rangle, b_0)$ is close to $\phi(\langle w^*, x \rangle)$.

**Lemma 6.3** (indicator to function). *For every smooth function $\phi$, every $\varepsilon \in \left(0, \frac{1}{\mathfrak{C}_{\mathfrak{s}}(\phi, 1)}\right)$, there exists a function $h : \mathbb{R}^2 \to [-\mathfrak{C}_{\varepsilon}(\phi, 1), \mathfrak{C}_{\varepsilon}(\phi, 1)]$ that is also $\mathfrak{C}_{\varepsilon}(\phi, 1)$-Lipschitz continuous on its first coordinate with the following two (equivalent) properties:*

*(a) For every $x_1 \in [-1, 1]$:*
$$\left| \mathbb{E}\left[ \mathbb{I}_{\alpha_1 x_1 + \beta_1 \sqrt{1 - x_1^2} + b_0 \geq 0} h(\alpha_1, b_0) \right] - \phi(x_1) \right| \leq \varepsilon$$
*where $\alpha_1, \beta_1, b_0 \sim \mathcal{N}(0, 1)$ are independent random variables.*

*(b) For every $w^*, x \in \mathbb{R}^d$ with $\|w^*\|_2 = \|x\|_2 = 1$:*
$$\left| \mathbb{E}\left[ \mathbb{I}_{\langle w, x \rangle + b_0 \geq 0} h(\langle w, w^* \rangle, b_0) \right] - \phi(\langle w^*, x \rangle) \right| \leq \varepsilon$$
*where $w \sim \mathcal{N}(0, \mathbf{I})$ is an $d$-dimensional Gaussian, $b_0 \sim \mathcal{N}(0, 1)$.*

*Furthermore, $h$ satisfies $\mathbb{E}_{\alpha_1, b_0 \sim \mathcal{N}(0,1)}\left[ h(\alpha_1, b_0)^2 \right] \leq (\mathfrak{C}_{\mathfrak{s}}(\phi, 1))^2$.*

If one designs a vector $w^{\circledast} = \left(0, \ldots, 0, 2h(\langle w, w^* \rangle, b_0)\right)$, then using $x_d = 1/2$, Lemma 6.3 implies $\left| \mathbb{E}\left[ \mathbb{I}_{\langle w, x \rangle + b_0 \geq 0} \langle w^{\circledast}, x \rangle \right] - \phi(\langle w^*, x \rangle) \right| \leq \varepsilon$. Therefore, Lemma 6.3 corresponds to Lemma 6.2 in the special case of a *single* neuron.

*Remark.* The reason Lemma 6.3b is equivalent to Lemma 6.3a consists of a few thinking steps. Without loss of generality, one can assume $w^* = (1, 0, 0, \ldots, 0)$ and write $\phi(\langle w^*, x \rangle) = \phi(x_1)$ and write $h(\langle w, w^* \rangle, b_0) = h(w_1, b_0)$ so it only depends on the first coordinate of $w$ and $b_0$. Under such simplifications, the second through last coordinates do not make any difference, so we can assume without loss of generality that $w^*, w, x$ are only in 2 dimensions, and write $w = (\alpha_1, \beta_1)$ and $x = (x_1, \sqrt{1 - x_1^2})$. In sum, proving Lemma 6.3a suffices in establishing Lemma 6.3b.

Given Lemma 6.3, we can directly apply it to the two-layer case (see Appendix B.1) to show the existence of good pseudo networks. As for the three-layer case, we need to apply Lemma 6.3 twice: once for (each neuron of) the second hidden layer and once for the output.

Consider the the input (without bias) to a single neuron of the second hidden layer at random initialization. Without loss of generality, say the first neuron, given as:
$$n_1(x) = \sum_{i \in [m_1]} v_{1,i}^{(0)} \sigma\left( \langle w_i^{(0)}, x \rangle + b_{1,i}^{(0)} \right) \quad .$$

Even though $n_1(x)$ is completely random, using Lemma 6.3, we can derive the following lemma which rewrites $n_1(x)$ in the direction of an arbitrary function $\phi$.

**Lemma 6.4** (information out of randomness). *For every smooth function $\phi$, every $w^* \in \mathbb{R}^d$ with $\|w^*\|_2 = 1$, for every $\varepsilon \in \left(0, \frac{1}{\mathfrak{C}_{\mathfrak{s}}(\phi, 1)}\right)$, there exists real-valued functions*

$$\rho(v_1^{(0)}, W^{(0)}, b_1^{(0)}), \ B(x, v_1^{(0)}, W^{(0)}, b_1^{(0)}), R(x, v_1^{(0)}, W^{(0)}, b_1^{(0)}), \ and \ \phi_{\varepsilon}(x)$$

*such that for every $x$:*
$$n_1(x) = \rho\left( v_1^{(0)}, W^{(0)}, b_1^{(0)} \right) \phi_{\varepsilon}(x) + B\left( x, v_1^{(0)}, W^{(0)}, b_1^{(0)} \right) + R\left( x, v_1^{(0)}, W^{(0)}, b_1^{(0)} \right).$$

*Moreover, letting $C = \mathfrak{C}_{\varepsilon}(\phi, 1)$ be the complexity of $\phi$, and if $v_{1,i}^{(0)} \sim \mathcal{N}(0, \frac{1}{m_2})$ and $w_{i,j}^{(0)}, b_{1,i}^{(0)} \sim \mathcal{N}(0, \frac{1}{m_1})$ are at random initialization, then we have*

*1. For every fixed $x$, $\rho\left( v_1^{(0)}, W^{(0)}, b_1^{(0)} \right)$ is independent of $B\left( x, v_1^{(0)}, W^{(0)}, b_1^{(0)} \right)$.*

*2. $\rho\left( v_1^{(0)}, W^{(0)}, b_1^{(0)} \right) \sim \mathcal{N}\left( 0, \frac{1}{100 C^2 m_2} \right)$.*

3. *For every $x$ with $\|x\|_2 = 1$, $|\phi_\varepsilon(x) - \phi(\langle w^*, x\rangle)| \le \varepsilon$.*

4. *For every fixed $x$ with $\|x\|_2 = 1$, with high probability $\left| R\left(x, v_1^{(0)}, W^{(0)}, b_1^{(0)}\right) \right| \le \widetilde{O}\left(\frac{1}{\sqrt{m_1 m_2}}\right)$ and $\left| B\left(x, v_1^{(0)}, W^{(0)}, b_1^{(0)}\right) \right| \le \widetilde{O}\left(\frac{1}{\sqrt{m_2}}\right)$.*

*Furthermore, there exists real-valued function $\widetilde{\rho}(v_1^{(0)})$ satisfying with high probability:*

$$\widetilde{\rho}(v_1^{(0)}) \sim \mathcal{N}\left(0, \frac{1}{100 C^2 m_2}\right) \quad \text{and} \quad \mathcal{W}_2(\rho|_{W^{(0)}, b_1^{(0)}}, \widetilde{\rho}) \le \widetilde{O}\left(\frac{1}{C\sqrt{m_1 m_2}}\right) .$$

Lemma 6.4 shows that, up to some small noise $B$ and $R$, we can "view" the input to any neuron of the second layer essentially as "a Gaussian variable $\rho$" times "the target function $\phi(\langle w^*, x\rangle)$" in the first layer of the target network. This allows us to apply Lemma 6.3 again for the output layer, so as to construct for instance a composite function $\Phi(\cdot)$ with $\phi_i$ as its inputs.[13]

*Remark.* Lemma 6.4 may sound weird at first look because random initialization cannot carry any information about the target. There is no contradiction here, because we will show, $B$ is essentially another Gaussian (with the same distribution as $\rho$) times $\sqrt{\text{constant} - \phi^2(\langle w^*, x\rangle)}$, thus $n_1(x)$ can still be *independent* of the value of $\phi(\langle w^*, x\rangle)$. Nevertheless, the decomposition in Lemma 6.4 shall enable us to show that, when we start to modify the hidden weights $W, V$, the learning process will start to *discover* this structure and make the weight of the term relating to $\phi(\langle w^*, x\rangle)$ *stand out* from other terms.

Using Lemma 6.4 and applying Lemma 6.3 once more, we can prove Lemma 6.2.

### 6.3 Coupling Between Real and Pseudo Networks

Suppose we are currently at weights $W^{(0)} + W' + W^\rho$, $V^{(0)} + V' + V^\rho$, where matrices $W^\rho, V^\rho$ are random Gaussian matrices such that:

$$V_{i,j}^\rho \sim \mathcal{N}(0, \sigma_v^2) \quad \text{and} \quad W_{i,j}^\rho \sim \mathcal{N}(0, \sigma_w^2)$$

for some $\sigma_v, \sigma_w \in [1/(m_1 m_2), 1]$ to be specified later, and $W', V'$ are matrices with bounded norms that can depend on the randomness of $W^{(0)}, b_1^{(0)}, V^{(0)}, b_2^{(0)}$. Intuitively, $W'$ and $V'$ capture how much the algorithm has moved away from the initialization, while $W^\rho, V^\rho$ are introduced for adding smoothness in the optimization, see Section 6.4.

Let us introduce the notion of pseudo networks at the current weights. Let

- $D_{w,x}$ denote the diagonal sign matrix of the first layer at random initialization $W^{(0)}$,

- $D_{v,x}$ denote the diagonal sign matrix of the second layer at random initialization $W^{(0)}, V^{(0)}$,

- $D_{w,x} + D'_{w,x} \in \{0,1\}^{m_1 \times m_1}$ denote the diagonal sign matrix of the first layer at weights $W^{(0)} + W' + W^\rho$ and $V^{(0)} + V' + V^\rho$, i.e., $[D_{w,x} + D'_{w,x}]_{i,i} = \mathbb{I}[\langle w_i^{(0)} + w'_i + w_i^{(\rho)}, x\rangle + b_{1,i}^{(0)} \ge 0]$.

- $D_{v,x} + D'_{v,x} \in \{0,1\}^{m_2 \times m_2}$ denote the diagonal sign matrix of the second layer at these weights.

For a fixed $r \in [k]$, let us denote row vector $a_r = (a_{r,i})_{i \in [m_2]}$. Define the pseudo network (and its semi-bias, bias-free versions) as

$$g_r(x; W, V) = a_r(D_{v,x} + D'_{v,x})\left(V(D_{w,x} + D'_{w,x})(Wx + b_1) + b_2\right)$$
$$g_r^{(b)}(x; W, V) = a_r(D_{v,x} + D'_{v,x})V(D_{w,x} + D'_{w,x})(Wx + b_1)$$
$$g_r^{(b,b)}(x; W, V) = a_r(D_{v,x} + D'_{v,x})V(D_{w,x} + D'_{w,x})Wx$$

As a sanity check, at $W^{(0)} + W' + W^\rho$, $V^{(0)} + V' + V^\rho$ the pseudo network equals the true one:

$$g_r\left(x; W^{(0)} + W' + W^\rho, V^{(0)} + V' + V^\rho\right) = f_r\left(x; W^{(0)} + W' + W^\rho, V^{(0)} + V' + V^\rho\right)$$

We state Lemma 6.5 below.

**Lemma 6.5** (coupling). *Suppose $\tau_v \in \left[0, 1\right]$, $\tau_w \in \left(\frac{1}{m_1^{3/2}}, \frac{1}{m_1^{1/2}}\right]$, $\sigma_w \in \left(\frac{1}{m_1^{3/2}}, \frac{\tau_w}{m_1^{1/4}}\right]$, $\sigma_v \in \left(0, \frac{1}{m_2^{1/2}}\right]$ and $\eta > 0$. Given fixed unit vector $x$, and perturbation matrices $W', V', W'', V''$ (that may depend on the randomness of $W^{(0)}, b_1^{(0)}, V^{(0)}, b_2^{(0)}$ and $x$) satisfying*

$$\|W'\|_{2,4} \leq \tau_w, \|V'\|_F \leq \tau_v, \|W''\|_{2,4} \leq \tau_w, \|V''\|_F \leq \tau_v ,$$

*and random diagonal matrix $\Sigma$ with each diagonal entry i.i.d. drawn from $\{\pm 1\}$, then with high probability the following holds:*

1. *(Sparse sign change).* $\|D'_{w,x}\|_0 \leq \widetilde{O}(\tau_w^{4/5} m_1^{6/5})$, $\|D'_{v,x}\|_0 \leq \widetilde{O}\left(\sigma_v m_2^{3/2} + \tau_v^{2/3} m_2 + \tau_w^{2/3} m_1^{1/6} m_2\right).$

2. *(Cross term vanish).*

$$g_r(x; W^{(0)} + W^\rho + W' + \eta\Sigma W'', V^{(0)} + V^\rho + V' + \eta V''\Sigma)$$
$$= g_r\left(x; W^{(0)} + W^\rho + W', V^{(0)} + V^\rho + V'\right) + g_r^{(b,b)}(x; \eta\Sigma W'', \eta V''\Sigma) + g_r'(x)$$

*where $\mathbb{E}_\Sigma[g_r'(x)] = 0$ and with high probability $|g_r'(x)| \leq \eta\widetilde{O}\left(\frac{\sqrt{m_2}\tau_v}{\sqrt{m_1}} + \sqrt{m_2}\tau_w\right).$*

The first statement "sparse sign change" of Lemma 6.5 says that, if we move from random initialization $W^{(0)}, V^{(0)}$ to $W^{(0)} + W' + W^\rho, V^{(0)} + V' + V^\rho$, then how many signs of the ReLUs (in each layer) will change, as a function of the norms of $W'$ and $V'$. This calculation is similar to [4] but slightly more involved due to the $\|\cdot\|_{2,4}$ norm that we use here.

The second statement "cross term vanish" of Lemma 6.5 studies, if we are currently at weights $(W, V) = (W^{(0)} + W' + W^\rho, V^{(0)} + V' + V^\rho)$ and want to move to $(W + \eta\Sigma W'', V + \eta V''\Sigma)$ where $\Sigma$ is a diagonal matrix with diagonal entries i.i.d. uniformly chosen from $\{\pm 1\}$, then how does the function value change in the pseudo network.

### 6.3.1 Coupling + Existence

We quickly point out a corollary by applying the coupling and existential lemmas together.

Recall in the existential Lemma 6.2, we have studied a pseudo network $G^{(0)}$ where the signs are determined at the random initialization $W^{(0)}, V^{(0)}$. Now, the coupling Lemma 6.5 says that the amount of sign change from $G^{(0)}$ to $G^{(b,b)}$ can be controlled. Therefore, if parameters are chosen appropriately, the existential Lemma 6.2 should also apply to $G^{(b,b)}$. Formally,

---

In the three-layer network results, we choose parameters

$$\tau_v' = \frac{1}{\sqrt{\varepsilon_0}} \frac{m_1^{1/2-0.005}}{m_2^{1/2}}, \quad \tau_w' = \frac{C_0}{\varepsilon_0^{1/4}} \frac{1}{m_1^{3/4-0.005}}, \quad \lambda_v = \frac{2}{\tau_v^2}, \quad \lambda_w = \frac{2}{(\tau_w')^4},$$

$$\sigma_v = \frac{1}{m_2^{1/2+0.01}}, \quad \sigma_w = \frac{1}{m_1^{1-0.01}}, \quad \tau_v = \frac{m_1^{1/2-0.001}}{m_2^{1/2}} \gg \tau_v', \quad \tau_w = \frac{1}{m_1^{3/4-0.01}} \gg \tau_w'$$

$$m_2 = m_1, \quad \varepsilon_a = 1,$$

for $C_0 = \mathfrak{C}_\varepsilon(\Phi, \sqrt{p_2}\mathfrak{C}_\varepsilon(\phi, 1)) \cdot \mathfrak{C}_\varepsilon(\phi, 1) \cdot \widetilde{O}(p_1\sqrt{p_2}k)$ and $\varepsilon = \frac{\varepsilon_0}{kp_1p_2^2\mathfrak{C}_s(\Phi, p_2\mathfrak{C}_s(\phi, 1))\mathfrak{C}_s(\phi, 1)^2}$

---

Table 1: Three-layer parameter choices (the constant inside $\Theta$ can depend on $\gamma$).
$\lambda_w, \lambda_v$ give the weights of the regularizer $\lambda_w\|\cdot\|_{2,4}^4 + \lambda_v\|\cdot\|_{2,2}^2$ in the objective (6.1) and (4.2).
$\sigma_w, \sigma_v$ give the amount of Gaussian perturbation we add to the objective (for analysis purpose).
$\tau_w', \tau_v'$ are set so that if the regularizer is bounded, it satisfies $\|W'\|_{2,4} \leq \tau_w'$ and $\|V'\|_F \leq \tau_v'$.
$\tau_w, \tau_v$ are set so that the coupling lemma works whenever $\|W'\|_{2,4} \leq \tau_w$ and $\|V'\|_F \leq \tau_v$.

**Corollary 6.6** (existence after coupling). *In the same setting as Lemma 6.2, given perturbation matrices $W', V'$ (that may depend on the randomness of the initialization and the data distribution*

$\mathcal{D}$) with
$$\|W'\|_{2,4} \le \tau_w, \|V'\|_F \le \tau_v .$$
*Using parameter choices from Table 1, w.h.p. there exist $W^*$ and $V^*$ (independent of the randomness of $W^\rho, V^\rho$) satisfying*

$$\|W^*\|_{2,\infty} = \max_i \|w_i^*\|_2 \le \frac{C_0}{m_1}, \quad \|V^*\|_{2,\infty} = \max_i \|v_i^*\|_2 \le \frac{\sqrt{m_1}}{m_2}$$

$$\mathbb{E}_{(x,y)\sim\mathcal{D}} \left[ \sum_{r=1}^{k} \left| f_r^*(x) - g_r^{(b,b)}(x; W^*, V^*) \right| \right] \le \varepsilon,$$

$$\mathbb{E}_{(x,y)\sim\mathcal{D}} \left[ L(G^{(b,b)}(x; W^*, V^*), y) \right] \le \mathsf{OPT} + \varepsilon.$$

As we shall see later, Corollary 6.6 gives rise to $W^*$ and $V^*$ that shall be used as a descent direction for the objective. (Corollary 6.6 did not use all the parameters inside Table 1, and some of those parameters shall be used in later subsections.)

## 6.4 Optimization

The naïvely approach is to

- use the property that solutions with good risks are dense in the parameter space (i.e., Corollary 6.6) to show that the optimization landscape of the overparameterized three-layer neural network is benign: it has no spurious local minimal or (more or less) even any second-order critical points; and

- use existing theorems on escaping saddle points (such as [19]) to show that SGD will not be stuck in saddle points and thus converges.

**Key issue.** Unfortunately, before digging into the details, there is already a big hole in this approach. ReLU networks are not second-order-differentiable: a ReLU activation does not have a well-defined Hessian/sub-Hessian at zero. One may naïvely think that since a ReLU network is infinite-order differentiable everywhere except a measure zero set, so we can safely ignore the Hessian issue and proceed by pretending that the Hessian of ReLU is always zero. This intuition is *very wrong*. Following it, we could have run into the absurd conclusion that any piece-wise linear function is convex, since the Hessian of it is *zero* almost everywhere. In other words, the only non-smooth point of ReLU has a Hessian value equal to the Dirac $\delta$-function, but such non-smooth points, albeit being measure zero, are actually turning points in the landscape. If we want a meaningful second-order statement of the ReLU network, *we must not naïvely ignore the "Hessian" of ReLU at zeros.*

**Smoothing.** To fix the naïve approach, we use Gaussian smoothing. Given any bounded function $f : \mathbb{R}^{m_3} \to \mathbb{R}$, we have that $\mathbb{E}_{\rho\sim\mathcal{N}(0,\sigma^2 \mathbf{I})}[f(x + \rho)]$ is a infinite-order differentiable function in $x$ as long as $\sigma > 0$. Thus, we can consider the smoothed version of the neural network: $F(x; W + W^\rho, V + V^\rho)$ where $W^\rho, V^\rho$ are random Gaussian matrices. We show that $\mathbb{E}[L(F(x; W + W^\rho, V + V^\rho), y)]$ also has the desired property of essentially having no second-order critical points. Perhaps worth pointing out, the Hessian of this smoothed function is *significantly different* from the original one. For example, $\mathbb{E}_{\rho\sim\mathcal{N}(0,1)}[\sigma(x + \rho)]$ has a Hessian value $\approx 1$ at all $x = [-1, 1]$, while in the original ReLU function $\sigma$, the Hessian is 0 *almost everywhere*.

In practice, since the solution $W_t, V_t$ are found by *stochastic* gradient descent starting from *random initialization*, they will have a non-negligible amount of *intrinsic* noise. Thus, the additional smoothing in the algorithm might not be needed by an observation in [29]. Smooth analysis [44] might also be used for analyzing the effect of such noise, but this is beyond the scope of this paper.

**Actual algorithm.** Let us consider the following smoothed, and regularized objective:

$$L'(\lambda_t, W_t, V_t) = \mathbb{E}_{W^\rho, V^\rho, (x,y)\sim\mathcal{Z}} \left[ L\left(\lambda_t F\left(x; W^{(0)} + W^\rho + W_t, V^{(0)} + V^\rho + V_t\right), y\right) \right]$$
$$+ R(\sqrt{\lambda_t} W_t, \sqrt{\lambda_t} V_t)$$

where $W^\rho, V^\rho$ are Gaussian random matrices with each entry i.i.d. from $\mathcal{N}(0, \sigma_w^2)$ and $\mathcal{N}(0, \sigma_v^2)$, respectively. $R(\sqrt{\lambda_t} W_t, \sqrt{\lambda_t} V_t) = \lambda_v \|\sqrt{\lambda_t} V_t\|_F^2 + \lambda_w \|\sqrt{\lambda_t} W_t\|_{2,4}^4$ and $\lambda_v, \lambda_w$ are set such that $\lambda_v \|\sqrt{\lambda_t} V^*\|_F^2 \le \varepsilon_0$ and $\lambda_w \|\sqrt{\lambda_t} W^*\|_{2,4}^4 \le \varepsilon_0$ for every $W^*$ and $V^*$ coming from Lemma 6.2.

See Algorithm 2 (first SGD variant) for the details. We prove the following lemma.

**Lemma 6.7** (descent direction). *For every $\varepsilon_0 \in (0,1)$ and $\varepsilon = \frac{\varepsilon_0}{kp_1 p_2^2 \mathfrak{C}_\mathfrak{s}(\Phi, p_2 \mathfrak{C}_\mathfrak{s}(\phi,1)) \mathfrak{C}_\mathfrak{s}(\phi,1)^2}$, for every constant $\gamma \in (0, 1/4]$, consider the parameter choices in Table 1, and consider any $\lambda_t, W_t, V_t$ (that may depend on the randomness of $W^{(0)}, b^{(0)}, V^{(0)}, b^{(1)}$ and $\mathcal{Z}$) with*

$$\lambda_t \in \left((\varepsilon / \log(m_1 m_2))^{\Theta(1)}, 1\right] \quad and \quad L'(\lambda_t, W_t, V_t) \in [(1+\gamma)\mathsf{OPT} + \Omega(\varepsilon_0/\gamma), \widetilde{O}(1)]$$

*With high probability over the random initialization, there exists $W^*, V^*$ with $\|W^*\|_F, \|V^*\|_F \leq 1$ such that for every $\eta \in \left[0, \frac{1}{\mathsf{poly}(m_1, m_2)}\right]$:*

$$\min\left\{\mathbb{E}_{\mathbf{\Sigma}}\left[L'\left(\lambda_t, W_t + \sqrt{\eta}\mathbf{\Sigma}W^*, V_t + \sqrt{\eta}V^*\mathbf{\Sigma}\right)\right], L'\left((1-\eta)\lambda_t, W_t, V_t\right)\right\}$$
$$\leq (1 - \eta\gamma/4)(L'(\lambda_t, W_t, V_t)) \ ,$$

*where $\mathbf{\Sigma} \in \mathbb{R}^{m_1 \times m_1}$ is a diagonal matrix with each diagonal entry i.i.d. uniformly drawn from $\{\pm 1\}$.*

Lemma 6.7 says one of the following two scenarios will happen. Either (1) there exist $W^*, V^*$ so that updating in a random direction $(\mathbf{\Sigma}W^*, V^*\mathbf{\Sigma})$ decreases the objective, or (2) performing weight decay decreases the objective. It is a simple exercise to check that, if (1) happens, then $F'(\lambda, \cdot, \cdot)$ has a very negative curvature in the Hessian (see Fact A.8) at the current point. Therefore, Lemma 6.7 essentially says that after appropriate regularization, every second-order critical point of $L'$ is approximately global minimum.

More interestingly, since (noisy) stochastic gradient descent is capable of finding approximate second-order critical points with a global convergence rate, one can show the following final convergence rate for minimizing $L'$ for Algorithm 2.

**Lemma 6.8** (convergence). *In the setting of Theorem 3, with probability at least $99/100$, Algorithm 2 (the first SGD variant) converges in $TT_w = \mathsf{poly}(m_1, m_2)$ iterations to a point*
$$L'(\lambda_T, W_T, V_T) \leq (1+\gamma)\mathsf{OPT} + \varepsilon_0.$$

### 6.4.1 Details of Smoothing

The next lemma shows that when doing a small update to the current weight, one can view the sign pattern as fixed for the smoothed objective, up to a small error. Specifically, for every input $x$ and every $W = W^{(0)} + W', V = V^{(0)} + V'$, let $\mathbf{\Sigma}$ be a random diagonal matrix with $\pm$ entries, and let

- $D_{w,x,\rho}$ denote the diagonal matrix with diagonals being 0-1 signs of the first layer at $W + W^\rho$;
- $D_{w,x,\rho,\eta}$ denote of the first layer at weights $W + W^\rho + \eta\mathbf{\Sigma}W''$;
- $D_{v,x,\rho}$ denote that of the second layer at weights $W + W^\rho$ and $V + V^\rho$; and
- $D_{v,x,\rho,\eta}$ denote that of the second layer at weights $W + W^\rho + \eta\mathbf{\Sigma}W''$ and $V + V^\rho + \eta\mathbf{\Sigma}V''$.

For a fixed $r \in [k]$, consider the real network $P_{\rho,\eta}$ and the pseudo network $P'_{\rho,\eta}$:

$$P_{\rho,\eta} := f_r(x; W + W^\rho + \eta\mathbf{\Sigma}W'', V + V^\rho + \eta V''\mathbf{\Sigma})$$
$$= a_r D_{v,x,\rho,\eta}\Big((V + V^\rho + \eta V''\mathbf{\Sigma})D_{w,x,\rho,\eta}\left((W + W^\rho + \eta\mathbf{\Sigma}W'')x + b_1\right) + b_2\Big)$$
$$P'_{\rho,\eta} := g_r(x; W + W^\rho + \eta\mathbf{\Sigma}W'', V + V^\rho + \eta V''\mathbf{\Sigma})$$
$$= a_r D_{v,x,\rho}\Big((V + V^\rho + \eta V''\mathbf{\Sigma})D_{w,x,\rho}\left((W + W^\rho + \eta\mathbf{\Sigma}W'')x + b_1\right) + b_2\Big).$$

We prove the following lemma:

**Lemma 6.9** (smoothed real vs pseudo). *There exists $\eta_0 = \frac{1}{\mathsf{poly}(m_1, m_2)}$ such that, for every $\eta \leq \eta_0$, for every fixed $x$ with $\|x\|_2 = 1$, for every $W', V', W'', V''$ that may depend on the randomness of the initialization and*

$$\|W'\|_{2,4} \leq \tau_w, \quad \|V'\|_{2,2} \leq \tau_v, \quad \|W''\|_{2,\infty} \leq \tau_{w,\infty}, \quad \|V''\|_{2,\infty} \leq \tau_{v,\infty}$$

*we have with high probability:*

$$\mathbb{E}_{W^\rho, V^\rho}\left[\frac{|P_{\rho,\eta} - P'_{\rho,\eta}|}{\eta^2}\right] = \widetilde{O}\left(m_1 \frac{\tau_{w,\infty}^2}{\sigma_w} + \frac{m_2 \tau_{w,\infty}^2}{\sigma_v} + \frac{m_2}{m_1} \frac{\tau_{v,\infty}^2}{\sigma_v}\right) + O_p(\eta).$$

*where $O_p$ hides polynomial factor of $m_1, m_2$.*

In other words, when working with a Gaussian smoothed network (with output $P_{\rho,\eta}$), it suffices to study a pseudo network (with output $P'_{\rho,\eta}$). In the proof of Lemma 6.7, this allows us to go from the real (smoothed) network to the pseudo (smoothed) network, and then apply Lemma 6.5.

## 6.5 Generalization

In our first variant of SGD Algorithm 2, we show a very crude Rademacher complexity bound that can be derived from the contraction lemma (a.k.a. Talagrand's concentration inequality).

**Lemma 6.10** (generalization for $L_R = L_1$)**.** *For every $\tau'_v, \tau'_w \geq 0$, every $\sigma_v \in (0, 1/\sqrt{m_2}]$, w.h.p. for every $r \in [k]$ and every $N \geq 1$, the empirical Rademacher complexity is bounded by*

$$\frac{1}{N} \mathbb{E}_{\xi \in \{\pm 1\}^N} \left[ \sup_{\|V'\|_F \leq \tau'_v, \|W'\|_{2,4} \leq \tau'_w} \sum_{i \in [N]} \xi_i f_r(x_i; W^{(0)} + W^\rho + W', V^{(0)} + V^\rho + V') \right]$$

$$\leq \widetilde{O} \left( \frac{\tau'_w m_1 \sqrt{m_2} + \tau'_v m_2}{\sqrt{N}} + \frac{\tau'_v \sqrt{m_1 m_2 \tau'_w (1/\sqrt{m_1} + \tau'_w)}}{N^{1/4}} \right) \ .$$

Since the population risk is bounded by the Rademacher complexity, combining this with Lemma 6.8, one can easily prove Theorem 3.

## 6.6 Second Variant of SGD

In our second variant of SGD Algorithm 1, we have added the Dropout-type noise matrix $\boldsymbol{\Sigma}$ directly into the objective $L_2$ (see (4.2)). Our new stochastic objective is the following.

$$L''(\lambda_t, W_t, V_t) = \mathbb{E}_{W^\rho, V^\rho, \boldsymbol{\Sigma}, x, y \sim \mathcal{Z}} \left[ L \left( \lambda_t F \left( W^{(0)} + W^\rho + \boldsymbol{\Sigma} W_t, V^{(0)} + V^\rho + V_t \boldsymbol{\Sigma}, x \right), y \right) \right]$$
$$+ R(\sqrt{\lambda_t} W_t, \sqrt{\lambda_t} V_t).$$

To show that this gives a better sampling complexity bound, we need the following stronger coupling lemma. It gives a somewhat better bound on the "cross term vanish" part comparing to the old coupling Lemma 6.5.

**Lemma 6.11** (stronger coupling)**.** *With high probability over the random initialization and over a random diagonal matrix $\boldsymbol{\Sigma}$ with diagonal entries i.i.d. generated from $\{-1, 1\}$, it satisfies that for every $W', V'$ with $\|V'\|_2 \leq \tau_v, \|W'\|_{2,4} \leq \tau_w$ for $\tau_v \in [0,1]$ and $\tau_w \in \left[ \frac{1}{m_1^{3/4}}, \frac{1}{m_1^{9/16}} \right]$, we have*

$$f_r(x; W^{(0)} + \boldsymbol{\Sigma} W', V^{(0)} + V' \boldsymbol{\Sigma}) = a_r D_{v,x}(V^{(0)} D_{w,x}(W^{(0)} x + b_1) + b_2) + a_r D_{v,x} V' D_{w,x} W' x$$

$$\pm \widetilde{O} \left( \tau_w^{8/5} m_1^{9/10} + \tau_w^{16/5} m_1^{9/5} \sqrt{m_2} + \frac{\sqrt{m_2}}{\sqrt{m_1}} \tau_v \right) \ .$$

*Under parameter choices Table 1, the last error term is at most $\varepsilon/k$.*

For this SGD variant, we also have the following the stronger Rademacher complexity bound. It relies on Lemma 6.11 to reduce the function class to pseudo networks, which are only *linear* functions in $W'$ and $V'$, and then computes its Rademacher complexity.

**Lemma 6.12** (generalization for $L_R = L_2$)**.** *For every $\tau'_v \in [0,1]$, $\tau'_w \in \left[ \frac{1}{m_1^{3/4}}, \frac{1}{m_1^{9/16}} \right]$, every $\sigma_w \in [0, 1/\sqrt{m_1}]$ and $\sigma_v \in [0, 1/\sqrt{m_2}]$, w.h.p. for every $r \in [k]$ and every $N \geq 1$, we have by our choice of parameters in Lemma 6.7, the empirical Rademacher complexity is bounded by*

$$\frac{1}{N} \mathbb{E}_{\xi \in \{\pm 1\}^N} \left[ \sup_{\|V'\|_F \leq \tau'_v, \|W'\|_{2,4} \leq \tau'_w} \left| \sum_{i \in [N]} \xi_i \mathbb{E}_{\boldsymbol{\Sigma}}[f_r(x_i; W^{(0)} + W^\rho + \boldsymbol{\Sigma} W', V^{(0)} + V^\rho + V' \boldsymbol{\Sigma})] \right| \right]$$

$$\leq \widetilde{O} \left( \frac{\tau'_w \tau'_v m_1^{1/4} \sqrt{m_2}}{\sqrt{N}} + \left( (\tau'_w)^{8/5} m_1^{9/10} + (\tau'_w)^{16/5} m_1^{9/5} \sqrt{m_2} + \frac{\sqrt{m_2}}{\sqrt{m_1}} \tau'_v \right) \right).$$

*Under parameter choices in Table 1, this is at most $\widetilde{O} \left( \frac{\tau'_w \tau'_v m_1^{1/4} \sqrt{m_2}}{\sqrt{N}} \right) + \varepsilon/k$.*

After plugging Lemma 6.11 and Lemma 6.12 into the final proof, we can show Theorem 2.

*Remark.* In contrast, without the Dropout-type noise matrix $\Sigma$, the error term in the old coupling Lemma 6.5 is too large so we cannot get better Rademacher complexity bounds.

# 7 Empirical Evaluations

(a) $N = 1000$ and vary $m$                (b) $m = 2000$ and vary $N$

Figure 2: Performance comparison. `3layer`/`2layer` stands for training (hidden weights) in three and two-layer neural networks. (`last`) stands for *conjugate kernel* [15], meaning training only the output layer. (`NTK`) stands for *neural tangent kernel* [27] with finite width.

---

**Setup.** We consider $\ell_2$ regression task on synthetic data where feature vectors $x \in \mathbb{R}^4$ are generated as normalized random Gaussian, and label is generated by target function $y = F^*(x) = (\tanh(8x_1) + \tanh(8x_2) + \tanh(8x_3) - 2)^2 \cdot \tanh(8x_4)$.

---

We discuss our experiment details for Figure 1.

Recall that we generate synthetic data where the feature vectors $x \in \mathbb{R}^4$ are generated as random Gaussian then normalized to norm 1, and labels are generated by target function $y = F^*(x) = (\sin(3x_1) + \sin(3x_2) + \sin(3x_3) - 2)^2 \cdot \cos(7x_4)$. Intuitively, the constants 3 and 7 control the complexity of the activation functions in the target function, and we choose these values to ensure that the two factors in the target function have roughly the same complexity.[14]

To be more consistent with our theorems, we implement fully connected neural networks and train only hidden weights (namely, $W$ in the two-layer case and $W, V$ in the two-layer case). We also implement NTK with respect to only hidden weights. For conjugate kernel, we only train the last (output) layer, that is, the weights $a_r \in \mathbb{R}^m$ for $r \in [k]$ in the language of this paper.

To be consistent with our theorems, we choose random initialization as follows. Entries of $a_r$ are i.i.d. from $\mathcal{N}(0, 1)$, and entries of $W, V, b_1, b_2$ are i.i.d. from $\mathcal{N}(0, \frac{1}{m})$. This ensures that the output at random initialization is $\Theta(1)$.

We use the default SGD optimizer of pytorch, with momentum 0.9, mini-batch size 50, learning rate `lr` and weight decay parameter `wd`. We carefully run each algorithm with respect to `lr` and `wd` in the set $\{10^{-k}, 2 \cdot 10^{-k}, 5 \cdot 10^{-k} : k \in \mathbb{Z}\}$, and presents the best one in terms of testing accuracy. In each parameter setting, we run SGD for 800 epochs, and decrease `lr` by 10 on epoch 400.

In Figure 2, we provide an additional experiment for target function $y = F^*(x) = (\tanh(8x_1) + \tanh(8x_2) + \tanh(8x_3) - 2)^2 \cdot \tanh(8x_4)$.

## 7.1 Justification of Our $\|W\|_{2,4}$ Regularizer

Recall our three-layer theorem requires a slightly unconventional regularizer, namely, the $\|W\|_{2,4}$ norm of the first layer to encourage weights to be more evenly distributed across neurons. Is this really necessary? To get some preliminary idea we run our aforementioned three-layer experiment on the first data set,

- once with the traditional weight decay on $W$ (which corresponds to minimizing $\|W\|_F$), and

Figure 3: Empirical comparison between regularizers $\|W\|_F$ and $\|W\|_{2,4}$.

- once with the $\|W\|_{2,4}$ norm without weight decay.

In both cases we best tune the learning rates as well as the weight decay parameter (or the weight of the $\|W\|_{2,4}^4$ regularizer). We present our findings in Figure 3.

In Figure 3(a), we observe that there is no real difference between the two regularizers in terms of *test error*. In one case ($m = 200$) using $\|W\|_{2,4}$ even gives slightly better test accuracy (but we do not want to claim this as a general phenomenon).

More importantly, since by Cauchy-Schwarz we have $\|W\|_F \geq \|W\|_{2,4} \geq m^{-1/4}\|W\|_F$. Let us consider the following norm ratio:

$$ratio := \frac{\|W\|_{2,4}^4 \cdot m}{\|W\|_F^4} \in [1, m]$$

If the $ratio$ is close to 1, then one can argue that the more *evenly distributed* the rows (i.e., neurons) of $W \in \mathbb{R}^{m \times d}$ are. In contrast, if $ratio$ is close to $m$ then the weights in $W$ are concentrated to a single row. (Of course, we cannot expect $ratio$ to be really close to 1 since there may not exist such solutions with good accuracy to begin with.)

In Figure 3(b), we see that $ratio$ is roughly the same between two types of regularizers. This means, even if we regularize only the Frobenius norm, weights are still sufficiently distributed across neurons comparing to what we can do by regularizing $\|W\|_{2,4}$. We leave it a future work to study why SGD encourages such implicit regularization.

# APPENDIX: COMPLETE PROOFS

We provide technical preliminaries in Appendix A, give our two-layer proofs in Appendix B, and three-layer proofs in Appendix C.

## A Technical Preliminaries

**Wasserstein distance.** The $\ell_2$ Wasserstein distance between random variables $A, B$ is

$$\mathcal{W}_2(A, B) := \sqrt{\inf_{(X,Y) \text{ s.t. } X \sim A, Y \sim B} \mathbb{E}\big[|X - Y|^2\big]}$$

where the infimum is taken over all possible joint distributions over $(X, Y)$ where the marginal on $X$ (resp. $Y$) is distributed in the same way as $A$ (resp. $B$).

Slightly abusing notation, in this paper, we say a random variable $X$ satisfies $|X| \leq B$ with high probability if (1) $|X| \leq B$ w.h.p. and (2) $\mathcal{W}_2(X, 0) \leq B$. For instance, if $g = \mathcal{N}(0, 1/m)$, then $|g| \leq \widetilde{O}(1/\sqrt{m})$ with high probability.

### A.1 Probability

**Lemma A.1** (Gaussian indicator concentration). *Let* $(n_1, \alpha_1, a_{1,1}, a_{2,1}), \cdots, (n_m, \alpha_m, a_{1,m}, a_{2,m})$ *be $m$ i.i.d. samples from some distribution, where within a 4-tuples:*

- *the marginal distribution of $a_{1,i}$ and $a_{2,i}$ is standard Gaussian $\mathcal{N}(0, 1)$;*
- *$n_i$ and $\alpha_i$ are not necessarily independent;*
- *$a_{1,i}$ and $a_{2,i}$ are independent; and*
- *$n_i$ and $\alpha_i$ are independent of $a_{1,i}$ and $a_{2,i}$.*

*Suppose $h \colon \mathbb{R} \to [-L, L]$ is a fixed function. Then, for every $B \geq 1$:*

$$\mathbf{Pr}\left[\left|\left(\sum_{i \in [m]} a_{1,i} a_{2,i} \mathbb{I}[n_i \geq 0] h(\alpha_i)\right)\right| \geq BL(\sqrt{m} + B)\right] \leq 4e^{-B^2/8}$$

*and*

$$\mathbf{Pr}\left[\left|\left(\sum_{i \in [m]} a_{1,i}^2 \mathbb{I}[n_i \geq 0] h(\alpha_i)\right) - m\mathbb{E}[a_{1,1}^2 \mathbb{I}[n_1 \geq 0] h(\alpha_1)]\right| \geq BL(\sqrt{m} + B)\right] \leq 4e^{-B^2/8}.$$

*Proof of Lemma A.1.* Let us consider a fixed $n_1, \alpha_1, \cdots, n_m, \alpha_m$, then since each $|\mathbb{I}[n_i \geq 0] h(\alpha_i)| \leq L$, by Gaussian chaos variables concentration bound (e.g., Example 2.15 in [37]) we have that

$$\mathbf{Pr}\left[\left|\left(\sum_{i \in [m]} a_{1,i} a_{2,i} \mathbb{I}[n_i \geq 0] h(\alpha_i)\right)\right| \geq BL(\sqrt{m} + B)\,\middle|\, \{n_i, \alpha_i\}_{i \in [m]}\right] \leq 4e^{-B^2/8}.$$

Since this holds for every choice of $\{n_i, \alpha_i\}_{i \in [m]}$ we can complete the proof. The second inequality follows from sub-exponential concentration bounds. □

**Proposition A.2.** *If $X_1, X_2$ are independent, and $X_1, X_3$ are independent conditional on $X_2$, then $X_1$ and $X_3$ are independent.*

*Proof.* For every $x_1, x_2, x_3$:

$$\mathbf{Pr}[X_1 = x_1, X_3 = x_3 \mid X_2 = x_2] = \mathbf{Pr}[X_1 = x_1 \mid X_2 = x_2]\,\mathbf{Pr}[X_3 = x_3 \mid X_2 = x_2]$$
$$= \mathbf{Pr}[X_1 = x_1]\,\mathbf{Pr}[X_3 = x_3 \mid X_2 = x_2].$$

Multiplying $\mathbf{Pr}[X_2 = x_2]$ on both side leads to:

$$\mathbf{Pr}[X_1 = x_1, X_3 = x_3, X_2 = x_2] = \mathbf{Pr}[X_1 = x_1]\,\mathbf{Pr}[X_3 = x_3, X_2 = x_2].$$

Marginalizing away $X_2$ gives $\mathbf{Pr}[X_1 = x_1, X_3 = x_3] = \mathbf{Pr}[X_1 = x_1]\,\mathbf{Pr}[X_3 = x_3]$, so $X_1$ and $X_3$ are independent. □

## A.2 Central Limit Theorem

The following Wasserstein distance bound of central limit theorem is easy to derive from known results:

**Lemma A.3** (new CLT). *Let $X_1, \cdots, X_m \in \mathbb{R}$ be $m$ independent zero-mean random variables with each $|X_i| \leq C$, $\sum_{i \in [m]} \mathbb{E}[X_i^2] = V$, then there exists $Z \sim \mathcal{N}(0, V)$ such that*

$$\mathcal{W}_2\left(\sum_{i=1}^m X_i, Z\right) = O\left(C \log m\right) \ .$$

*Proof.* Define $\Sigma_i = \mathbb{E}[X_i^2]$ and without loss of generality assume $\Sigma_1 \geq \Sigma_2 \geq \cdots \geq \Sigma_m$. Let us apply [52, Lemma 1.6] on each $i$, then for every $t$ such that $t \geq \frac{5C^2}{\Sigma_i}$, let $Y_t \sim \mathcal{N}(0, t\Sigma_i)$ be independent of $X_i$, we have

$$\mathcal{W}_2(Y_t, Y_{t-1} + X_i) \leq \frac{5C}{t} \ .$$

In other words —by choosing $t = R_i/\Sigma_i$— we have for every $R_i \geq 5C^2$, letting $Z_i \sim \mathcal{N}(0, R_i)$, $Z_{i+1} \sim \mathcal{N}(0, R_i + \Sigma_i)$ be independent gaussian (also independent of $X_i$), it satisfies

$$\mathcal{W}_2(Z_{i+1}, Z_i + X_i) \leq \frac{5C\Sigma_i}{R_i} \ .$$

Repeatedly applying the above inequality and starting with $Z_1 \sim \mathcal{N}(0, 5C^2)$ and choosing $R_i = 5C^2 + \sum_{j=1}^{i-1} \Sigma_j$, we have $Z_i \sim \mathcal{N}(0, 5C^2 + \sum_{j=1}^{i-1} \Sigma_j)$ and $Z_{m+1} \sim \mathcal{N}(0, 5C^2 + V)$. Using $\Sigma_1 \geq \Sigma_2 \geq \cdots \geq \Sigma_m$, we know that $\frac{5C\Sigma_i}{R_i} \leq \frac{5C}{i-1}$ for $i \geq 2$. This implies that

$$\mathcal{W}_2\left(Z_{m+1}, \sum_{i=1}^m X_i\right) \leq \sum_{i=2}^m \frac{5C}{i-1} + \mathcal{W}_2\left(Z_2, Z_1 + X_1\right) = O\left(C \log m\right) + \mathcal{W}_2\left(Z_2, Z_1 + X_1\right)$$

Finally, since $|X_1| \leq C$ we have $\mathcal{W}_2(X_1, 0) \leq C$ and $\Sigma_1^2 \leq C^2$. By triangle inequality we have

$$\mathcal{W}_2\left(Z_2, Z_1 + X_1\right) \leq \mathcal{W}_2(X_1, 0) + \mathcal{W}_2(Z_1, 0) + \mathcal{W}_2(Z_2, 0) = O(C)$$

There exists $Z \sim \mathcal{N}(0, V)$ such that $\mathcal{W}_2(Z_{m+1}, Z) = O(C)$. All of these together imply

$$\mathcal{W}_2\left(Z, \sum_{i=1}^m X_i\right) = O\left(C \log m\right) \ . \qquad \square$$

## A.3 Interval Partition

**Lemma A.4** (Interval Partition). *For every $\tau \leq \frac{1}{100}$, there exists a function $\mathfrak{s}: [-1, 1] \times \mathbb{R} \to \{-1, 0, 1\}$ and a set $I(y) \subset [-2, 2]$ for every $y \in [-1, 1]$ such that, for every $y \in [-1, 1]$,*

1. *(Indicator).* $\mathfrak{s}(y, g) = 0$ *if* $g \notin I(y)$*, and* $\mathfrak{s}(y, g) \in \{-1, 1\}$ *otherwise.*

2. *(Balanced).* $\mathbf{Pr}_{g \sim \mathcal{N}(0,1)}[g \in I(y)] = \tau$ *for every* $y \in [-1, 1]$*.*

3. *(Symmetric).* $\mathbf{Pr}_{g \sim \mathcal{N}(0,1)}[\mathfrak{s}(y, g) = 1] = \mathbf{Pr}_{g \sim \mathcal{N}(0,1)}[\mathfrak{s}(y, g) = -1]$*.*

4. *(Unbiased).* $\mathbb{E}_{g \sim \mathcal{N}(0,1)}[\mathfrak{s}(y, g)g \mid g \in I(y)] = y$*.*

5. *(Bounded).* $\max_{x \in I(y)}\{\mathfrak{s}(y, x)x\} - \min_{x \in I(y)}\{\mathfrak{s}(y, x)x\} \leq 10\tau$*.*

6. *(Lipschitz).* $|I(y_1) \triangle I(y_2)| \leq O(|y_2 - y_1|)$ *, where* $|I| := \int_{x \in I} dx$ *is the measure of set* $I \subseteq \mathbb{R}$*.*

*We refer to $I(y)$ as an "interval" although it may actually consist of two disjoint closed intervals.*

*Proof of Lemma A.4.* Let us just prove the case when $y \geq 0$ and the other case is by symmetry. It is clear that, since there are only two degrees of freedom, there is a unique interval $I_1(y) = [y - a(y), y + b(y)]$ with $a(y), b(y) \geq 0$ such that

1. *(Half probability).* $\mathbf{Pr}_{g \sim \mathcal{N}(0,1)}[g \in I_1(y)] = \frac{\tau}{2}$*.*

2. *(Unbiased).* $\mathbb{E}_{g \sim \mathcal{N}(0,1)}[g \mid g \in I_1(y)] = y$*.*

Next, consider two cases:

1. Suppose $[y - a(y), y + b(y)]$ and $[-y - b(y), -y + a(y)]$ are disjoint. In this case, we just define $I(y) := [y - a(y), y + b(y)] \cup [-y - b(y), -y + a(y)]$ and define

$$
\mathfrak{s}(y, g) := \begin{cases} 1 & \text{if } g \in [y - a(y), y + b(y)]; \\ -1 & \text{if } g \in [-y - b(y), -y + a(y)]; \\ 0 & \text{otherwise.} \end{cases}
$$

2. $[y - a(y), y + b(y)]$ and $[-y - b(y), -y + a(y)]$ intersect. In this case, consider the unique interval

$$
I_2(y) = [-e(y), e(y)]
$$

where $e(y) \geq 0$ is defined so that

$$
\mathbb{E}_{g \sim \mathcal{N}(0,1)}[g \mid g \in I_2(y) \wedge g > 0] = \mathbb{E}_{g \sim \mathcal{N}(0,1)}[|g| \mid g \in I_2(y)] = y \ .
$$

It must satisfy $\mathbf{Pr}_{g \sim \mathcal{N}(0,1)}[g \in I_2(y) \wedge g > 0] < \tau/2$, because otherwise we must have $y - a(y) \geq 0$ and the two intervals should not have intersected.

Define $\tau'(y) = \mathbf{Pr}_{g \sim \mathcal{N}(0,1)}[g \in I_2(y)] < \tau$. Let $c(y) > e(y)$ be the unique positive real such that

$$
\mathop{\mathbf{Pr}}_{g \sim \mathcal{N}(0,1)} [g \in [e(y), c(y)]] = \frac{\tau - \tau'(y)}{2} \ .
$$

Let $d(y) \in [e(y), c(y)]$ be the unique real such that

$$
\mathop{\mathbf{Pr}}_{g \sim \mathcal{N}(0,1)} [g \in [e(y), d(y)]] = \mathop{\mathbf{Pr}}_{g \sim \mathcal{N}(0,1)} [g \in [d(y), c(y)]] \ .
$$

Finally, we define $I(y) = [-c(y), c(y)]$ and

$$
\mathfrak{s}(y, g) := \begin{cases} 1 & \text{if } g \in [0, e(y)] \cup [e(y), d(y)] \cup [-d(y), -e(y)]; \\ -1 & \text{if } g \in [-e(y), 0] \cup [d(y), c(y)] \cup [-c(y), -d(y)]; \\ 0 & \text{otherwise.} \end{cases}
$$

In both cases, one can carefully verify that properties 1, 2, 3, 4 hold. Property 5 follows from the standard property of Gaussian random variable under condition $\tau \leq 1/100$ and $y \in [-1, 1]$.

To check the final Lipschitz continuity property, recall for a standard Gaussian distribution, inside interval $[-\frac{1}{10}, \frac{1}{10}]$ it behaves, up to multiplicative constant factor, similar to a uniform distribution. Therefore, the above defined functions $a(y)$ and $b(y)$ are $O(1)$-Lipschitz continuous in $y$. Let $y_0 \geq 0$ be the unique constant such that $y - a(y) = 0$ (it is unique because $y - a(y)$ monotonically decreases as $y \to 0+$. It is clear that for $y_0 \leq y_1 \leq y_2$ it satisfies

$$
|I(y_1) \triangle I(y_2)| \leq O(y_2 - y_1) \ .
$$

As for the turning point of $y = y_0$, it is clear that

$$
\lim_{y \to y_0+} I(y) = [-y_0 - b(y_0), y_0 + b(y_0)] = [-e(y_0), e(y_0)] = \lim_{y \to y_0-} I(y)
$$

so the function $I(\cdot)$ is continuous at point $y = y_0$. Finally, consider $y \in [-y_0, y_0]$. One can verify that $e(y)$ is $O(1)$-Lipschitz continuous in $y$, and therefore the above defined $\tau'(y)$, $d(y)$ and $c(y)$ are also $O(1)$-Lipschitz in $y$. This means, for $-y_0 \leq y_1 \leq y_2 \leq y_0$, it also satisfies

$$
|I(y_1) \triangle I(y_2)| \leq O(y_2 - y_1) \ .
$$

This proves the Lipschitz continuity of $I(y)$. $\qquad\square$

## A.4 Hermite polynomials

**Definition A.5.** *Let $h_i (i \geq 0)$ denote the degree-$i$ (probabilists') Hermite polynomial*

$$
h_i(x) := i! \sum_{m=0}^{\lfloor i/2 \rfloor} \frac{(-1)^m}{m!(i - 2m)!} \frac{x^{i-2m}}{2^m}
$$

*satisfying the orthogonality constraint*

$$
\mathbb{E}_{x \sim \mathcal{N}(0,1)}[h_i(x) h_j(x)] = \sqrt{2\pi} j! \delta_{i,j}
$$

where $\delta_{i,j} = 1$ if $i = j$ and $\delta_{i,j} = 0$ otherwise. They have the following summation and multiplication formulas.

$$h_i(x + y) = \sum_{k=0}^{i} \binom{i}{k} x^{i-k} h_k(y),$$

$$h_i(\gamma x) = \sum_{k=0}^{\lfloor \frac{i}{2} \rfloor} \gamma^{i-2k} (\gamma^2 - 1)^k \binom{i}{2k} \frac{(2k)!}{k!} 2^{-k} h_{i-2k}(x).$$

**Lemma A.6.**

*(a) For even $i > 0$, for any $x_1 \in [0, 1]$ and $b$,*

$$\mathbb{E}_{\alpha, \beta \sim \mathcal{N}(0,1)} \left[ h_i \left( \alpha x_1 + \beta \sqrt{1 - x_1^2} \right) \mathbb{I}[\alpha \geq b] \right] = p_i x_1^i, \text{ where}$$

$$p_i = (i - 1)!! \frac{\exp(-b^2/2)}{\sqrt{2\pi}} \sum_{r=1, r \text{ odd}}^{i-1} \frac{(-1)^{\frac{i-1-r}{2}}}{r!!} \binom{i/2 - 1}{(r-1)/2} b^r.$$

*(b) For odd $i > 0$, for any $x_1 \in [0, 1]$ and $b$,*

$$\mathbb{E}_{\alpha, \beta \sim \mathcal{N}(0,1)} \left[ h_i \left( \alpha x_1 + \beta \sqrt{1 - x_1^2} \right) \mathbb{I}[\alpha \geq b] \right] = p_i x_1^i, \text{ where}$$

$$p_i = (i - 1)!! \frac{\exp(-b^2/2)}{\sqrt{2\pi}} \sum_{r=0, r \text{ even}}^{i-1} \frac{(-1)^{\frac{i-1-r}{2}}}{r!!} \binom{i/2 - 1}{(r-1)/2} b^r.$$

*(Throughout this paper, the binomial $\binom{n}{m}$ is defined as $\frac{\Gamma(n+1)}{\Gamma(m+1)\Gamma(n+1-m)}$, and this allows us to write for instance $\binom{5/2}{-1/2}$ without notation change.)*

*Proof.* Using the summation formula of Hermite polynomial, we have:

$$h_i \left( \alpha x_1 + \beta \sqrt{1 - x_1^2} \right) = \sum_{k=0}^{i} \binom{i}{k} (\alpha x_1)^{i-k} h_k \left( \beta \sqrt{1 - x_1^2} \right).$$

Using the multiplication formula of Hermite polynomial, we have:

$$h_k \left( \beta \sqrt{1 - x_1^2} \right) = \sum_{j=0}^{\lfloor \frac{k}{2} \rfloor} \left( \sqrt{1 - x_1^2} \right)^{k-2j} (-x_1^2)^j \binom{k}{2j} \frac{(2j)!}{j!} 2^{-j} h_{k-2j}(\beta).$$

For even $k$, since $\mathbb{E}_{\beta \sim \mathcal{N}(0,1)}[h_n(\beta)] = 0$ for $n > 0$, we have

$$\mathbb{E}_{\beta \sim \mathcal{N}(0,1)} \left[ h_k \left( \beta \sqrt{1 - x_1^2} \right) \right] = (-x_1^2)^{k/2} \frac{k!}{(k/2)!} 2^{-k/2},$$

and for odd $k$,

$$\mathbb{E}_{\beta \sim \mathcal{N}(0,1)} \left[ h_k \left( \beta \sqrt{1 - x_1^2} \right) \right] = 0.$$

This implies

$$\mathbb{E}_{\beta \sim \mathcal{N}(0,1)} \left[ h_i \left( \alpha x_1 + \beta \sqrt{1 - x_1^2} \right) \right] = \sum_{k=0, k \text{ even}}^{i} \binom{i}{k} (\alpha x_1)^{i-k} (-x_1^2)^{k/2} \frac{k!}{(k/2)!} 2^{-k/2}$$

$$= x_1^i \sum_{k=0, k \text{ even}}^{i} \binom{i}{k} \alpha^{i-k} \frac{k!}{(k/2)!} (-2)^{-k/2}.$$

Therefore,

$$\mathbb{E}_{\alpha, \beta \sim \mathcal{N}(0,1)} \left[ h_i \left( \alpha x_1 + \beta \sqrt{1 - x_1^2} \right) \mathbb{I}[\alpha \geq b] \right]$$

$$= x_1^i \sum_{k=0,k \text{ even}}^{i} \binom{i}{k} \mathbb{E}_{\alpha \sim \mathcal{N}(0,1)}[\alpha^{i-k} \mathbb{I}[\alpha \geq b]] \frac{k!}{(k/2)!}(-2)^{-k/2} \quad . \tag{A.1}$$

Define

$$L_{i,b} := \mathbb{E}_{\alpha \sim \mathcal{N}(0,1)}[\alpha^i \mathbb{I}[\alpha \geq b]].$$

(a) Consider even $i > 0$. By Lemma A.7, we have for even $i \geq 0$:

$$L_{i,b} = (i-1)!! \Phi(0,1;b) + \phi(0,1;b) \sum_{j=1,j \text{ odd}}^{i-1} \frac{(i-1)!!}{j!!} b^j.$$

So

$$\mathbb{E}_{\alpha,\beta \sim \mathcal{N}(0,1)} \left[ h_i \left( \alpha x_1 + \beta \sqrt{1 - x_1^2} \right) \mathbb{I}[\alpha \geq b] \right]$$

$$= x_1^i \left( \sum_{k=0,k \text{ even}}^{i} \binom{i}{k} L_{i-k,b} \frac{k!}{(k/2)!}(-2)^{-k/2} \right)$$

$$= x_1^i \left( \sum_{k=0,k \text{ even}}^{i} \binom{i}{k} (i-k-1)!! \Phi(0,1;b) \frac{k!}{(k/2)!}(-2)^{-k/2} \right)$$

$$+ x_1^i \phi(0,1;b) \left( \sum_{k=0,k \text{ even}}^{i} \binom{i}{k} \left( \sum_{j=1,j \text{ odd}}^{i-k-1} \frac{(i-k-1)!!}{j!!} b^j \right) \frac{k!}{(k/2)!}(-2)^{-k/2} \right).$$

Since

$$\sum_{k=0,k \text{ even}}^{i} \binom{i}{k} (i-k-1)!! \frac{k!}{(k/2)!}(-2)^{-k/2} = \sum_{k=0,k \text{ even}}^{i} \frac{i!(i-k-1)!!}{(i-k)!(k/2)!} \frac{(-1)^{k/2}}{2^{k/2}}$$

$$= \sum_{k=0,k \text{ even}}^{i} \frac{i!}{(i-k)!!(k/2)!} \frac{(-1)^{k/2}}{2^{k/2}}$$

$$= (i-1)!! \sum_{k=0,k \text{ even}}^{i} \frac{i!!}{(i-k)!!(k/2)!} \frac{(-1)^{k/2}}{2^{k/2}}$$

$$= (i-1)!! \sum_{k=0,k \text{ even}}^{i} \binom{i/2}{k/2}(-1)^{k/2}$$

$$= 0,$$

we know that

$$\mathbb{E}_{\alpha,\beta \sim \mathcal{N}(0,1)} \left[ h_i \left( \alpha x_1 + \beta \sqrt{1 - x_1^2} \right) \mathbb{I}[\alpha \geq b] \right] = x_1^i (i-1)!! \phi(0,1;b) \sum_{r=1,r \text{ odd}}^{i-1} c_r b^r$$

where $c_r$ is given by:

$$c_r := \frac{1}{(i-1)!!} \sum_{k=0,k \text{ even}}^{i-1-r} \binom{i}{k} \frac{(i-k-1)!!}{r!!} \frac{k!}{(k/2)!}(-2)^{-k/2}$$

$$= \sum_{k=0,k \text{ even}}^{i-1-r} \binom{i/2}{k/2} \frac{(-1)^{k/2}}{r!!}$$

$$= \sum_{j=0}^{(i-1-r)/2} \binom{i/2}{j} \frac{(-1)^j}{r!!}$$

$$= \sum_{j=0}^{(i-1-r)/2} \binom{j-i/2-1}{j} \frac{1}{r!!}$$

$$
= \frac{1}{r!!} \binom{-i/2 + (i-1-r)/2}{(i-1-r)/2}
$$

$$
= \frac{(-1)^{(i-1-r)/2}}{r!!} \binom{i/2-1}{(i-1-r)/2}
$$

$$
= \frac{(-1)^{\frac{i-1-r}{2}}}{r!!} \binom{i/2-1}{(r-1)/2}.
$$

(b) Consider odd $i > 0$. By Lemma A.7, we have for odd $i > 0$:

$$
L_{i,b} = \phi(0,1;b) \sum_{j=0, j \text{ even}}^{i-1} \frac{(i-1)!!}{j!!} b^j.
$$

So

$$
\mathbb{E}_{\alpha, \beta \sim \mathcal{N}(0,1)} \left[ h_i \left( \alpha x_1 + \beta \sqrt{1-x_1^2} \right) \mathbb{I}[\alpha \geq b] \right]
$$

$$
= x_1^i \left( \sum_{k=0, k \text{ even}}^{i} \binom{i}{k} L_{i-k,b} \frac{k!}{(k/2)!} (-2)^{-k/2} \right)
$$

$$
= x_1^i \phi(0,1;b) \left( \sum_{k=0, k \text{ even}}^{i} \binom{i}{k} \left( \sum_{j=0, j \text{ even}}^{i-k-1} \frac{(i-k-1)!!}{j!!} b^j \right) \frac{k!}{(k/2)!} (-2)^{-k/2} \right)
$$

$$
= x_1^i (i-1)!! \phi(0,1;b) \sum_{r=0, r \text{ even}}^{i-1} c_r b^r
$$

where $c_r$ is given by:

$$
c_r := \frac{1}{(i-1)!!} \sum_{k=0, k \text{ even}}^{i-1-r} \binom{i}{k} \frac{(i-k-1)!!}{r!!} \frac{k!}{(k/2)!} (-2)^{-k/2}
$$

$$
= \frac{(-1)^{\frac{i-1-r}{2}}}{r!!} \binom{i/2-1}{(r-1)/2},
$$

by a similar calculation as in the even $i$ case.

The proof is completed. $\qquad\square$

**Lemma A.7.** *Define $L_{i,b}$ as:*

$$
L_{i,b} := \mathbb{E}_{\alpha \sim \mathcal{N}(0,1)}[\alpha^i \mathbb{I}[\alpha \geq b]].
$$

*Then $L_{i,b}$'s are given by the recursive formula:*

$$
L_{0,b} = \Phi(0,1;b) := \Pr_{\alpha \sim \mathcal{N}(0,1)}[\alpha \geq b],
$$

$$
L_{1,b} = \phi(0,1;b) := \mathbb{E}_{\alpha \sim \mathcal{N}(0,1)}[\alpha \mathbb{I}[\alpha \geq b]] = \frac{\exp(-b^2/2)}{\sqrt{2\pi}},
$$

$$
L_{i,b} = b^{i-1} \phi(0,1;b) + (i-1) L_{i-2,b}.
$$

*As a result (with the convention that $0!! = 1$ and $(-1)!! = 1$)*

*for even $i \geq 0$:* $\qquad$ $L_{i,b} = (i-1)!! \Phi(0,1;b) + \phi(0,1;b) \sum_{j=1, j \text{ odd}}^{i-1} \frac{(i-1)!!}{j!!} b^j$

*for odd $i > 0$:* $\qquad$ $L_{i,b} = \phi(0,1;b) \sum_{j=0, j \text{ even}}^{i-1} \frac{(i-1)!!}{j!!} b^j$ .

*One can verify that for $b \geq 0$,*

$$
L_{i,b} \leq O(1) e^{-b^2/2} \cdot \sum_{j=0}^{i-1} \frac{(i-1)!!}{j!!} b^j
$$

*Proof.* The base cases $L_{0,b}$ and $L_{1,b}$ are easy to verify. Then the lemma comes from induction. $\square$

## A.5 Optimization

**Fact A.8.** *For every $B$-second-order smooth function $f : \mathbb{R}^d \to \mathbb{R}$, every $\varepsilon > 0$, every $\eta \in \left(0, O(\frac{\varepsilon^2}{B^2})\right]$, every fixed vector $x \in \mathbb{R}^d$, suppose there is a random vector $x_2 \in \mathbb{R}^d$ with $\mathbb{E}[x_2] = 0$ and $\|x_2\|_2 = 1$ satisfying*

$$\mathbb{E}_{x_2}[f\left(x + \sqrt{\eta}x_2\right)] \le f(x) - \eta\varepsilon \ .$$

*Then, $\lambda_{\min}(\nabla^2 f(x)) \le -\varepsilon$, where $\lambda_{\min}$ is the minimal eigenvalue.*

*Proof of Fact A.8.* We know that

$$f\left(x + \sqrt{\eta}x_2\right) = f(x) + \langle \nabla f(x), \sqrt{\eta}x_2 \rangle + \frac{1}{2}\left(\sqrt{\eta}x_2\right)^\top \nabla^2 f(x)\left(\sqrt{\eta}x_2\right) \pm O(B\eta^{1.5}).$$

Taking expectation, we know that

$$\mathbb{E}[f\left(x + \sqrt{\eta}x_2\right)] = f(x) + \eta\frac{1}{2}\mathbb{E}\left[x_2^\top \nabla^2 f(x)x_2\right] \pm O(B\eta^{1.5})$$

Thus, $\mathbb{E}\left[x_2^\top \nabla^2 f(x)x_2\right] \le -\varepsilon$, which completes the proof. $\square$

We also recall the following convergence theorem of SGD for escaping saddle point.[15]

**Lemma A.9** (escape saddle points, Theorem 6 of [19])**.** *Suppose a function $f : \mathbb{R}^d \to \mathbb{R}$ has its stochastic gradient bounded by $B$ in Euclidean norm, is absolutely bounded $|f(x)| \le B$, is $B$-smooth, and is $B$-second-order smooth, then for every $\delta > 0$, every $p \in (0,1)$, with probability at least $1 - p$, noisy SGD outputs a point $x_T$ after $T = \mathsf{poly}(d, B, 1/\delta, 1/p)$ iterations such that*

$$\nabla^2 f(x_T) \succeq -\delta\mathbf{I} \quad and \quad f(x_T) \le f(x_0) + \delta \cdot \mathsf{poly}(d, B, 1/p)$$

## A.6 Rademacher Complexity

Let $\mathcal{F}$ be a set of functions $\mathbb{R}^d \to \mathbb{R}$ and $\mathcal{X} = (x_1, \dots, x_N)$ be a finite set of samples. Recall the *empirical Rademacher complexity* with respect to $\mathcal{X}$ of $\mathcal{F}$ is

$$\widehat{\mathfrak{R}}(\mathcal{X}; \mathcal{F}) := \mathbb{E}_{\xi \sim \{\pm 1\}^N}\left[\sup_{f \in \mathcal{F}} \frac{1}{N}\sum_{i=1}^{N} \xi_i f(x_i)\right]$$

**Lemma A.10** (Rademacher generalization)**.** *Suppose $\mathcal{X} = (x_1, \dots, x_N)$ where each $x_i$ is generated i.i.d. from a distribution $\mathcal{D}$. If every $f \in \mathcal{F}$ satisfies $|f| \le b$, for every $\delta \in (0,1)$ with probability at least $1 - \delta$ over the randomness of $\mathcal{Z}$, it satisfies*

$$\sup_{f \in \mathcal{F}}\left|\mathbb{E}_{x \sim \mathcal{D}}[f(x)] - \frac{1}{N}\sum_{i=1}^{N} f(x_i)\right| \le 2\widehat{\mathfrak{R}}(\mathcal{Z}; \mathcal{F}) + O\left(\frac{b\sqrt{\log(1/\delta)}}{\sqrt{N}}\right) \ .$$

**Corollary A.11.** *If $\mathcal{F}_1, \dots, \mathcal{F}_k$ are $k$ classes of functions $\mathbb{R}^d \to \mathbb{R}$ and $L_x \colon \mathbb{R}^k \to [-b, b]$ is a 1-Lipschitz continuous function for any $x \sim \mathcal{D}$, then*

$$\sup_{f_1 \in \mathcal{F}_1, \dots, f_k \in \mathcal{F}_k}\left|\mathbb{E}_{x \sim \mathcal{D}}[L_x(f_1(x), \dots, f_k(x))] - \frac{1}{N}\sum_{i=1}^{N} L_x(f(x_i))\right| \le O\Big(\sum_{r=1}^{k} \widehat{\mathfrak{R}}(\mathcal{Z}; \mathcal{F}_r)\Big) + O\left(\frac{b\sqrt{\log(1/\delta)}}{\sqrt{N}}\right) \ .$$

*Proof.* Let $\mathcal{F}'$ be the class of functions by composing $L$ with $\mathcal{F}_1, \dots, \mathcal{F}_k$, that is, $\mathcal{F}' = \{L_x \circ (f_1, \dots, f_k) \mid f_1 \in \mathcal{F}_1 \cdots f_k \in \mathcal{F}_k\}$. By the (vector version) of the contraction lemma of Rademacher complexity[16] it satisfies $\widehat{\mathfrak{R}}(\mathcal{Z}; \mathcal{F}') \le O(1) \cdot \sum_{r=1}^{k} \widehat{\mathfrak{R}}(\mathcal{Z}; \mathcal{F}_r)$. $\square$

**Proposition A.12.** *We recall some basic properties of the Rademacher complexity. Let $\sigma\colon \mathbb{R} \to \mathbb{R}$ be a fixed 1-Lipschitz function.*

(a) *($\ell_2$ linear) Suppose $\|x\|_2 \leq 1$ for all $x \in \mathcal{X}$. The class $\mathcal{F} = \{x \mapsto \langle w, x\rangle \mid \|w\|_2 \leq B\}$ has Rademacher complexity $\widehat{R}(\mathcal{X}; \mathcal{F}) \leq O(\frac{B}{\sqrt{N}})$.*

(b) *($\ell_1$ linear) Suppose $\|x\|_\infty \leq 1$ for all $x \in \mathcal{X} \subseteq \mathbb{R}^m$. The class $\mathcal{F} = \{x \mapsto \langle w, x\rangle \mid \|w\|_1 \leq B\}$ has Rademacher complexity $\widehat{R}(\mathcal{X}; \mathcal{F}) \leq O(\frac{B\sqrt{\log m}}{\sqrt{N}})$.*

(c) *(addition) $\widehat{R}(\mathcal{X}; \mathcal{F}_1 + \mathcal{F}_2) = \widehat{R}(\mathcal{X}; \mathcal{F}_1) + \widehat{R}(\mathcal{X}; \mathcal{F}_2)$.*

(d) *(contraction) $\widehat{R}(\mathcal{X}; \sigma \circ \mathcal{F}) \leq \widehat{R}(\mathcal{X}; \mathcal{F})$.*

(e) *Given $\mathcal{F}_1, \dots, \mathcal{F}_m$ classes of functions $\mathcal{X} \to \mathbb{R}$ and suppose $w \in \mathbb{R}^m$ is a fixed vector, then*
$$\mathcal{F}' = \left\{ x \mapsto \textstyle\sum_{j=1}^m w_j \sigma(f_j(x)) \,\Big|\, f_j \in \mathcal{F}_j \right\}$$
*satisfies $\widehat{R}(\mathcal{X}; \mathcal{F}') \leq 2\|w\|_1 \max_{j \in [m]} \widehat{R}(\mathcal{X}; \mathcal{F}_j)$.*

(f) *Given $\mathcal{F}_1, \dots, \mathcal{F}_m$ classes of functions $\mathcal{X} \to \mathbb{R}$ and suppose for each $j \in [m]$ there exist a function $f_j^{(0)} \in \mathcal{F}_j$ satisfying $\sup_{x \in \mathcal{X}} |\sigma(f_j^{(0)}(x))| \leq R$, then*
$$\mathcal{F}' = \left\{ x \mapsto \textstyle\sum_{j=1}^m v_j \sigma(f_j(x)) \,\Big|\, f_j \in \mathcal{F}_j \wedge \|v\|_1 \leq B \wedge \|v\|_\infty \leq D \right\}$$
*satisfies $\widehat{R}(\mathcal{X}; \mathcal{F}') \leq 2D \sum_{j \in [m]} \widehat{R}(\mathcal{X}; \mathcal{F}_j) + O\big(\frac{BR\log m}{\sqrt{N}}\big)$.*

*Proof.* The first three are trivial, and the contraction lemma is classical (see for instance [45]). The derivations of Lemma A.12e and Lemma A.12f are less standard.

For Lemma A.12e, let us choose an arbitrary $f_j^{(0)}$ from each $\mathcal{F}_j$. We write $f \in \mathcal{F}$ to denote $(f_1, \dots, f_m) \in \mathcal{F}_1 \times \cdots \times \mathcal{F}_m$.

$$\mathbb{E}_\xi\Big[ \sup_{f \in \mathcal{F}} \sum_{i \in [N]} \xi_i \sum_{j=1}^m w_j \sigma(f_j(x_i)) \Big] \overset{①}{=} \mathbb{E}_\xi\Big[ \sup_{f \in \mathcal{F}} \sum_{i \in [N]} \xi_i \sum_{j=1}^m w_j \big(\sigma(f_j(x_i)) - \sigma(f_j^{(0)}(x_i))\big) \Big]$$

$$\leq \mathbb{E}_\xi\Big[ \sup_{f \in \mathcal{F}} \sum_{j=1}^m |w_j| \Big| \sum_{i \in [N]} \xi_i \big(\sigma(f_j(x_i)) - \sigma(f_j^{(0)}(x_i))\big) \Big| \Big]$$

$$= \mathbb{E}_\xi\Big[ \sum_{j=1}^m |w_j| \sup_{f_j \in \mathcal{F}_j} \Big| \sum_{i \in [N]} \xi_i \big(\sigma(f_j(x_i)) - \sigma(f_j^{(0)}(x_i))\big) \Big| \Big]$$

$$\overset{②}{\leq} 2\mathbb{E}_\xi\Big[ \sum_{j=1}^m |w_j| \sup_{f_j \in \mathcal{F}_j} \sum_{i \in [N]} \xi_i \big(\sigma(f_j(x_i)) - \sigma(f_j^{(0)}(x_i))\big) \Big]$$

$$\overset{③}{=} 2\mathbb{E}_\xi\Big[ \sum_{j=1}^m |w_j| \sup_{f_j \in \mathcal{F}_j} \sum_{i \in [N]} \xi_i \sigma(f_j(x_i)) \Big]$$

$$\leq 2\|w\|_1 \cdot N \max_{j \in [m]} \widehat{\mathfrak{R}}(\mathcal{X}; \sigma \circ \mathcal{F}_j) \overset{④}{\leq} 2N\|w\|_1 \cdot \max_{j \in [m]} \widehat{\mathfrak{R}}(\mathcal{X}; \mathcal{F}_j) \ .$$

Above, ① is because $\xi_i w_j \sigma(f_j^{(0)}(x_i))$ is independent of $f_j$ and thus zero in expectation; ② uses the non-negativity of $\sup_{f_j \in \mathcal{F}_j} \sum_{i \in [N]} \xi_i \big(\sigma(f_j(x_i)) - \sigma(f_j^{(0)}(x_i))\big)$; ③ is for the same reason as ①; and ④ is by Proposition A.12d.

---

vector case without absolute value, see [38].

For Lemma A.12f,

$$\mathbb{E}_\xi\Big[\sup_{f\in\mathcal{F},\|v\|_1\leq B,\|v\|_\infty\leq D}\sum_{i\in[N]}\xi_i\sum_{j=1}^m v_j\sigma(f_j(x_i))\Big]$$

$$=\mathbb{E}_\xi\Big[\sup_{f\in\mathcal{F},\|v\|_1\leq B,\|v\|_\infty\leq D}\sum_{i\in[N]}\xi_i\sum_{j=1}^m v_j\big(\sigma(f_j(x_i))-\sigma(f_j^{(0)}(x_i))\big)+\sum_{i\in[N]}\xi_i\sum_{j=1}^m v_j\sigma(f_j^{(0)}(x_i))\Big]$$

$$\leq\mathbb{E}_\xi\Big[\sup_{f\in\mathcal{F},\|v\|_\infty\leq D}\sum_{i\in[N]}\xi_i\sum_{j=1}^m v_j\big(\sigma(f_j(x_i))-\sigma(f_j^{(0)}(x_i))\big)\Big]+\mathbb{E}_\xi\Big[\sup_{\|v\|_1\leq B}\sum_{i\in[N]}\xi_i\sum_{j=1}^m v_j\sigma(f_j^{(0)}(x_i))\Big]$$

$$\leq D\mathbb{E}_\xi\Big[\sum_{j\in[m]}\sup_{f_j\in\mathcal{F}_j}\Big|\sum_{i\in[N]}\xi_i\big(\sigma(f_j(x_i))-\sigma(f_j^{(0)}(x_i))\big)\Big|\Big]+\mathbb{E}_\xi\Big[\sup_{\|v\|_1\leq B}\sum_{i\in[N]}\xi_i\sum_{j=1}^m v_j\sigma(f_j^{(0)}(x_i))\Big]$$

$$\overset{①}{\leq} 2D\mathbb{E}_\xi\Big[\sum_{j\in[m]}\sup_{f_j\in\mathcal{F}_j}\sum_{i\in[N]}\xi_i\big(\sigma(f_j(x_i))-\sigma(f_j^{(0)}(x_i))\big)\Big]+\mathbb{E}_\xi\Big[\sup_{\|v\|_1\leq B}\sum_{i\in[N]}\xi_i\sum_{j=1}^m v_j\sigma(f_j^{(0)}(x_i))\Big]$$

$$=2D\mathbb{E}_\xi\Big[\sum_{j\in[m]}\sup_{f_j\in\mathcal{F}_j}\sum_{i\in[N]}\xi_i\sigma(f_j(x_i))\Big]+\mathbb{E}_\xi\Big[\sup_{\|v\|_1\leq B}\sum_{i\in[N]}\xi_i\sum_{j=1}^m v_j\sigma(f_j^{(0)}(x_i))\Big]$$

$$=2D\cdot N\sum_{j\in[m]}\widehat{\mathfrak{R}}(\mathcal{X};\sigma\circ\mathcal{F}_j)+\mathbb{E}_\xi\Big[\sup_{\|v\|_1\leq B}\sum_{i\in[N]}\xi_i\sum_{j=1}^m v_j\sigma(f_j^{(0)}(x_i))\Big]$$

$$\overset{②}{=}2D\cdot N\sum_{j\in[m]}\widehat{\mathfrak{R}}(\mathcal{X};\sigma\circ\mathcal{F}_j)+O(BR\sqrt{N}\log m)\ .$$

Above, ① is from the same derivation as Proposition A.12e; and ② uses Proposition A.12b by viewing function class $\big\{x\mapsto\sum_{j=1}^m v_j\cdot\sigma(f_j^{(0)}(x))\big\}$ as linear in $v$. □

## B  Proofs for Two-Layer Networks

Recall from (3.1) the target $F^*=(f_1^*,\cdots,f_k^*)$ for our two-layer case is

$$f_r^*(x):=\sum_{i=1}^p a_{r,i}^*\phi_i(\langle w_{1,i}^*,x\rangle)\langle w_{2,i}^*,x\rangle$$

We consider another function defined as $G(x;W)=(g_1(x;W),\ldots,g_k(x;W))$ for the weight matrix $W$, where

$$g_r(x;W):=\sum_{i=1}^m a_{r,i}^{(0)}(\langle w_i,x\rangle+b_i^{(0)})\mathbb{I}[\langle w_i^{(0)},x\rangle+b_i^{(0)}\geq 0]$$

where $w_i$ is the $r$-th row of $W$ and $w_i^{(0)}$ is the $r$-th row of $W^{(0)}$ (our random initialization). We call this $G$ a pseudo network. For convenience, we also define a pseudo network $G^{(b)}(x;W)=(g_1^{(b)}(x;W),\ldots,g_k^{(b)}(x;W))$ without bias

$$g_r^{(b)}(x;W):=\sum_{i=1}^m a_{r,i}^{(0)}\langle w_i,x\rangle\mathbb{I}[\langle w_i^{(0)},x\rangle+b_i^{(0)}\geq 0],$$

**Roadmap.** Our analysis begins with showing that w.h.p. over the random initialization, there exists a pseudo network in the neighborhood of the initialization that can approximate the target function (see Section B.1). We then show that, near the initialization, the pseudo network approximates the actual ReLU network $F$ (see Section B.2). Therefore, there exists a ReLU network near the initialization approximating the target. Furthermore, it means that the loss surface of the ReLU network is close to that of the pseudo network, which is convex. This then allows us to show the training converges (see Section B.3). Combined with a generalization bound localized to the initialization, we can prove the final theorem that SGD learns a network with small risk.

The proof is much simpler than the three-layer case. First, the optimization landscape is almost convex, so a standard argument for convex optimization applies. While for three-layer case, the optimization landscape is no longer this nice and needs an escaping-from-saddle-point argument, which in turn requires several technicalities (smoothing, explicit regularization, weight decay, and Dropout-like noise). Second, the main technical part of the analysis, the proof of the existential result, only needs to deal with approximating one layer of neurons in the target, while in the three-layer case, a composition of the neurons needs to be approximating, requiring additional delicate arguments. It is also similar with the generalization bounds. However, we believe that the analysis in the two-layer case already shows some of the key ideas, and suggest the readers to read it before reading that for the three-layer case.

## B.1 Existential Result

The main focus of this subsection is to show that there exists a good pseudo network near the initialization. (Combining with the coupling result of the next subsection, this translates to the *real* network near the initialization.)

**Lemma B.1.** *For every $\varepsilon \in (0, \frac{1}{pk\mathfrak{C}_s(\phi,1)})$, letting $\varepsilon_a = \varepsilon/\widetilde{\Theta}(1)$, there exists*

$$M = \mathsf{poly}(\mathfrak{C}_\varepsilon(\phi,1), 1/\varepsilon) \quad and \quad C_0 = \widetilde{\Theta}\big(\mathfrak{C}_\varepsilon(\phi,1)\big)$$

*such that if $m \geq M$, then with high probability there exists $W^{\circledast} = (w_1^{\circledast}, \ldots, w_m^{\circledast})$ with $\|W^{\circledast}\|_{2,\infty} \leq \frac{kpC_0}{\varepsilon_a m}$ and $\|W^{\circledast}\|_F \leq \widetilde{O}(\frac{kp\mathfrak{C}_s(\phi,1)}{\varepsilon_a \sqrt{m}})$ and*

$$\mathbb{E}_{(x,y)\sim\mathcal{D}} \left[ \sum_{r=1}^k \left| f_r^*(x) - g_r^{(b)}(x; W^{\circledast}) \right| \right] \leq \varepsilon,$$

*and consequently,*

$$\mathbb{E}_{(x,y)\sim\mathcal{D}} \left[ L(G^{(b)}(x; W^{\circledast}), y) \right] \leq \mathsf{OPT} + \varepsilon.$$

**Corollary B.2.** *In the same setting as Lemma B.1, we have that w.h.p.*

$$\mathbb{E}_{(x,y)\sim\mathcal{D}} \left[ \sum_{r=1}^k \left| f_r^*(x) - g_r(x; W^{(0)} + W^{\circledast}) \right| \right] \leq \varepsilon,$$

*and consequently,*

$$\mathbb{E}_{(x,y)\sim\mathcal{D}} \left[ L(G(x; W^{(0)} + W^{\circledast}), y) \right] \leq \mathsf{OPT} + \varepsilon.$$

*Proof of Lemma B.1.* Recall the pseudo network without bias is given by

$$g_r^{(b)}(x; W) = \sum_{i=1}^m a_{r,i}^{(0)}(\langle w_i, x \rangle + b_i^{(0)}) \mathbb{I}[\langle w_i^{(0)}, x \rangle + b_i^{(0)} \geq 0].$$

Also recall from Lemma 6.3 that, for each $i \in [p]$, there is function $h^{(i)} \colon \mathbb{R}^2 \to \mathbb{R}$ with $|h^{(i)}| \leq \mathfrak{C}_\varepsilon(\phi,1)$, satisfying

$$\left| \mathbb{E}\left[ \mathbb{I}_{\alpha_1 x_1 + \beta_1 \sqrt{1-x_1^2} + b_0 \geq 0} h^{(i)}(\alpha_1, b_0) \right] - \phi_i(x_1) \right| \leq \varepsilon$$

where $\alpha_1, \beta_1, b_0 \sim \mathcal{N}(0,1)$ are independent random Gaussians.

**Fit a single function $a_{r,i}^* \phi_i(\langle w_{1,i}^*, x \rangle) \langle w_{2,i}^*, x \rangle$.** We first fix some $r \in [k]$ and $i \in [p]$ and construct weights $w_j^{\circledast} \in \mathbb{R}^d$. Define

$$w_j^{\circledast} := \frac{1}{\varepsilon_a^2} a_{r,j}^{(0)} a_{r,i}^* h^{(i)}\left( \sqrt{m}\langle w_j^{(0)}, w_{1,i}^* \rangle, \sqrt{m} b_j^{(0)} \right) w_{2,i}^* \tag{B.1}$$

where $\sqrt{m}\langle w_j^{(0)}, w_{1,i}^* \rangle, \sqrt{m} b_j^{(0)}$ has the same distribution with $\alpha_1, b_0$ in Lemma 6.3. By Lemma 6.3, we have that

$$\mathbb{E}_{w_j^{(0)}, b_j^{(0)}, a_{r,j}^{(0)}} \left[ a_{r,j}^{(0)} \mathbb{I}_{\langle w_j^{(0)}, x \rangle + b_j^{(0)} \geq 0} \langle w_j^{\circledast}, x \rangle \right]$$

$$= \mathbb{E}_{w_j^{(0)}, b_j^{(0)}} \left[ a_{r,i}^* \mathbb{I}_{\langle w_j^{(0)}, x \rangle + b_j^{(0)} \geq 0} h^{(i)}\left( \sqrt{m}\langle w_j^{(0)}, w_{1,i}^* \rangle, \sqrt{m} b_j^{(0)} \right) \langle w_{2,i}^*, x \rangle \right]$$

$$= a^*_{r,i}\phi_i(\langle w^*_{1,i}, x\rangle)\langle w^*_{2,i}, x\rangle \pm \varepsilon.$$

**Fit a combination** $\sum_{i\in[p]} a^*_{r,i}\phi_i(\langle w^*_{1,i}, x\rangle)\langle w^*_{2,i}, x\rangle.$   We can re-define (the norm grows by a maximum factor of $p$)

$$w^{\circledast}_j = \frac{1}{\varepsilon_a^2} a^{(0)}_{r,j} \sum_{i\in[p]} a^*_{r,i} h^{(i)}\left(\sqrt{m}\langle w^{(0)}_j, w^*_{1,i}\rangle, \sqrt{m}b^{(0)}_j\right) w^*_{2,i}$$

and the same above argument gives

$$\mathbb{E}_{w^{(0)}_j, b^{(0)}_j, a^{(0)}_{r,j}}\left[a^{(0)}_{r,j}\mathbb{I}_{\langle w^{(0)}_j, x\rangle + b^{(0)}_j \geq 0}\langle w^{\circledast}_j, x\rangle\right] = \sum_{i\in[p]} a^*_{r,i}\phi_i(\langle w^*_{1,i}, x\rangle)\langle w^*_{2,i}, x\rangle \pm \varepsilon p.$$

**Fit multiple outputs.**   If there are $k$ outputs let us re-define (the norm grows by a maximum factor of $k$)

$$w^{\circledast}_j = \frac{1}{\varepsilon_a^2} \sum_{r\in[k]} a^{(0)}_{r,j} \sum_{i\in[p]} a^*_{r,i} h^{(i)}\left(\sqrt{m}\langle w^{(0)}_j, w^*_{1,i}\rangle, \sqrt{m}b^{(0)}_j\right) w^*_{2,i}. \tag{B.2}$$

and consider the quantity

$$\Xi_{r,j} := a^{(0)}_{r,j}\langle w^{\circledast}_j, x\rangle\mathbb{I}[\langle w^{(0)}_j, x\rangle + b^{(0)}_j \geq 0] \ .$$

By definition of the initialization, we know that for $r' \neq r$, $\mathbb{E}[a^{(0)}_{r,j}a^{(0)}_{r',j}] = 0$. Thus, for every $r \in [k]$, it satisfies

$$\mathbb{E}_{w^{(0)}_j, b^{(0)}_j, a^{(0)}_{1,j},\ldots,a^{(0)}_{k,j}}\left[\Xi_{r,j}\right]$$

$$= \mathbb{E}_{w^{(0)}_j, b^{(0)}_j, a^{(0)}_{1,j},\ldots,a^{(0)}_{k,j}}\left[\sum_{r'\in[k]} \frac{a^{(0)}_{r,j}a^{(0)}_{r',j}}{\varepsilon_a^2} \sum_{i\in[p]}\mathbb{I}_{\langle w^{(0)}_j, x\rangle + b^{(0)}_j \geq 0}a^*_{r',i} h^{(i)}\left(\sqrt{m}\langle w^{(0)}_j, w^*_{1,i}\rangle, \sqrt{m}b^{(0)}_j\right)\langle w^*_{2,i}, x\rangle\right]$$

$$= \mathbb{E}_{w^{(0)}_j, b^{(0)}_j}\left[\sum_{i\in[p]}\mathbb{I}_{\langle w^{(0)}_j, x\rangle + b^{(0)}_j \geq 0}a^*_{r,i} h^{(i)}\left(\sqrt{m}\langle w^{(0)}_j, w^*_{1,i}\rangle, \sqrt{m}b^{(0)}_j\right)\langle w^*_{2,i}, x\rangle\right]$$

$$= \sum_{i\in[p]} a^*_{r,i}\phi_i(\langle w^*_{1,i}, x\rangle))\langle w^*_{2,i}, x\rangle \pm p\varepsilon = f^*_r(x) \pm p\varepsilon \ .$$

Now, re-scaling each $w^{\circledast}_j$ by a factor of $\frac{1}{m}$ and re-scaling $\varepsilon$ by $\frac{1}{2pk}$, we can write

$$g^{(b)}_r(x; W^{\circledast}) = \sum_{j=1}^m \Xi_{r,j} \quad \text{and} \quad \mathbb{E}\left[g^{(b)}_r(x; W^{\circledast})\right] = f^*_r(x) \pm \frac{\varepsilon}{2k} \ .$$

Now, we apply the concentration from Lemma A.1, which implies for our parameter choice of $m$, with high probability

$$|g^{(b)}_r(x; W^{\circledast}) - f^*_r(x)| \leq \frac{\varepsilon}{k} \ .$$

The above concentration holds for every fixed $x$ with high probability, and thus also holds in expectation with respect to $(x, y) \sim \mathcal{D}$. This proves the first statement. As for the second statement on $L(G^{(b)}(x; W^{\circledast}), y)$, it follows from the Lipschitz continuity of $L$.

**Norm on $W^{\circledast}$.**   According to its definition in (B.2), we have for each $j \in [m]$, with high probability $\|w^{\circledast}_j\|_2 \leq \widetilde{O}\left(\frac{kpC_0}{\varepsilon_a m}\right)$ (here the additional $\frac{1}{m}$ is because we have re-scaled $w^{\circledast}_j$ by $\frac{1}{m}$). This means $\|W^{\circledast}\|_{2,\infty} \leq \widetilde{O}\left(\frac{kpC_0}{\varepsilon_a m}\right)$. As for the Frobenius norm,

$$\|W^{\circledast}\|_F^2 = \sum_{j\in[m]} \|w^{\circledast}_j\|^2 \leq \sum_{j\in[m]} \widetilde{O}\left(\frac{k^2p}{\varepsilon_a^2 m^2}\right) \cdot \sum_{i\in[p]} h^{(i)}\left(\sqrt{m}\langle w^{(0)}_j, w^*_{1,i}\rangle, \sqrt{m}b^{(0)}_j\right)^2 \tag{B.3}$$

Now, for each $i \in [p]$, we know that $\sum_{j\in[m]} h^{(i)}\left(\sqrt{m}\langle w^{(0)}_j, w^*_{1,i}\rangle, \sqrt{m}b^{(0)}_j\right)^2$ is a summation of i.i.d. random variables, each with expectation at most $\mathfrak{C}_{\mathfrak{s}}(\phi, 1)^2$ by Lemma 6.3. Applying concen-

tration, we have with high probability

$$\sum_{j\in[m]} h^{(i)}\left(\sqrt{m}\langle w_j^{(0)}, w_{1,i}^*\rangle, \sqrt{m}b_j^{(0)}\right)^2 \le m \cdot \mathfrak{C}_{\mathfrak{s}}(\phi,1)^2 + \sqrt{m} \cdot C_0^2 \le 2m\mathfrak{C}_{\mathfrak{s}}(\phi,1)^2$$

Putting this back to (B.3) we have $\|W^*\|_F^2 \le \widetilde{O}(\frac{k^2 p^2 \mathfrak{C}_{\mathfrak{s}}(\phi,1)^2}{\varepsilon_a^2 m})$. $\qquad\square$

*Proof of Corollary B.2.* Let $W^*$ be the weights constructed in Lemma B.1 to approximate up to error $\varepsilon/2$. Then

$$|g_r(x; W^{(0)} + W^*) - g_r^{(b)}(x; W^*)| = \left|\sum_{i=1}^m a_{r,i}^{(0)}(\langle w_i^{(0)}, x\rangle + b_i^{(0)})\mathbb{I}[\langle w_i^{(0)}, x\rangle + b_i^{(0)} \ge 0]\right|.$$

By standard concentration (which uses the randomness of $w^{(0)}$ together with the randomness of $a_{r,1}^{(0)}, \ldots, a_{r,m}^{(0)}$), the above quantity is with high probability bounded by $\widetilde{O}(\varepsilon_a) = \varepsilon/2$. This is the only place we need parameter choice $\varepsilon_a$. $\qquad\square$

## B.2   Coupling

Here we show that the weights after a properly bounded amount of updates stay close to the initialization, and thus the pseudo network is close to the real network using the same weights.

**Lemma B.3** (Coupling).   *For every unit vector $x$, w.h.p. over the random initialization, for every time step $t \ge 1$, we have the following. Denote $\tau = \varepsilon_a \eta t$.*

*(a) For at most $\widetilde{O}(\tau\sqrt{km})$ fraction of $i \in [m]$:*

$$\mathbb{I}[\langle w_i^{(0)}, x\rangle + b_i^{(0)} \ge 0] \neq \mathbb{I}[\langle w_i^{(t)}, x\rangle + b_i^{(0)} \ge 0].$$

*(b) For every $r \in [k]$,*

$$\left|f_r(x; W^{(0)} + W_t) - g_r(x; W^{(0)} + W_t)\right| = \widetilde{O}(\varepsilon_a k \tau^2 m^{3/2}).$$

*(c) For every $y$:*

$$\left\|\frac{\partial}{\partial W}L(F(x; W^{(0)} + W_t), y) - \frac{\partial}{\partial W}L(G(x; W^{(0)} + W_t), y)\right\|_{2,1}$$
$$\le \widetilde{O}(\varepsilon_a k\tau m^{3/2} + \varepsilon_a^2 k^2 \tau^2 m^{5/2}).$$

*Proof of Lemma B.3.* Let us recall

$$f_r(x; W) = \sum_{i=1}^m a_{r,i}^{(0)}(\langle w_i, x\rangle + b_i^{(0)})\mathbb{I}[\langle w_i, x\rangle + b_i^{(0)} \ge 0]$$

$$g_r(x; W) = \sum_{i=1}^m a_{r,i}^{(0)}(\langle w_i, x\rangle + b_i^{(0)})\mathbb{I}[\langle w_i^{(0)}, x\rangle + b_i^{(0)} \ge 0]. \qquad (\text{B.4})$$

(a) W.h.p. over the random initialization, there is $B = \widetilde{O}(1)$ so that every $|a_{r,i}^{(0)}| \le \varepsilon_a B$. Thus, by the 1-Lipschitz continuity of $L$, for every $i \in [m]$ and every $t \ge 0$,

$$\left\|\frac{\partial f_r(x; W^{(0)} + W_t)}{\partial w_i}\right\|_2 \le \varepsilon_a B \quad \text{and} \quad \left\|\frac{\partial L(F(x; W^{(0)} + W_t), y)}{\partial w_i}\right\|_2 \le \sqrt{k}\varepsilon_a B \quad (\text{B.5})$$

which implies that $\left\|w_i^{(t)} - w_i^{(0)}\right\|_2 \le \sqrt{k}B\varepsilon_a\eta t = \sqrt{k}B\tau$. Accordingly, define $\mathcal{H}$

$$\mathcal{H} := \left\{i \in [m] \,\Big|\, \left|\langle w_i^{(0)}, x\rangle + b_i^{(0)}\right| \ge 2\sqrt{k}B\tau\right\}.$$

so it satisfies for every $i \in \mathcal{H}$,

$$\left|\left(\langle w_i^{(t)}, x\rangle + b_i^{(0)}\right) - \left(\langle w_i^{(0)}, x\rangle + b_i^{(0)}\right)\right| \le \sqrt{k}B\tau$$

which implies $\mathbb{I}[\langle w_i^{(0)}, x\rangle + b_i^{(0)} \geq 0] = \mathbb{I}[\langle w_i^{(t)}, x\rangle + b_i^{(0)} \geq 0]$.

Now, we need to bound the size of $\mathcal{H}$. Since $\langle w_i^{(0)}, x\rangle \sim \mathcal{N}(0, 1/m)$, and $b_i^{(0)} \sim \mathcal{N}(0, 1/m)$, by standard property of Gaussian one can derive $|\mathcal{H}| \geq m(1 - \widetilde{O}(\sqrt{k}B\tau\sqrt{m})) = m(1 - \widetilde{O}(\tau\sqrt{km}))$ with high probability.

(b) It is clear from (B.4) that $f_r$ and $g_r$ only differ on the indices $i \notin \mathcal{H}$. For such an index $i \notin \mathcal{H}$, the sign of $\langle w_i^{(t)}, x\rangle + b_i^{(0)}$ is different from that of $\langle w_i^{(0)}, x\rangle + b_i^{(0)}$ and their difference is at most $\sqrt{k}B\tau$. This contributes at most $\varepsilon_a B \cdot \sqrt{k}B\tau$ difference between $f_r$ and $g_r$. Then the bound follows from that there are only $\widetilde{O}(\tau m\sqrt{km})$ many $i \notin \mathcal{H}$.

(c) Note that
$$\frac{\partial}{\partial w_i} L(F(x; W^{(0)} + W_t), y) = \nabla L(F(x; W^{(0)} + W_t), y) \frac{\partial}{\partial w_i} F(x; W^{(0)} + W_t)$$
By the Lipschitz smoothness assumption on $L$ and Lemma B.3b, we have
$$\|\nabla L(F(x; W^{(0)} + W_t), y) - \nabla L(G(x; W^{(0)} + W_t), y)\|_2 \leq \widetilde{O}(\varepsilon_a k^{3/2}\tau^2 m^{3/2}) \ . \quad \text{(B.6)}$$
For $i \in \mathcal{H}$ we have $\mathbb{I}[\langle w_i^{(0)}, x\rangle + b_i^{(0)} \geq 0] = \mathbb{I}[\langle w_i^{(t)}, x\rangle + b_i^{(0)} \geq 0]$, so $\frac{\partial F(x; W^{(0)} + W_t)}{\partial w_i} = \frac{\partial G(x; W^{(0)} + W_t)}{\partial w_i}$, and thus the difference is only caused by (B.6). Using (B.5), each such $i$ contributes at most $\widetilde{O}(\varepsilon_a^2 k^2\tau^2 m^{3/2})$, totaling $\widetilde{O}(\varepsilon_a^2 k^2\tau^2 m^{5/2})$.

For $i \notin \mathcal{H}$, it contributes at most $\sqrt{k}\varepsilon_a B$ because of (B.5), and there are $\widetilde{O}(\tau m\sqrt{km})$ many such $i$'s, totaling $\widetilde{O}(\varepsilon_a k\tau m^{3/2})$.

$\square$

## B.3 Optimization

Recall for $z = (x, y)$
$$L_F(z; W_t) := L(F(x; W^{(0)} + W_t), y) \ ,$$
$$L_G(z; W_t) := L(G(x; W^{(0)} + W_t), y) \ .$$
For the set of samples $\mathcal{Z}$, define
$$L_F(\mathcal{Z}; W) := \frac{1}{|\mathcal{Z}|} \sum_{(x,y)\in\mathcal{Z}} L(F(x; W + W^{(0)}), y) \ ,$$
$$L_G(\mathcal{Z}; W) := \frac{1}{|\mathcal{Z}|} \sum_{(x,y)\in\mathcal{Z}} L(G(x; W + W^{(0)}), y) \ .$$
We show the following lemma:

**Lemma B.4.** *For every $\varepsilon \in (0, \frac{1}{pk\mathfrak{C}_\mathfrak{s}(\phi,1)})$, letting $\varepsilon_a = \varepsilon/\widetilde{\Theta}(1)$ and $\eta = \widetilde{\Theta}(\frac{1}{\varepsilon km})$, there exists*
$$M = \mathsf{poly}(\mathfrak{C}_\varepsilon(\phi, 1), 1/\varepsilon) \quad \text{and} \quad T = \Theta\Big(\frac{k^3 p^2 \cdot \mathfrak{C}_\mathfrak{s}(\phi, 1)^2}{\varepsilon^2}\Big)$$
*such that, w.h.p., if $m \geq M$ then*
$$\frac{1}{T}\sum_{t=0}^{T-1} L_F(\mathcal{Z}; W_t) \leq \mathsf{OPT} + \varepsilon.$$

*Proof of Lemma B.4.* Let $W^*$ be constructed in Corollary B.2. Recall $L(\cdot, y)$ is convex and $G(x; W)$ is linear in $W$ so $L_G(z; W)$ is convex in $W$. By such convexity, we have
$$L_G(\mathcal{Z}; W_t) - L_G(\mathcal{Z}; W^*) \leq \langle \nabla L_G(\mathcal{Z}; W_t), W_t - W^*\rangle$$
$$\leq \|\nabla L_G(\mathcal{Z}; W_t) - \nabla L_F(\mathcal{Z}; W_t)\|_{2,1}\|W_t - W^*\|_{2,\infty}$$
$$+ \langle \nabla L_F(\mathcal{Z}; W_t), W_t - W^*\rangle.$$

We also have

$$\|W_{t+1} - W^*\|_F^2 = \|W_t - \eta \nabla L_F(z^{(t)}, W_t) - W^*\|_F^2$$
$$= \|W_t - W^*\|_F^2 - 2\eta \langle \nabla L_F(z^{(t)}, W_t), W_t - W^* \rangle$$
$$+ \eta^2 \|\nabla L_F(z^{(t)}, W_t)\|_F^2,$$

so

$$L_G(\mathcal{Z}; W_t) - L_G(\mathcal{Z}; W^*) \le \|\nabla L_G(\mathcal{Z}; W_t) - \nabla L_F(\mathcal{Z}; W_t)\|_{2,1} \|W_t - W^*\|_{2,\infty}$$
$$+ \frac{\|W_t - W^*\|_F^2 - \mathbb{E}_{z^{(t)}}[\|W_{t+1} - W^*\|_F^2]}{2\eta}$$
$$+ \frac{\eta}{2} \|\nabla L_F(W_t, z^{(t)})\|_F^2. \tag{B.7}$$

Recall from (B.5) that with high probability over the random initialization,

$$\|W_t\|_{2,\infty} = \widetilde{O}(\sqrt{k}\varepsilon_a \eta t) \ , \quad \|W_t - W^*\|_{2,\infty} = \widetilde{O}(\sqrt{k}\varepsilon_a \eta t + \frac{kpC_0}{\varepsilon_a m}) \tag{B.8}$$
$$\text{and} \quad \|\nabla L_F(W_t, z^{(t)})\|_F^2 = \widetilde{O}(\varepsilon_a^2 km) \ ,$$

where $C_0$ is as in Lemma B.1. By Lemma B.3c, w.h.p. we know

$$\|\nabla L_G(\mathcal{Z}; W_t) - \nabla L_F(\mathcal{Z}; W_t)\|_{2,1} \le \Delta = \widetilde{O}(\varepsilon_a^2 k\eta T m^{3/2} + \varepsilon_a^4 k^2(\eta T)^2 m^{5/2}).$$

Therefore, averaging up (B.7) from $t = 0$ to $T - 1$ we have that

$$\frac{1}{T}\sum_{t=0}^{T-1} \mathbb{E}_{sgd}[L_G(\mathcal{Z}; W_t)] - L_G(\mathcal{Z}; W^*) \le \widetilde{O}\left(\sqrt{k}\varepsilon_a \eta T\Delta + \frac{kpC_0}{\varepsilon_a m}\Delta\right) + \frac{\|W_0 - W^*\|_F^2}{2\eta T}$$
$$+ \widetilde{O}(k\varepsilon_a^2 \eta m).$$

Note that $\|W_0 - W^*\|_F^2 = \|W^*\|_F^2 \le \widetilde{O}(\frac{k^2p^2\mathfrak{C}_\mathfrak{s}(\phi,1)^2}{\varepsilon_a^2 m})$ from Lemma B.1. Also recall $\varepsilon_a = \Theta(\varepsilon)$. By choosing $\eta = \widetilde{\Theta}(\frac{\varepsilon}{km\varepsilon_a^2})$ and $T = \widetilde{\Theta}\left(k^3p^2\mathfrak{C}_\mathfrak{s}(\phi,1)^2/\varepsilon^2\right)$ we have $\Delta = \widetilde{O}(k^6p^4\mathfrak{C}_\mathfrak{s}(\phi,1)^4 m^{1/2}/\varepsilon^2)$. Thus when $m$ is large enough we have:

$$\frac{1}{T}\sum_{t=0}^{T-1} \mathbb{E}_{sgd}[L_G(\mathcal{Z}; W_t)] - L_G(\mathcal{Z}; W^*) \le O(\varepsilon).$$

By the coupling in Lemma B.3b, we know that $L_F(\mathcal{Z}; W_t)$ is $o(\varepsilon)$-close to $L_G(\mathcal{Z}; W_t)$; by applying Corollary B.2, we know that $L_G(\mathcal{Z}; W^*)$ is $\varepsilon$-close to OPT. This finishes the proof. □

## B.4 Generalization

The generalization can be bounded via known Rademacher complexity results. Recall

$$f_r(x; W^{(0)} + W') := \sum_{i=1}^m a_{r,i}^{(0)}\sigma(\langle w_i^{(0)} + w_i', x\rangle + b_i^{(0)}) \ .$$

We have the following simple lemma (see also [35, Theorem 43])[17]

**Lemma B.5** (two-layer network Rademacher complexity). *For every $\tau_{w,\infty} \ge 0$, w.h.p. for every $r \in [k]$ and every $N \ge 1$, we have the empirical Rademacher complexity bounded by*

$$\frac{1}{N}\mathbb{E}_{\xi \in \{\pm 1\}^N}\left[\sup_{\|W'\|_{2,\infty} \le \tau_{w,\infty}} \sum_{i \in [N]} \xi_i f_r(x_i; W^{(0)} + W')\right] \le \widetilde{O}\left(\frac{\varepsilon_a m\tau_{w,\infty}}{\sqrt{N}}\right).$$

*Proof.* The proof consists of the following simple steps.

- $\{x \mapsto \langle w_j', x\rangle \mid \|w_j'\|_2 \le \tau_{w,\infty}\}$ has Rademacher complexity $O(\frac{\tau_{w,\infty}}{\sqrt{N}})$ by Proposition A.12a.

- $\{x \mapsto \langle w_j^{(0)} + w_j', x\rangle + b_j \mid \|w_j'\|_2 \le \tau_{w,\infty}\}$ has Rademacher complexity $O(\frac{\tau_{w,\infty}}{\sqrt{N}})$ because singleton class has zero complexity and adding it does not affect complexity by

Proposition A.12c.

- $\{x \mapsto f_r(x; W^{(0)} + W') \mid \|W'\|_{2,\infty} \le \tau_{w,\infty}\}$ has Rademacher complexity $\widetilde{O}(\frac{\varepsilon_a m \tau_{w,\infty}}{\sqrt{N}})$ because w.h.p. $\|a_r^{(0)}\|_1 \le \widetilde{O}(\varepsilon_a m)$ and Proposition A.12e. $\qquad \square$

## B.5 Theorem 1: Two-Layer

*Proof of Theorem 1.* First, we can apply Lemma B.4 to bound the training loss. That is

$$\frac{1}{T} \sum_{t=0}^{T-1} \mathbb{E}_{(x,y) \in \mathcal{Z}} L(F(x; W_t + W^{(0)}), y) \le \mathsf{OPT} + \varepsilon.$$

Also recall from (B.8) and our parameter choices for $\eta, T$ that

$$\|W_t\|_{2,\infty} \le O\left(\frac{\mathsf{poly}(k, p, \log m) \cdot \mathfrak{C}_\mathfrak{s}(\phi, 1)^2}{\varepsilon^2 m}\right).$$

so we can choose $\tau_{w,\infty} = O(\frac{\mathsf{poly}(k,p,\log m) \cdot \mathfrak{C}_\mathfrak{s}(\phi,1)^2}{\varepsilon^2 m})$. For each $(x, y) \sim \mathcal{D}$, it is a simple exercise to verify that $|L(F(x; W_t + W^{(0)}), y)| \le O(\frac{\mathsf{poly}(k,p,\log m) \cdot \mathfrak{C}_\mathfrak{s}(\phi,1)^2}{\varepsilon})$ with high probability.[18] Thus, we can plug the Rademacher complexity Lemma B.5 together with $b = O(\frac{\mathsf{poly}(k,p,\log m) \cdot \mathfrak{C}_\mathfrak{s}(\phi,1)^2}{\varepsilon})$ into standard generalization statement Corollary A.11. It gives

$$\left| \mathbb{E}_{(x,y) \in \mathcal{D}} L(F(x; W_t + W^{(0)}), y) - \mathbb{E}_{(x,y) \in \mathcal{Z}} L(F(x; W_t + W^{(0)}), y) \right|$$
$$\le O(\frac{\mathsf{poly}(k, p, \log m) \cdot \mathfrak{C}_\mathfrak{s}(\phi, 1)^2}{\varepsilon \sqrt{N}}) \ . \tag{B.9}$$

This completes the proof with large enough $N$. $\qquad \square$

*Remark* B.6. Strictly speaking, $|g_r(x; W)| \le \widetilde{O}(\varepsilon_a)$ does not hold for every $x$ in $\mathcal{D}$, thus the loss function $L$ is not absolutely bounded so one cannot apply Corollary A.11 directly.[19] We only have the statement that for each sample $x$, the loss function $|L(F(x; W_t + W^{(0)}), y)| \le b$ is bounded by some parameter $b$ *with high probability*. By union bound, with high probability this can hold for all the training samples (but possibly not all the testing samples). A simple fix here is to apply a truncation (for analysis purpose only) on the loss function $L$ to make it always bounded by $b$. Then, we can apply Corollary A.11: the population risk "$\mathbb{E}_{(x,y) \in \mathcal{D}} L(\cdots)$" in (B.9) becomes truncated but the empirical risk "$\mathbb{E}_{(x,y) \in \mathcal{Z}} L(\cdots)$" in (B.9) stays unchanged. In other words, the *truncated* population risk must be small according to Corollary A.11. Finally, we can remove truncation from the population risk, because in the rare event that $|L(F(x; W_t + W^{(0)}), y)|$ exceeds $b$, it is at most $\mathsf{poly}(m)$ so becomes negligible when evaluating the expectation $\mathbb{E}_{(x,y) \in \mathcal{D}} L(\cdots)$.

*Remark* B.7. In the above proof, it appears that $N$ scales with $\varepsilon^{-4}$ which may seemingly be larger than $T$ which only scales with $\varepsilon^{-2}$. We are aware of a proof that tightens $N$ to be on the order of $\varepsilon^{-2}$. It uses standard (but complicated) martingale analysis and creates extra difficulty that is irrelevant to neural networks in general. We choose not to present it for simplicity.

## C Proofs for Three-Layer Networks

Our three-layer proofs follow the same structure as our proof overview in Section 6.

### C.1 Existential Results

#### C.1.1 Lemma 6.3: Indicator to Function

Recall without loss of generality it suffices to prove Lemma 6.3a.

**Lemma 6.3a** (indicator to function). *For every smooth function $\phi$, every $\varepsilon \in \left(0, \frac{1}{\mathfrak{C}_{\mathfrak{s}}(\phi,1)}\right)$, we have that there exists a function $h : \mathbb{R}^2 \to [-\mathfrak{C}_\varepsilon(\phi,1), \mathfrak{C}_\varepsilon(\phi,1)]$ such that for every $x_1 \in [-1,1]$:*

$$\left| \mathbb{E}\left[ \mathbb{I}_{\alpha_1 x_1 + \beta_1 \sqrt{1-x_1^2} + b_0 \geq 0} h(\alpha_1, b_0) \right] - \phi(x_1) \right| \leq \varepsilon$$

*where $\alpha_1, \beta_1 \sim \mathcal{N}(0,1)$ and $b_0 \sim \mathcal{N}(0,1)$ are independent random variables. Furthermore:*

- *$h$ is $\mathfrak{C}_\varepsilon(\phi,1)$-Lipschitz on the first coordinate.*

- *$\mathbb{E}_{\alpha_1, b_0 \sim \mathcal{N}(0,1)}\left[ h(\alpha_1, b_0)^2 \right] \leq (\mathfrak{C}_{\mathfrak{s}}(\phi,1))^2.$*

For notation simplicity, let us denote $w_0 = (\alpha_1, \beta_1)$ and $x = (x_1, \sqrt{1-x_1^2})$ where $\alpha_1, \beta_1$ are two independent random standard Gaussians.

Throughout the proof, we also take an alternative view of the randomness. We write $\langle w_0, x \rangle = \alpha$ and $\alpha_1 = \alpha x_1 + \sqrt{1-x_1^2}\beta$ for two independent $\alpha, \beta \sim \mathcal{N}(0,1)$.[20]

We first make a technical claim involving in fitting monomials in $x_1$. Its proof is in Section C.1.2.

**Claim C.1.** *Recall $h_i(x)$ is the degree-$i$ Hermite polynomial (see Definition A.5). For every integer $i \geq 1$ there exists constant $p'_i$ with $|p'_i| \geq \frac{(i-1)!!}{200i^2}$ such that*

*for even $i$ :* $\quad x_1^i = \frac{1}{p'_i} \mathbb{E}_{w_0 \sim \mathcal{N}(0,\mathbf{I}), b_0 \sim \mathcal{N}(0,1)} \left[ h_i(\alpha_1) \cdot \mathbb{I}[0 < -b_0 \leq 1/(2i)] \cdot \mathbb{I}[\langle x, w_0 \rangle + b_0 \geq 0] \right]$

*for odd $i$ :* $\quad x_1^i = \frac{1}{p'_i} \mathbb{E}_{w_0 \sim \mathcal{N}(0,\mathbf{I}), b_0 \sim \mathcal{N}(0,1)} \left[ h_i(\alpha_1) \cdot \mathbb{I}[|b_0| \leq 1/(2i)] \cdot \mathbb{I}[\langle x, w_0 \rangle + b_0 \geq 0] \right]$

We next use Claim C.1 to fit arbitrary functions $\phi(x_1)$. By Taylor expansion, we have

$$\phi(x_1) = c_0 + \sum_{i=1,\text{ odd } i}^{\infty} c_i x_1^i + \sum_{i=2,\text{ even } i}^{\infty} c_i x_1^i$$

$$= c_0 + \sum_{i=1}^{\infty} c'_i \cdot \mathbb{E}_{\alpha,\beta,b_0 \sim \mathcal{N}(0,1)} \left[ h_i(\alpha_1) \cdot \mathbb{I}[q_i(b_0)] \cdot \mathbb{I}[\langle x, w_0 \rangle + b_0 \geq 0] \right]$$

where

$$c'_i := \frac{c_i}{p'_i} \ , \quad |c'_i| \leq \frac{200i^2 |c_i|}{(i-1)!!} \quad \text{and} \quad q_i(b_0) = \begin{cases} |b_0| \leq 1/(2i), & i \text{ is odd}; \\ 0 < -b_0 \leq 1/(2i), & i \text{ is even}. \end{cases} \tag{C.1}$$

The next technical claim carefully bounds the absolute values of the Hermite polynomials. Its proof is in Section C.1.2.

**Claim C.2.** *Setting $B_i := 100i^{1/2} + 10\sqrt{\log \frac{1}{\varepsilon}}$, we have*

(a) $\sum_{i=1}^{\infty} |c'_i| \cdot \mathbb{E}_{z \sim \mathcal{N}(0,1)} \left[ |h_i(z)| \cdot \mathbb{I}[|z| \geq B_i] \right] \leq \epsilon/8$

(b) $\sum_{i=1}^{\infty} |c'_i| \cdot \mathbb{E}_{z \sim \mathcal{N}(0,1)} \left[ |h_i(B_i)| \cdot \mathbb{I}[|z| \geq B_i] \right] \leq \epsilon/8$

(c) $\sum_{i=1}^{\infty} |c'_i| \cdot \mathbb{E}_{z \sim \mathcal{N}(0,1)} \left[ |h_i(z)| \cdot \mathbb{I}[|z| \leq B_i] \right] \leq \frac{1}{2}\mathfrak{C}_\varepsilon(\phi,1)$

(d) $\sum_{i=1}^{\infty} |c'_i| \cdot \mathbb{E}_{z \sim \mathcal{N}(0,1)} \left[ \left| \frac{d}{dz} h_i(z) \right| \cdot \mathbb{I}[|z| \leq B_i] \right] \leq \frac{1}{2}\mathfrak{C}_\varepsilon(\phi,1)$

Now, let us define $\widehat{h}_i(\alpha_1) := h_i(\alpha_1) \cdot \mathbb{I}[|\alpha_1| \leq B_i] + h_i(\text{sign}(\alpha_1)B_i) \cdot \mathbb{I}[|\alpha_1| > B_i]$ as the truncated version of the Hermite polynomial $h_i(\cdot)$.

Using Claim C.2, we have

$$\phi(x_1) = c_0 + R'(x_1) + \sum_{i=1}^{\infty} c'_i \cdot \mathbb{E}_{\alpha,\beta,b_0 \sim \mathcal{N}(0,1)} \left[ \widehat{h}_i(\alpha_1) \cdot \mathbb{I}[q_i(b_0)] \cdot \mathbb{I}[\langle x, w_0 \rangle + b_0 \geq 0] \right]$$

where $|R'(x_1)| < \epsilon/4$ uses Claim C.2a and Claim C.2b. In other words, if we define

$$h(\alpha_1, b_0) := 2c_0 + \sum_{i=1}^{\infty} c_i' \cdot \widehat{h}_i(\alpha_1) \cdot \mathbb{I}[q_i(b_0)]$$

then we have

$$\left| \mathbb{E}_{\alpha,\beta,b_0 \sim \mathcal{N}(0,1)} \left[ \mathbb{I}[\langle x, w_0 \rangle + b_0 \geq 0] \cdot h(\alpha_1, b_0) \right] - \phi(x_1) \right| = |R'(x_1)| \leq \varepsilon/4 \ .$$

As for the range of $h$, we use Claim C.2b and Claim C.2c to derive that

$$|h(\alpha_1, b_0)| \leq 2c_0 + \frac{\varepsilon}{8} + \frac{1}{2}\mathfrak{C}_\varepsilon(\phi, 1) \leq \mathfrak{C}_\varepsilon(\phi, 1) \ .$$

As for the Lipschitz continuity of $h$ on its first coordinate $\alpha_1$, we observe that for each $i > 0$, $\widehat{h}_i(z)$ has zero sub-gradient for all $|z| \geq B_i$. Therefore, it suffices to bound $\left| \frac{d}{dz} h_i(z) \right|$ for $|z| < B_i$. Replacing the use of Claim C.2c by Claim C.2d immediately give us the same bound on the Lipschitz continuity of $h$ with respect to $\alpha_1$.

As for the expected square $\mathbb{E}_{\alpha_1, b_0 \sim \mathcal{N}(0,1)} \left[ h(\alpha_1, b_0)^2 \right]$, we can write

$$h(\alpha_1, b_0) = 2c_0 + \sum_{i=1}^{\infty} c_i' \cdot \widehat{h}_i(\alpha_1) \cdot \mathbb{I}[q_i(b_0)] \overset{\textcircled{1}}{=} 2c_0 + \sum_{i=1}^{\infty} c_i' \cdot h_i(\alpha_1) \cdot \mathbb{I}[q_i(b_0)] \pm \frac{\varepsilon}{4}$$

Above, ① uses Claim C.2a and Claim C.2b. Using the othogonality condition of Hermite polynomials (that is, $\mathbb{E}_{x \sim \mathcal{N}(0,1)}[h_i(x)h_j(x)] = \sqrt{2\pi}j!\delta_{i,j}$ from Definition A.5), we immediately have

$$\mathbb{E}_{\alpha_1, b_0 \sim \mathcal{N}(0,1)}[h(\alpha_1, b_0)^2] \leq O(\varepsilon^2 + c_0^2) + O(1) \cdot \sum_{i=1}^{\infty} (c_i')^2 (i!) \cdot \mathbb{E}_{b_0}[\mathbb{I}[q_i(b_0)]]$$

$$\leq O(\varepsilon^2 + c_0^2) + O(1) \cdot \sum_{i=1}^{\infty} \frac{(c_i')^2 (i!)}{i}$$

$$\leq O(\varepsilon^2 + c_0^2) + O(1) \cdot \sum_{i=1}^{\infty} \frac{(i!) \cdot i^3 \cdot |c_i|^2}{((i-1)!!)^2}$$

$$\leq O(\varepsilon^2 + c_0^2) + O(1) \cdot \sum_{i=1}^{\infty} i^{3.5} \cdot |c_i|^2 \leq (\mathfrak{C}_{\mathsf{s}}(\phi, 1))^2 \ .$$

Above, ① uses inequality $\frac{i!}{((i-1)!!)^2} \leq 2\sqrt{i}$ for all $i \geq 1$.

This finishes the proof of Lemma 6.3a. ∎

### C.1.2 Proofs of Claim C.1 and Claim C.2

*Proof of Claim C.1.* We treat the two cases separately.

**Even $i$.** By Lemma A.6, we know that

$$\mathbb{E}_{w_0 \sim \mathcal{N}(0,\mathbf{I}), b_0 \sim \mathcal{N}(0,1)} \left[ h_i(\alpha_1) \cdot \mathbb{I}[0 < -b_0 \leq 1/(2i)] \cdot \mathbb{I}[\langle x, w_0 \rangle + b_0 \geq 0]] \right.$$

$$= \mathbb{E}_{b_0 \sim \mathcal{N}(0,1)} \left[ \mathbb{E}_{\alpha,\beta \sim \mathcal{N}(0,1)} \left[ h_i \left( \alpha x_1 + \beta \sqrt{1 - x_1^2} \right) \cdot \mathbb{I}[\alpha \geq -b_0] \right] \cdot \mathbb{I}[0 < -b_0 \leq 1/(2i)] \right]$$

$$= \mathbb{E}_{b_0 \sim \mathcal{N}(0,1)} \left[ p_i \cdot \mathbb{I}[0 < -b_0 \leq 1/(2i)]] \times x_1^i$$

where

$$p_i = (i-1)!! \frac{\exp(-b_0^2/2)}{\sqrt{2\pi}} \sum_{r=1, r \text{ odd}}^{i-1} \frac{(-1)^{\frac{i-1-r}{2}}}{r!!} \binom{i/2 - 1}{(r-1)/2} (-b_0)^r \ . \tag{C.2}$$

We try to bound the coefficient "$\mathbb{E}_{b_0 \sim \mathcal{N}(0,1)} [p_i \cdot \mathbb{I}[0 < -b_0 \leq 1/(2i)]]$" as follows. Define $c_r$ as:

$$c_r := \frac{(-1)^{\frac{i-1-r}{2}}}{r!!} \binom{i/2 - 1}{(r-1)/2}.$$

Then, for $0 \leq -b_0 \leq \frac{1}{2i}$, we know that for all $1 < r \leq i - 1$, $r$ odd:

$$|c_r(-b_0)^r| \leq \frac{1}{4}|c_{r-2}(-b_0)^{r-2}|,$$

which implies

$$\left| \sum_{r=1, r \text{ odd}}^{i-1} c_r(-b_0)^r \right| \geq \frac{2}{3}|c_1 b_0| = \frac{2}{3}|b_0|$$

and

$$\text{sign}\left( \sum_{r=1, r \text{ odd}}^{i-1} c_r(-b_0)^r \right) = \text{sign}(c_1)$$

is independent of the randomness of $b_0$. Therefore, using the formula of $p_i$ in (C.2):

$$\left| \mathbb{E}_{b_0 \sim \mathcal{N}(0,1)}[p_i \cdot \mathbb{I}[0 \leq -b_0 \leq 1/(2i)]] \right|$$

$$= \left| \mathbb{E}_{b_0 \sim \mathcal{N}(0,1)} \left[ (i-1)!! \frac{\exp(-b_0^2/2)}{\sqrt{2\pi}} \sum_{r=1, r \text{ odd}}^{i-1} c_r(-b_0)^r \cdot \mathbb{I}[0 \leq -b_0 \leq 1/(2i)] \right] \right|$$

$$\geq \mathbb{E}_{b_0 \sim \mathcal{N}(0,1)} \left[ (i-1)!! \frac{\exp(-b_0^2/2)}{\sqrt{2\pi}} \frac{2}{3}|b_0| \cdot \mathbb{I}[0 \leq -b_0 \leq 1/(2i)] \right]$$

$$\geq \frac{(i-1)!!}{100 i^2}.$$

**Odd $i$.** Similarly, by Lemma A.6, we have

$$\mathbb{E}_{w_0 \sim \mathcal{N}(0, \mathbf{I}), b_0 \sim \mathcal{N}(0,1)} \left[ h_i(\alpha_1) \cdot \mathbb{I}[|b_0| \leq 1/(2i)] \cdot \mathbb{I}[\langle x, w_0 \rangle + b_0 \geq 0] \right]$$

$$= \mathbb{E}_{b_0 \sim \mathcal{N}(0,1)} \left[ p_i \cdot \mathbb{I}[|b_0| \leq 1/(2i)] \right] \times x_1^i$$

where

$$p_i = (i-1)!! \frac{\exp(-b_0^2/2)}{\sqrt{2\pi}} \sum_{r=0, r \text{ even}}^{i-1} \frac{(-1)^{\frac{i-1-r}{2}}}{r!!} \binom{i/2 - 1}{(r-1)/2} (-b_0)^r . \tag{C.3}$$

This time we bound the coefficient "$\mathbb{E}_{b_0 \sim \mathcal{N}(0,1)}[p_i \cdot \mathbb{I}[|b_0| \leq 1/(2i)]]$" as follows. Define $c_r$ as:

$$c_r := \frac{(-1)^{\frac{i-1-r}{2}}}{r!!} \binom{i/2 - 1}{(r-1)/2}.$$

Then, for $|b_0| \leq \frac{1}{2i}$, we know that for all even $r$ in $1 < r \leq i-1$ it satisfies

$$|c_r(-b_0)^r| \leq \frac{1}{4}|c_{r-2}(-b_0)^{r-2}|,$$

which implies

$$\left| \sum_{r=0, r \text{ even}}^{i-1} c_r(-b_0)^r \right| \geq \frac{2}{3}|c_0| = \frac{2}{3}\left| \binom{i/2 - 1}{-1/2} \right| > \frac{1}{2i}.$$

and

$$\text{sign}\left( \sum_{r=1, r \text{ odd}}^{i-1} c_r(-b_0)^r \right) = \text{sign}(c_0)$$

is independent of the randomness of $b_0$. Therefore, using the formula of $p_i$ in (C.3):

$$\left| \mathbb{E}_{b_0 \sim \mathcal{N}(0,1)}[p_i \cdot \mathbb{I}[|b_0| \leq 1/(2i)]] \right|$$

$$= \left| \mathbb{E}_{b_0 \sim \mathcal{N}(0,1)} \left[ (i-1)!! \frac{\exp(-b_0^2/2)}{\sqrt{2\pi}} \sum_{r=1, r \text{ odd}}^{i-1} c_r(-b_0)^r \cdot \mathbb{I}[|b_0| \leq 1/(2i)] \right] \right|$$

$$\geq \mathbb{E}_{b_0 \sim \mathcal{N}(0,1)} \left[ (i-1)!! \frac{\exp(-b_0^2/2)}{\sqrt{2\pi}} \frac{1}{2i} \cdot \mathbb{I}[|b_0| \leq 1/(2i)] \right]$$

$$\geq \frac{(i-1)!!}{100 i^2}.$$

$\square$

*Proof of Claim C.2.* By the definition of Hermite polynomial (see Definition A.5), we have that

$$|h_i(x)| \le \sum_{j=0}^{\lfloor i/2 \rfloor} \frac{i!|x|^{i-2j}}{j!(i-2j)!2^j} \le \sum_{j=0}^{\lfloor i/2 \rfloor} \frac{|x|^{i-2j}i^{2j}}{j!} \tag{C.4}$$

Using our bound on $c_i'$ (see (C.1)), we have

$$|c_i' h_i(z)| \le O(1)|c_i| \frac{i^4}{i!!} \sum_{j=0}^{\lfloor i/2 \rfloor} \frac{|z|^{i-2j}i^{2j}}{j!} \quad . \tag{C.5}$$

(a) Denote by $b = B_i = 100i^{1/2}\theta$ for notational simplicity (for some parameter $\theta \ge 1$ that we shall choose later). We have

$$|c_i'| \cdot \mathbb{E}_{z \sim \mathcal{N}(0,1)}[|h_i(z)| \cdot \mathbb{I}[|z| \ge b]] \le 2|c_i'| \cdot \mathbb{E}_{z \sim \mathcal{N}(0,1)} \left[ \sum_{j=0}^{\lfloor i/2 \rfloor} \frac{|z|^{i-2j}i^{2j}\mathbb{I}[z \ge b]}{j!} \right]$$

$$= O(1)|c_i| \frac{i^4}{i!!} \sum_{j=0}^{\lfloor i/2 \rfloor} \frac{L_{i-2j,b} \cdot i^{2j}}{j!} \quad , \tag{C.6}$$

where recall from Lemma A.7 that

$$L_{i,b} \le O(1)e^{-b^2/2} \cdot \sum_{j=0}^{i-1} \frac{(i-1)!!}{j!!}b^j$$

$$\le O(1)e^{-b^2/2} \cdot \sum_{j=0}^{i-1} \frac{(i-1)!!}{j!!} \left( 100i^{1/2}\theta \right)^j$$

$$\le O(1)e^{-b^2/2} (100\theta)^i \cdot (i-1)!! \cdot \sum_{j=0}^{i-1} \frac{i^{j/2}}{j!!}$$

$$\le O(1)e^{-b^2/2} (200\theta)^i \cdot (i-1)!! \qquad\qquad \left(\text{using } \sum_{j=0}^{i-1} \frac{i^{j/2}}{j!!} \le 2^i\right)$$

Thus we have

$$\sum_{j=0}^{\lfloor i/2 \rfloor} \frac{L_{i-2j,b} \cdot i^{2j}}{j!} \le O(1) (200\theta)^i e^{-b^2/2} \sum_{j=0}^{\lfloor i/2 \rfloor} \frac{(i+1-2j)!! \cdot i^{2j}}{j!}$$

$$\le O(1) (400\theta)^i e^{-b^2/2} \sum_{j=0}^{\lfloor i/2 \rfloor} \frac{i^{(i-2j)/2} \cdot i^{2j}}{j!}$$

$$= O(1) (400\theta)^i e^{-b^2/2} i^{i/2} \sum_{j=0}^{\lfloor i/2 \rfloor} \frac{i^j}{j!}$$

$$\overset{①}{\le} O(1) (1200\theta)^i e^{-b^2/2} i^{i/2} \tag{C.7}$$

$$\overset{②}{\le} O(i^{i/2}) \cdot 1200^i \cdot \left( \theta \cdot e^{-10^4\theta^2} \right)^i$$

$$\overset{③}{\le} i^{i/2} \cdot \frac{1}{100^i} \cdot \varepsilon^2$$

$$\overset{④}{\le} i^{i/2} \cdot \frac{1}{100^i} \cdot \frac{\varepsilon}{|c_i|} \tag{C.8}$$

Above, inequality ① uses $\sum_{j=0}^{i-1} \frac{i^j}{j!} \le 3^i$; inequality ② uses our definition of $b = B_i$; inequality ③ uses $(\theta \cdot e^{-10^4\theta^2})^i \le \frac{\varepsilon^2}{100000^i}$ for $\theta = 1 + \frac{\sqrt{\log(1/\varepsilon)}}{10\sqrt{i}}$; and inequality ④ uses $\varepsilon|c_i| \le 1$. Putting this back to (C.6), we have

$$\sum_{i=1}^{\infty} |c_i'| \cdot \mathbb{E}_{z \sim \mathcal{N}(0,1)}[|h_i(z)| \cdot \mathbb{I}[|z| \ge b]] \le O(1) \sum_{i=1}^{\infty} \frac{i^4}{i!!} \frac{\varepsilon}{100^i} i^{i/2} \le \frac{\varepsilon}{8} \quad .$$

Above, in the last inequality we have used $\frac{i^4}{i!!}i^{i/2} \le 40 \cdot 4^i$ for $i \ge 1$.

(b) Similar to the previous case, we calculate that

$$|c_i'| \cdot \mathbb{E}_{z\sim\mathcal{N}(0,1)}[|h_i(b)| \cdot \mathbb{I}[|z| \ge b]]| \le |c_i'| \cdot \mathbb{E}_{z\sim\mathcal{N}(0,1)}\left[\sum_{j=0}^{\lfloor i/2\rfloor} \frac{b^{i-2j}i^{2j}}{j!}\mathbb{I}[|z| \ge b]\right]$$

$$\le O(1)|c_i|\frac{i^4}{i!!} \cdot L_{0,b}b^i \sum_{j=0}^{\lfloor i/2\rfloor} \frac{b^{-2j}\cdot i^{2j}}{j!}$$

$$\overset{①}{\le} O(1)|c_i|\frac{i^4}{i!!} \cdot e^{-b^2/2}b^i \sum_{j=0}^{\lfloor i/2\rfloor} \frac{i^j}{j!}$$

$$\overset{②}{\le} O(1)|c_i|\frac{i^4}{i!!} \cdot e^{-b^2/2}(3b)^i$$

$$\le |c_i|\frac{i^4}{i!!} \cdot e^{-b^2/2}(3b)^i \quad.$$

Above, inequality ① uses $b^{2j} = B_i^{2j} \ge (10i)^j$; and inequality ② uses again $\sum_{j=0}^{i-1}\frac{i^j}{j!} \le 3^i$. Using this and continue from (C.7) of the previous case, we finish the proof.

(c) Again denote by $b = B_i = 100i^{1/2}\theta$ for notational simplicity. By Eq. (C.5), it holds that

$$\sum_{i=1}^{\infty}|c_i'| \cdot \mathbb{E}_{z\sim\mathcal{N}(0,1)}\big[|h_i(z)| \cdot \mathbb{I}[|z| \le B_i]\big] \le O(1)\sum_{i=1}^{\infty}|c_i|\frac{i^4}{i!!}\sum_{j=0}^{\lfloor i/2\rfloor}\frac{B_i^{i-2j}i^{2j}}{j!}$$

$$\le O(1)\sum_{i=1}^{\infty}|c_i|\frac{i^4}{i!!}\sum_{j=0}^{\lfloor i/2\rfloor}\left(100i^{1/2}\theta\right)^{i-2j}\frac{i^{2j}}{j!}$$

$$\le O(1)\sum_{i=1}^{\infty}|c_i|\frac{i^4}{i!!}\left(100i^{1/2}\theta\right)^i\sum_{j=0}^{\lfloor i/2\rfloor}\frac{i^j}{j!}$$

$$\overset{①}{\le}\sum_{i=1}^{\infty}|c_i|\left(O(1)\theta\right)^i$$

$$\overset{②}{\le}\sum_{i=1}^{\infty}|c_i|\left(O(1)\big(1+\frac{\sqrt{\log(1/\varepsilon)}}{10\sqrt{i}}\big)\right)^i$$

$$\le \frac{1}{2}\mathfrak{C}_\varepsilon(\phi,1) \quad.$$

Here, in ① we use the fact that $\frac{i^j}{j!} \le 10^i$; in ② we use the fact that $\left(\frac{a}{b}\right)^b \le e^a$ for all $b \in [1,a]$.

(d) By the definition of Hermite polynomial (see Definition A.5), we can also bound

$$\left|\frac{d}{dx}h_i(x)\right| \le \sum_{j=0}^{\lfloor i/2\rfloor}|x|^{i-2j}i^{2j}$$

which is the same upper bound comparing to (C.4). Therefore, the same proof of Claim C.2c also applies to $\left|\frac{d}{dx}h_i(x)\right|$. $\qquad\square$

### C.1.3 Lemma 6.4: Information out of Randomness

Let us consider a single neural of the second layer at random initialization, given as:

$$n_1(x) = \sum_{i\in[m_1]} v_{1,i}^{(0)}\sigma\left(\langle w_i^{(0)}, x\rangle + b_{1,i}^{(0)}\right)$$

**Lemma 6.4** (information out of randomness). *For every smooth function $\phi$, every $w^* \in \mathbb{R}^d$ with $\|w^*\|_2 = 1$, for every $\varepsilon \in \left(0, \frac{1}{\mathfrak{C}_{\mathfrak{s}}(\phi, 1)}\right)$, there exists real-valued functions*

$$\rho(v_1^{(0)}, W^{(0)}, b_1^{(0)}), \; B(x, v_1^{(0)}, W^{(0)}, b_1^{(0)}), \; R(x, v_1^{(0)}, W^{(0)}, b_1^{(0)}), \; and \; \phi_\varepsilon(x)$$

*such that for every $x$:*

$$n_1(x) = \rho\left(v_1^{(0)}, W^{(0)}, b_1^{(0)}\right)\phi_\varepsilon(x) + B\left(x, v_1^{(0)}, W^{(0)}, b_1^{(0)}\right) + R\left(x, v_1^{(0)}, W^{(0)}, b_1^{(0)}\right).$$

*Moreover, letting $C = \mathfrak{C}_\varepsilon(\phi, 1)$ be the complexity of $\phi$, and if $v_{1,i}^{(0)} \sim \mathcal{N}(0, \frac{1}{m_2})$ and $w_{i,j}^{(0)}, b_{1,i}^{(0)} \sim \mathcal{N}(0, \frac{1}{m_1})$ are at random initialization, then we have*

1. *For every fixed $x$, $\rho\left(v_1^{(0)}, W^{(0)}, b_1^{(0)}\right)$ is independent of $B\left(x, v_1^{(0)}, W^{(0)}, b_1^{(0)}\right)$.*

2. $\rho\left(v_1^{(0)}, W^{(0)}, b_1^{(0)}\right) \sim \mathcal{N}\left(0, \frac{1}{100C^2 m_2}\right)$.

3. *For every $x$ with $\|x\|_2 = 1$, $|\phi_\varepsilon(x) - \phi(\langle w^*, x\rangle)| \leq \varepsilon$.*

4. *For every fixed $x$ with $\|x\|_2 = 1$, with high probability $\left|R\left(x, v_1^{(0)}, W^{(0)}, b_1^{(0)}\right)\right| \leq \widetilde{O}\left(\frac{1}{\sqrt{m_1 m_2}}\right)$ and $\left|B\left(x, v_1^{(0)}, W^{(0)}, b_1^{(0)}\right)\right| \leq \widetilde{O}\left(\frac{1}{\sqrt{m_2}}\right)$.*

*Furthermore, there exists real-valued function $\widetilde{\rho}(v_1^{(0)})$ satisfying with high probability:*

$$\widetilde{\rho}(v_1^{(0)}) \sim \mathcal{N}\left(0, \frac{1}{100C^2 m_2}\right) \quad and \quad \mathcal{W}_2(\rho|_{W^{(0)}, b_1^{(0)}}, \widetilde{\rho}) \leq \widetilde{O}\left(\frac{1}{C\sqrt{m_1 m_2}}\right) \; .$$

Before going to proofs, we recall from Section A that we have overridden the notion of "with high probability" so the above statement implies $\mathcal{W}_2(R, 0) \leq O\left(\frac{1}{\sqrt{m_1 m_2}}\right)$ and $\mathcal{W}_2(B, 0) \leq \widetilde{O}\left(\frac{1}{\sqrt{m_2}}\right)$.

*Proof of Lemma 6.4.* Without loss of generality we assume $w^* = e_1$. Recall that

$$n_1(x) = \sum_{i \in [m_1]} v_{1,i}^{(0)} \sigma\left(\langle w_i^{(0)}, x\rangle + b_{1,i}^{(0)}\right)$$

By Lemma 6.3, for every $\varepsilon > 0$, there exists a function $h$ such that for every unit $x$ with $x_d = \frac{1}{2}$ and every $i \in [m_1]$:

$$\mathbb{E}_{w_i^{(0)} \sim \mathcal{N}(0, \frac{1}{m_1}), b_{1,i}^{(0)} \sim \mathcal{N}(0, \frac{1}{m_1})}\left[h\left(w_{i,1}^{(0)}, b_{1,i}^{(0)}\right) x_d \mathbb{I}[\langle w_i^{(0)}, x\rangle + b_{1,i}^{(0)} \geq 0]\right] = \frac{\phi_\varepsilon(x_1)}{C} = \frac{\phi_\varepsilon(\langle w^*, x\rangle)}{C} \tag{C.9}$$

with

$$|\phi_\varepsilon(\langle w^*, x\rangle) - \phi(\langle w^*, x\rangle)| \leq \varepsilon$$

and $\left|h\left(w_{i,1}^{(0)}, b_{1,i}^{(0)}\right)\right| \in [0, 1]$. (Here to apply Lemma 6.3, we have re-scaled $h$ in Lemma 6.3 by $\frac{1}{C}$ and re-scaled $\alpha_1, \beta_1, b_0$ in Lemma 6.3 by $\frac{1}{\sqrt{m_1}}$.)

Throughout the proof, we fix some parameter $\tau$ (that we shall in the end choose $\tau = \frac{1}{100}$). Let us construct the sign function $\mathfrak{s}: [-1, 1] \times \mathbb{R} \to \{-1, 0, 1\}$ and the set function $I: [-1, 1] \ni y \mapsto I(y) \subset \mathbb{R}$ given in Lemma A.4. Now, for every $w_{i,1}^{(0)}$, define

$$I_i := I\left(h\left(w_{i,1}^{(0)}, b_{1,i}^{(0)}\right)\right) \subset [-2, 2]$$

Also define set

$$\mathcal{S} := \left\{i \in [m_1]: \sqrt{m_2} v_{1,i}^{(0)} \in I_i\right\} \; .$$

We define what we call "*effective sign* of $v_{1,i}^{(0)}$" to be

$$s_i := \mathfrak{s}\left(h\left(w_{i,1}^{(0)}, b_{1,i}^{(0)}\right), \sqrt{m_2} v_{1,i}^{(0)}\right)$$

By the definition of $I_i$ and $\mathfrak{s}$ (see Lemma A.4), we claim that $\mathcal{S}$ and the "effective sign" of those $v_{1,i}^{(0)}$ in this set $i \in \mathcal{S}$ are independent of $W^{(0)}$. Indeed, for any fixed choice of $W^{(0)}$, each $i \in [m_1]$ is in set $\mathcal{S}$ with probability $\tau$, and for each $i \in \mathcal{S}$, $s_i$ is $\pm 1$ each with half probability. In other words, the following unit vector $u \in \mathbb{R}^{m_1}$ is independent of $W^{(0)}$:

$$u_i := \begin{cases} \frac{s_i}{\sqrt{|\mathcal{S}|}} & \text{if } i \in \mathcal{S}; \\ 0 & \text{if } i \notin \mathcal{S}. \end{cases}$$

Since each $i \in \mathcal{S}$ with probability $\tau$, we also know with high probability:

$$|\mathcal{S}| = \tau m_1 \pm O(\sqrt{\tau m_1}) \ . \tag{C.10}$$

Conversely, conditioning on $\mathcal{S} = \mathcal{S}_0$ and $\{s_i\}_{i \in \mathcal{S}} = s$ being fixed (or equivalently on $u$ being fixed), the distribution of $W^{(0)}$ is also unchanged. Since the entries of $W^{(0)}$ are i.i.d. generated from $\mathcal{N}(0, 1/m_1)$, we can write $W^{(0)} \in \mathbb{R}^{m_1 \times d}$ as

$$W^{(0)} = \alpha u e_d^\top + \beta$$

where $\alpha := u^\top W^{(0)} e_d \sim \mathcal{N}\left(0, \frac{1}{m_1}\right)$ and $\beta \in \mathbb{R}^{m_1 \times d}$ are two independent random variables given $u$ (the entries of $\beta$ are not i.i.d.) This factorizes out the randomness of the last column of $W^{(0)}$ along the direction $u$, and in particular,

- $\alpha$ is independent of $u$.

A simple observation here is that, although $\alpha \sim \mathcal{N}\left(0, \frac{1}{m_1}\right)$, if we fix $W^{(0)}$ (or $b_1^{(0)}$) then the distribution of $\alpha$ is not Gaussian. Fortunately, fixing $W^{(0)}$ and $b_1^{(0)}$, we still have that the coordinates of $s = (s_1, \dots, s_{m_1})$ are i.i.d. (each $s_i$ is zero with probability $1 - \tau$, and $s_i$ is $\pm 1$ each with probability $\tau/2$).

Let $\alpha|_{W^{(0)}, b_1^{(0)}}$ denote the conditional distribution of $\alpha$. With high probability over $W^{(0)}$, $\|W^{(0)} e_d\|_\infty \leq \widetilde{O}\left(\frac{1}{\sqrt{m_1}}\right)$. Fixing the support of $u$ to be $\mathcal{S}$, we know that for every $i \in \mathcal{S}$, $u_i$ is i.i.d. $\pm \frac{1}{\sqrt{|\mathcal{S}|}}$. This implies that fixing $W^{(0)}, b_1^{(0)}, \mathcal{S}$, the quantity

$$\alpha = u^\top W^{(0)} e_d = \sum_{i \in \mathcal{S}} u_i [W^{(0)} e_d]_i \tag{C.11}$$

is a sum of $|\mathcal{S}|$ many independent, mean zero random variables with each $|u_i [W^{(0)} e_d]_i| \leq \widetilde{O}\left(\frac{1}{\sqrt{m_1 |\mathcal{S}|}}\right)$ and $\sum_{i \in \mathcal{S}} u_i [W^{(0)} e_d]_i^2 = \frac{1}{m_1} \pm \widetilde{O}\left(\frac{1}{\sqrt{\tau} m_1^{3/2}}\right)$ w.h.p. Applying any Wasserstein distance bound of central limit theorem (see Lemma A.3), we know that there exists some random Gaussian $g \sim \mathcal{N}(0, \frac{1}{m_1})$ that is independent of $W_0$ or $b_1^{(0)}$ such that w.h.p.

$$\mathcal{W}_2(\alpha|_{W^{(0)}, b_1^{(0)}}, g) \leq \widetilde{O}\left(\frac{1}{\sqrt{\tau} m_1}\right) \ . \tag{C.12}$$

We can write

$$n_1(x) = \sum_{i \in [m_1]} v_{1,i}^{(0)} \sigma\left(\langle w_i^{(0)}, x \rangle + b_{1,i}^{(0)}\right)$$

$$= \underbrace{\sum_{i \notin \mathcal{S}} v_{1,i}^{(0)} \sigma\left(\langle w_i^{(0)}, x \rangle + b_{1,i}^{(0)}\right)}_{:= B_1\left(x, v_1^{(0)}, W^{(0)}, b_1^{(0)}\right)} + \sum_{i \in \mathcal{S}} v_{1,i}^{(0)} \sigma\left(\langle w_i^{(0)}, x \rangle + b_{1,i}^{(0)}\right),$$

By definition of $B_1$, conditioning on the randomness of $u$, we know that $B_1$ is independent of $\alpha$ — because $\langle w_i^{(0)}, x \rangle + b_{1,i}^{(0)} = \langle \beta_i, x \rangle + b_{1,i}^{(0)}$ for $i \notin \mathcal{S}$. Since $u$ and $\alpha$ are independent, we know that $\alpha$ and $B_1$ are independent by Proposition A.2. We continue to write

$$n_1(x) - B_1$$
$$= \sum_{i \in \mathcal{S}} v_{1,i}^{(0)} \sigma\left(\langle w_i^{(0)}, x \rangle + b_{1,i}^{(0)}\right)$$

$$= \sum_{i \in \mathcal{S}} v_{1,i}^{(0)} \mathbb{I}[\langle w_i^{(0)}, x \rangle + b_{1,i}^{(0)} \geq 0] \left( \frac{\alpha s_i}{\sqrt{|\mathcal{S}|}} x_d + \langle \beta_i, x \rangle + b_{1,i}^{(0)} \right)$$

$$= \underbrace{\sum_{i \in \mathcal{S}} v_{1,i}^{(0)} \mathbb{I}[\langle w_i^{(0)}, x \rangle + b_{1,i}^{(0)} \geq 0] \frac{\alpha s_i}{\sqrt{|\mathcal{S}|}} x_d}_{:=T_3} + \underbrace{\sum_{i \in \mathcal{S}} v_{1,i}^{(0)} \mathbb{I}[\langle w_i^{(0)}, x \rangle + b_{1,i}^{(0)} \geq 0] \left( \langle \beta_i, x \rangle + b_{1,i}^{(0)} \right)}_{:=T_4}$$

**First consider $T_3$.** For each $i \in \mathcal{S}$ we have

$$\left| s_i \cdot v_{1,i}^{(0)} - \frac{1}{\sqrt{m_2}} h\left( w_{i,1}^{(0)}, b_{1,i}^{(0)} \right) \right| \leq O\left( \frac{1}{\sqrt{m_2}} \right) \quad \text{and}$$

$$\mathbb{E}_{v_{1,i}^{(0)}} \left[ s_i \cdot v_{1,i}^{(0)} - \frac{1}{\sqrt{m_2}} h\left( w_{i,1}^{(0)}, b_{1,i}^{(0)} \right) \mid i \in \mathcal{S} \right] = 0 .$$

Above, the first row is because $\sqrt{m_2} v_{1,i}^{(0)} \in I_i$ and the Bounded-ness property of the interval (see Lemma A.4); and the second row by the Unbiased property of the interval (see Lemma A.4). By concentration, for fixed vector $x$, with high probability over the randomness of $V^{(0)}$:

$$\left| \sum_{i \in \mathcal{S}} \left( v_{1,i}^{(0)} s_i - \frac{1}{\sqrt{m_2}} h\left( w_{i,1}^{(0)}, b_{1,i}^{(0)} \right) \right) \mathbb{I}[\langle w_i^{(0)}, x \rangle + b_{1,i}^{(0)} \geq 0] \right| \leq \widetilde{O}\left( \frac{\sqrt{|\mathcal{S}|}}{\sqrt{m_2}} \right) .$$

In other words,

$$T_3 = \sum_{i \in \mathcal{S}} \frac{\alpha}{\sqrt{m_2 |\mathcal{S}|}} h\left( w_{i,1}^{(0)}, b_{1,i}^{(0)} \right) x_d \mathbb{I}[\langle w_i^{(0)}, x \rangle + b_{1,i}^{(0)} \geq 0] + R_1$$

where $R_1 = R_1\left( x, v_1^{(0)}, W^{(0)}, b_1^{(0)} \right)$ satisfies $|R_1| \leq \widetilde{O}\left( \frac{|\alpha|}{\sqrt{|\mathcal{S}|}} \cdot \frac{\sqrt{|\mathcal{S}|}}{\sqrt{m_2}} \right) \leq \widetilde{O}\left( \frac{1}{\sqrt{m_1 m_2}} \right)$. We write

$$T_5 := \frac{T_3 - R_1}{\alpha} = \sum_{i \in \mathcal{S}} \frac{1}{\sqrt{m_2 |\mathcal{S}|}} h\left( w_{i,1}^{(0)}, b_{1,i}^{(0)} \right) x_d \mathbb{I}[\langle w_i^{(0)}, x \rangle + b_{1,i}^{(0)} \geq 0] .$$

By (C.9) (i.e., the property of $h$), we know that for every fixed $x$, using concentration bound, with high probability over $W^{(0)}$ and $b^{(0)}$:

$$\left| T_5 - \frac{\sqrt{|\mathcal{S}|}}{\sqrt{m_2} C} \phi_\varepsilon(\langle w^*, x \rangle) \right| \leq \widetilde{O}\left( \frac{1}{\sqrt{m_2}} \right) .$$

and thus by (C.10)

$$\left| \frac{C\sqrt{m_2}}{\sqrt{\tau m_1}} T_5 - \phi_\varepsilon(\langle w^*, x \rangle) \right| \leq \widetilde{O}\left( \frac{C}{\sqrt{\tau m_1}} \right) .$$

Let us define

$$\rho\left( v_1^{(0)}, W^{(0)}, b_1^{(0)} \right) := \frac{\sqrt{\tau m_1}}{C\sqrt{m_2}} \alpha \sim \mathcal{N}\left( 0, \frac{\tau}{C^2 m_2} \right) .$$

Then,

$$T_3 = \rho\left( v_1^{(0)}, W^{(0)}, b_1^{(0)} \right) \cdot \phi_\varepsilon(\langle w^*, x \rangle) + R_1 + R_2\left( x, v_1^{(0)}, W^{(0)}, b_1^{(0)} \right) ,$$

$$\text{where } |R_2| \leq \widetilde{O}\left( \frac{C}{\sqrt{\tau m_1}} \right) \times \frac{\tau}{C\sqrt{m_2}} = \widetilde{O}\left( \frac{\sqrt{\tau}}{\sqrt{m_1 m_2}} \right)$$

Note that $\alpha$ is independent of $u$ so $\rho$ is also independent of $u$. We can also define

$$\widetilde{\rho}\left( v_1^{(0)} \right) := \frac{\sqrt{\tau m_1}}{C\sqrt{m_2}} g \sim \mathcal{N}\left( 0, \frac{\tau}{C^2 m_2} \right) .$$

and using (C.12) we can derive the desired bound on $\mathcal{W}_2(\rho|_{W^{(0)}, b_1^{(0)}}, \widetilde{\rho})$ in the statement of Lemma 6.4.

**Next consider $T_4$.** Recall

$$T_4 := \sum_{i \in \mathcal{S}} v_{1,i}^{(0)} \mathbb{I}[\langle w_i^{(0)}, x \rangle + b_{1,i}^{(0)} \geq 0] \left( \langle \beta_i, x \rangle + b_{1,i}^{(0)} \right)$$

For fixed unit vector $x$, with high probability, we have that

$$\left| \alpha \frac{s_i}{\sqrt{|\mathcal{S}|}} \right| = \left| \frac{\alpha}{\sqrt{|\mathcal{S}|}} \right| = \widetilde{O}\left( \frac{1}{\sqrt{|\mathcal{S}| m_1}} \right)$$

and therefore

$$\langle w_i^{(0)}, x \rangle + b_{1,i}^{(0)} = \frac{\alpha s_i}{\sqrt{|\mathcal{S}|}} x_d + \langle \beta_i, x \rangle + b_{1,i}^{(0)} = \langle \beta_i, x \rangle + b_{1,i}^{(0)} \pm \widetilde{O}\left( \frac{1}{\sqrt{|\mathcal{S}| m_1}} \right) \ ,$$

By the above formula, for an index $i \in \mathcal{S}$ to have $\mathbb{I}[\langle \beta_i, x \rangle + b_{1,i}^{(0)} \geq 0] \neq \mathbb{I}[\langle w_i^{(0)}, x \rangle + b_{1,i}^{(0)} \geq 0]$, it must satisfy

$$\left| (\langle w_i^{(0)}, x \rangle + b_{1,i}^{(0)}) \right| \leq \widetilde{O}(\frac{1}{\sqrt{|\mathcal{S}| m_1}}) \ . \tag{C.13}$$

Thus, for fixed $\mathcal{S}$, since $W^{(0)}$ is independent of $\mathcal{S}$, with high probability over the randomness of $W^{(0)}$, there are at most $\widetilde{O}(\sqrt{|\mathcal{S}|})$ many indices $i \in \mathcal{S}$ satisfying (C.13). In other words, using $|v_{1,i}^{(0)}| \leq \widetilde{O}(1/\sqrt{m_2})$ with high probability, we have

$$T_4 = \underbrace{\sum_{i \in \mathcal{S}} v_{1,i}^{(0)} \mathbb{I}[\langle \beta_i, x \rangle + b_{1,i}^{(0)} \geq 0] \left( \langle \beta_i, x \rangle + b_{1,i}^{(0)} \right)}_{:=T_6} + R_3$$

with $R_3 = R_3\left( x, v_1^{(0)}, W^{(0)}, b_1^{(0)} \right)$ satisfying $|R_3| \leq \widetilde{O}\left( \frac{1}{\sqrt{m_1 m_2}} \right)$ with high probability.

**To bound $T_6$.** Recall

$$\mathcal{S} := \left\{ i \in [m_1]: \sqrt{m_2} v_{1,i}^{(0)} \in I\left( h\left( w_{i,1}^{(0)}, b_{1,i}^{(0)} \right) \right) \right\} \ .$$

and we can define a similar notion

$$\mathcal{S}' := \left\{ i \in [m_1]: \sqrt{m_2} v_{1,i}^{(0)} \in I\left( h\left( \beta_{i,1}, b_{1,i}^{(0)} \right) \right) \right\} \ .$$

Observe that

$$h\left( w_{i,1}^{(0)}, b_{1,i}^{(0)} \right) = h\left( \beta_{i,1} + \langle \alpha u_i e_d, e_1 \rangle, b_{1,i}^{(0)} \right) = h\left( \beta_{i,1}, b_{1,i}^{(0)} \right) \ .$$

Therefore, we have $\mathcal{S} = \mathcal{S}'$ and can write

$$T_6 = \underbrace{\sum_{i \in \mathcal{S}'} v_{1,i}^{(0)} \mathbb{I}[\langle \beta_i, x \rangle + b_{1,i}^{(0)} \geq 0] \left( \langle \beta_i, x \rangle + b_{1,i}^{(0)} \right)}_{:=B_2(x, v_1^{(0)}, W^{(0)}, b_1^{(0)})}$$

Again, conditioning on the randomness of $u$, we know that $B_2$ is independent of $\alpha$. Since $u$ and $\alpha$ are independent, we know that $\alpha$ and $B_2$ are also independent (by Proposition A.2). In other words, setting $B = B_1 + B_2$, $B$ and $\alpha$ (and therefore $\rho$) are independent.

Let $R = R_1 + R_2 + R_3$ be the residual term, setting $\tau = \frac{1}{100}$, we have $|R| \leq \widetilde{O}\left( \frac{1}{\sqrt{m_1 m_2}} \right)$ with high probability.

As for the norm bound on $B$, recall

$$n_1(x) = \sum_{i \in [m_1]} v_{1,i}^{(0)} \sigma\left( \langle w_i^{(0)}, x \rangle + b_{1,i}^{(0)} \right)$$

and by our random initialization, $n_1(x) \sim \mathcal{N}\left( 0, \frac{1}{m_2} \| \sigma(W^{(0)} x + b_1^{(0)}) \|_2^2 \right)$. At the same time, with high probability $\| \sigma(W^{(0)} x + b_1^{(0)}) \|_2^2 = O(1)$. Therefore, we know $|n_1(x)| \leq \widetilde{O}(\frac{1}{\sqrt{m_2}})$, and this implies $|B| \leq \widetilde{O}(\frac{1}{\sqrt{m_2}})$ with high probability. $\qquad \square$

### C.1.4  Lemma 6.2: Existence

**Lemma 6.2** (existence). *For every $\varepsilon \in \left(0, \frac{1}{kp_1p_2^2 \mathfrak{C}_s(\Phi, p_2\mathfrak{C}_s(\phi,1))\mathfrak{C}_s(\phi,1)^2}\right)$, there exists*

$$M = \mathsf{poly}\left(\mathfrak{C}_\varepsilon\left(\Phi, \sqrt{p_2}\mathfrak{C}_\varepsilon(\phi,1)\right), \frac{1}{\varepsilon}\right)$$

$$C_0 = \mathfrak{C}_\varepsilon(\Phi, \sqrt{p_2}\mathfrak{C}_\varepsilon(\phi,1)) \cdot \mathfrak{C}_\varepsilon(\phi,1) \cdot \widetilde{O}(p_1\sqrt{p_2}k)$$

*such that if $m_1, m_2 \geq M$, then with high probability, there exists weights $W^*, V^*$ with*

$$\|W^*\|_{2,\infty} = \max_i \|w_i^*\|_2 \leq \frac{C_0}{m_1}, \quad \|V^*\|_{2,\infty} = \max_i \|v_i^*\|_2 \leq \frac{\sqrt{m_1}}{m_2}$$

*such that*

$$\mathbb{E}_{(x,y)\sim\mathcal{D}}\left[\sum_{r=1}^{k}\left|f_r^*(x) - g_r^{(0)}(x; W^*, V^*)\right|\right] \leq \varepsilon,$$

*and hence,*

$$\mathbb{E}_{(x,y)\sim\mathcal{D}}\left[L(G^{(0)}(x; W^*, V^*), y)\right] \leq \mathsf{OPT} + \varepsilon.$$

Let us mostly focus on proving Lemma 6.2 for a single term

$$a^* \cdot \Phi\left(\sum_{j\in[p_2]} v_{1,j}^*\phi_{1,j}(\langle w_{1,j}^*, x\rangle)\right)\left(\sum_{j\in[p_2]} v_{2,j}^*\phi_{2,j}(\langle w_{2,j}^*, x\rangle)\right) \quad.$$

Extending it to multiple terms and multiple outputs, that is

$$f_r^*(x) = \sum_{i\in[p_1]} a_{r,i}^*\Phi_i\left(\sum_{j\in[p_2]} v_{1,i,j}^*\phi_{1,j}(\langle w_{1,j}^*, x\rangle)\right)\left(\sum_{j\in[p_2]} v_{2,i,j}^*\phi_{2,j}(\langle w_{2,j}^*, x\rangle)\right)$$

is rather straightforward (and indeed a repetition of the proof of Lemma B.1 in the two-layer case).

The proof consists of several steps.

**Step 1: Existence in expectation**

Recall that the input (without bias) to each neuron at random initialization in the second hidden layer is

$$n_i(x) = \sum_{r\in[m_1]} v_{i,r}^{(0)}\sigma\left(\langle w_r^{(0)}, x\rangle + b_{1,r}^{(0)}\right)$$

We first use Lemma 6.4 to derive the following claim:

**Claim C.3.** *For every $\varepsilon \in (0, 1/\mathfrak{C}_s(\phi,1))$, there exists real-value functions $\phi_{1,j,\varepsilon}(\cdot)$ satisfying*

$$|\phi_{1,j,\varepsilon}(\langle w_{1,j}^*, x\rangle) - \phi_{1,j}(\langle w_{1,j}^*, x\rangle)| \leq \varepsilon \text{ for all } j \in [p_2] \text{ and } \|x\|_2 = 1,$$

*and the following holds. Denote by*

$$C := \mathfrak{C}_\varepsilon(\phi,1), \quad C' := 10C\sqrt{p_2}, \quad \phi_{1,j,\varepsilon}(x) := \frac{1}{C'}\phi_{1,j,\varepsilon}(\langle w_{1,j}^*, x\rangle).$$

*For every $i \in [m_2]$, there exist independent Gaussians[21]*

$$\alpha_{i,j} \sim \mathcal{N}(0, 1/m_2) \text{ and } \beta_i(x) \sim \mathcal{N}\left(0, \frac{1}{m_2}\left(1 - \sum_{j\in[p_2]} \phi_{1,j,\varepsilon}^2(x)\right)\right) \quad,$$

*satisfying*

$$\mathcal{W}_2\left(n_i(x), \sum_{j\in[p_2]} \alpha_{i,j}\phi_{1,j}(x) + \beta(x)\right) \leq O\left(\frac{p_2^{2/3}}{m_1^{1/6}\sqrt{m_2}}\right)$$

*Proof of Claim C.3.* Let us define $p_2S$ many chunks of the first layer, each chunk corresponds to a set $\mathcal{S}_{j,l}$ of cardinality $|\mathcal{S}_{j,l}| = \frac{m_1}{p_2S}$ for $j \in [p_2], l \in [S]$, such that

$$\mathcal{S}_{j,l} = \left\{ (j-1)\frac{m_1}{p_2} + (l-1)\frac{m_1}{p_2S} + k \,\middle|\, k \in \left[\frac{m_1}{p_2S}\right] \right\} \subset [m_1]$$

Let us then denote $v_i[j,l]$ to be $(v_{i,r})_{r \in \mathcal{S}_{j,l}}$ and $W[j,l]$ to be $(W_r)_{r \in \mathcal{S}_{j,l}}$. Recall that the input (without bias) to each neuron at random initialization in the second hidden layer is

$$n_i(x) = \sum_{r \in [m_1]} v_{i,r}^{(0)} \sigma\left(\langle w_r^{(0)}, x \rangle + b_{1,r}^{(0)}\right)$$

$$= \sum_{j \in [p_2]} \sum_{l \in [S]} \sum_{r \in \mathcal{S}_{j,l}} v_{i,r}^{(0)} \sigma\left(\langle w_r^{(0)}, x \rangle + b_{1,r}^{(0)}\right) \ .$$

For each $j \in [p_2]$ and $l \in [S]$, let us apply Lemma 6.4 to the summation $\sum_{r \in \mathcal{S}_{j,l}} \dots$ in the above formula, to approximate $\phi_{1,j}(\langle w_{1,j}^*, x \rangle)$. (We need to replace $m_1$ with $\frac{m_1}{p_2S}$ and scale up $W^{(0)}$ and $b_1^{(0)}$ by $\sqrt{p_2S}$ before applying Lemma 6.4)). It tells us we can write $n_i(x)$ as:

$$n_i(x) = \sum_{j \in [p_2], l \in [S]} \rho_j\left(v_i^{(0)}[j,l], W^{(0)}[j,l], b_1^{(0)}[j,l]\right) \phi_{1,j,\varepsilon}(\langle w_{1,j}^*, x \rangle)$$

$$+ \sum_{j \in [p_2], l \in [S]} B_j\left(x, v_i^{(0)}[j,l], W^{(0)}[j,l], b_1^{(0)}[j,l]\right) + R_j\left(x, v_i^{(0)}[j,l], W^{(0)}[j,l], b_1^{(0)}[j,l]\right)$$

where random variables $\rho_{j,l} := \rho_j\left(v_i^{(0)}[j,l], W^{(0)}[j,l], b_1^{(0)}[j,l]\right) \sim \mathcal{N}(0, \frac{1}{100C^2 m_2(p_2S)})$ are independent Gaussian for different $j$ and $l$. Let $C := \mathfrak{C}_\varepsilon(\phi, 1)$. We know that

$$\rho_j := \sum_{l \in [S]} \rho_{j,l} \sim \mathcal{N}\left(0, \frac{1}{100C^2 p_2 m_2}\right) = \mathcal{N}\left(0, \frac{1}{C'^2 m_2}\right)$$

for

$$C' := 10C\sqrt{p_2} \ .$$

Moreover, $|\phi_{1,j,\varepsilon}(\langle w_{1,j}^*, x \rangle) - \phi_{1,j}(\langle w_{1,j}^*, x \rangle)| \leq \varepsilon$ for each $j \in [p_2]$.

Let us then denote

$$B_j^{\mathsf{s}}(x) := \sum_{l \in [S]} B_j\left(x, v_i^{(0)}[j,l], W^{(0)}[j,l], b_1^{(0)}[j,l]\right)$$

$$R_j^{\mathsf{s}}(x) := \sum_{l \in [S]} R_j\left(x, v_i^{(0)}[j,l], W^{(0)}[j,l], b_1^{(0)}[j,l]\right)$$

Lemma 6.4 tells us that random variables $\rho_j$ are independent of $B_j(x)$, and with high probability

$$|R_j^{\mathsf{s}}(x)| \leq S \times \widetilde{O}\left(\frac{1}{\sqrt{(m_1/(p_2S))m_2}} \frac{1}{\sqrt{p_2S}}\right) \leq \widetilde{O}\left(\frac{S}{\sqrt{m_1 m_2}}\right) \text{ and}$$

$$\left|B_j\left(x, v_i^{(0)}[j,l], W^{(0)}[j,l], b_1^{(0)}[j,l]\right)\right| \leq \frac{1}{\sqrt{p_2S}} \times \widetilde{O}\left(\frac{1}{\sqrt{m_2}}\right) = \widetilde{O}\left(\frac{1}{\sqrt{m_2 p_2S}}\right)$$

Let us apply the Wasserstein distance version of the central limit theorem (see for instance [18, Theorem 1]) [22]: since $B_j^{\mathsf{s}}(x)$ is the summation of $S$ i.i.d random variables, there is a Gaussian random variable $\beta_j(x)$ only depending on the randomness of $B_j^{\mathsf{s}}(x)$ such that

$$\mathcal{W}_2(B_j^{\mathsf{s}}(x), \beta_j(x)) \leq \widetilde{O}\left(\frac{1}{\sqrt{m_2 p_2S}}\right)$$

Define $\beta'(x) = \sum_{j\in[p_2]} \beta_j(x)$, we know that $\beta'(x)$ is a Gaussian random variable independent of all the $\rho_j$ with

$$\mathcal{W}_2\left(n_i(x), \sum_{j\in[p_2]} \rho_j\phi_{1,j,\varepsilon}(\langle w_{1,j}^*, x\rangle) + \beta'(x)\right) \leq \widetilde{O}\left(\frac{Sp_2}{\sqrt{m_1 m_2}} + \frac{\sqrt{p_2}}{\sqrt{m_2 S}}\right)$$

Let us slightly override notation and denote

$$\phi_{1,j,\varepsilon}(x) = \frac{1}{C'}\phi_{1,j,\varepsilon}(\langle w_{1,j}^*, x\rangle),$$

We then have that variables $\alpha_{i,j} := C'\rho_j \sim \mathcal{N}(0, 1/m_2)$ are i.i.d. and

$$\mathcal{W}_2\left(n_i(x), \sum_{j\in[p_2]} \alpha_{i,j}\phi_{1,j,\varepsilon}(x) + \beta'(x)\right) \leq \widetilde{O}\left(\frac{Sp_2}{\sqrt{m_1 m_2}} + \frac{\sqrt{p_2}}{\sqrt{m_2 S}}\right)$$

Since by our random initialization, $n_i(x) \sim \mathcal{N}\left(0, \frac{1}{m_2}\left\|\sigma\left(W^{(0)}x + b_1^{(0)}\right)\right\|_2^2\right)$, and since for every every unit vector $x$, with high probability $\left\|\sigma\left(W^{(0)}x + b_1^{(0)}\right)\right\|_2^2 = 1 \pm \widetilde{O}\left(\frac{1}{\sqrt{m_1}}\right)$, we can write

$$\mathcal{W}_2(n_i(x), g) \leq \widetilde{O}(\frac{1}{\sqrt{m_1 m_2}}) \quad \text{for } g \sim \mathcal{N}(0, \frac{1}{m_2}). \tag{C.14}$$

Since we can write $g = \sum_{j\in[p_2]} \alpha_{i,j}\phi_{1,j,\varepsilon}(x) + \beta_i(x)$ for

$$\beta_i(x) \sim \mathcal{N}\left(0, \frac{1}{m_2}\left(1 - \sum_{j\in[p_2]} \phi_{1,j,\varepsilon}^2(x)\right)\right),$$

being an independent from $\alpha_{i,1}, \ldots, \alpha_{i,p_2}$, we conclude that (by choosing $S = (m_1/p_2)^{1/3}$)

$$\mathcal{W}_2\left(n_i(x), \sum_{j\in[p_2]} \alpha_{i,j}\phi_{1,j,\varepsilon}(x) + \beta_i(x)\right) \leq \widetilde{O}\left(\frac{1}{\sqrt{m_1 m_2}} + \frac{Sp_2}{\sqrt{m_1 m_2}} + \frac{\sqrt{p_2}}{\sqrt{m_2 S}}\right) \leq O\left(\frac{p_2^{2/3}}{m_1^{1/6}\sqrt{m_2}}\right).$$
$$\tag{C.15}$$

This finishes the proof of Claim C.3. $\qquad\square$

**Claim C.4.** *In the same notations as Claim C.3, there exists function $h\colon \mathbb{R}^2 \to [-C'', C'']$ for $C'' = \mathfrak{C}_\varepsilon(\Phi, C')$ such that for every $i \in [m_2]$,*

$$\mathbb{E}\left[\mathbb{I}_{n_i(x)+b_{2,i}^{(0)}\geq 0} h\left(\sum_{j\in[p_2]} v_{1,j}^*\alpha_{i,j}, b_{2,i}^{(0)}\right)\left(\sum_{j\in[p_2]} v_{2,j}^*\phi_{2,j}(\langle w_{2,j}^*, x\rangle)\right)\right]$$

$$= \Phi\left(\sum_{j\in[p_2]} v_{1,j}^*\phi_{1,j}(\langle w_{1,j}^*, x\rangle)\right)\left(\sum_{j\in[p_2]} v_{2,j}^*\phi_{2,j}(\langle w_{2,j}^*, x\rangle)\right) \pm O\left(p_2^2\mathfrak{C}_\mathfrak{s}(\Phi, p_2\mathfrak{C}_\mathfrak{s}(\phi, 1))\mathfrak{C}_\mathfrak{s}(\phi, 1)\varepsilon\right)$$

*Proof of Claim C.4.* Let us slightly override notation and denote
$$\phi_{2,j}(x) := \phi_{2,j}(\langle w_{2,j}^*, x\rangle).$$

We apply Lemma 6.3 again with $\phi$ chosen as $\Phi'(z) = \Phi(C'x)$.[23] We know there exists a function $h\colon \mathbb{R}^2 \to [-C'', C'']$ for $C'' = \mathfrak{C}_\varepsilon(\Phi', 1) = \mathfrak{C}_\varepsilon(\Phi, C')$ such that

$$\mathbb{E}_{\alpha_i, \beta_i}\left[\mathbb{I}_{\sum_{j\in[p_2]}\alpha_{i,j}\phi_{1,j,\varepsilon}(x)+\beta_i(x)+b_{2,i}^{(0)}\geq 0} h\left(\sum_{j\in[p_2]} v_{1,j}^*\alpha_{i,j}, b_{2,i}^{(0)}\right)\left(\sum_{j\in[p_2]} v_{2,j}^*\phi_{2,j}(x)\right)\right]$$

$$= \Phi \left( C' \sum_{j \in [p_2]} v_{1,j}^* \phi_{1,j,\varepsilon}(x) \right) \left( \sum_{j \in [p_2]} v_{2,j}^* \phi_{2,j}(x) \right) \pm \varepsilon C''' \tag{C.16}$$

where

$$C''' = \sup_{x: \|x\|_2 \leq 1} \left| \sum_{j \in [p_2]} v_{2,j}^* \phi_{2,j}(x) \right| \leq p_2 \mathfrak{C}_{\mathfrak{s}}(\phi, 1) \ .$$

Next, we wish to use the Wasserstein bound from Claim C.3 to replace $\sum_{j \in [p_2]} \alpha_{i,j} \phi_{1,j,\varepsilon}(x) + \beta_i(x)$ with $n_i(x)$. We derive that

$$\mathbb{E} \left[ \mathbb{I}_{n_i(x) + b_{2,i}^{(0)} \geq 0} h \left( \sum_{j \in [p_2]} v_{1,j}^* \alpha_{i,j}, b_{2,i}^{(0)} \right) \left( \sum_{j \in [p_2]} v_{2,j}^* \phi_{2,j}(x) \right) \right]$$

$$\overset{①}{=} \mathbb{E} \left[ \mathbb{I}_{\alpha_{i,j}\phi_{1,j,\varepsilon}(x) + \beta_i(x) + b_{2,i}^{(0)} \geq 0} h \left( \sum_{j \in [p_2]} v_{1,j}^* \alpha_{i,j}, b_{2,i}^{(0)} \right) \left( \sum_{j \in [p_2]} v_{2,j}^* \phi_{2,j}(x) \right) \right]$$

$$\pm O \left( \mathcal{W}_2 \left( n_i(x), \sum_{j \in [p_2]} \alpha_{i,j}\phi_{1,j,\varepsilon}(x) + \beta_i(x) \right) \sqrt{m_2} C''' C'' \right) \tag{C.17}$$

$$\overset{②}{=} \Phi \left( C' \sum_{j \in [p_2]} v_{1,j}^* \phi_{1,j,\varepsilon}(x) \right) \left( \sum_{j \in [p_2]} v_{2,j}^* \phi_{2,j}(x) \right) \pm O \left( \varepsilon C''' + \frac{C'' C''' p_2^{2/3}}{m_1^{1/6}} \right)$$

$$\overset{③}{=} \Phi \left( \sum_{j \in [p_2]} v_{1,j}^* \phi_{1,j}(\langle w_{1,j}^*, x \rangle) \right) \left( \sum_{j \in [p_2]} v_{2,j}^* \phi_{2,j}(\langle w_{2,j}^*, x \rangle) \right) \pm O \left( \varepsilon C''' + C''' \cdot (\varepsilon p_2) L_\Phi + \frac{C'' C''' p_2^{2/3}}{m_1^{1/6}} \right)$$

$$\overset{④}{=} \Phi \left( \sum_{j \in [p_2]} v_{1,j}^* \phi_{1,j}(\langle w_{1,j}^*, x \rangle) \right) \left( \sum_{j \in [p_2]} v_{2,j}^* \phi_{2,j}(\langle w_{2,j}^*, x \rangle) \right) \pm O \left( p_2^2 \mathfrak{C}_{\mathfrak{s}}(\Phi, p_2 \mathfrak{C}_{\mathfrak{s}}(\phi, 1)) \mathfrak{C}_{\mathfrak{s}}(\phi, 1) \varepsilon \right) \ .$$

Above, ① uses the fact that $n_i \sim \mathcal{N} \left( 0, \frac{1}{m_2} \left\| \sigma \left( W^{(0)} x + b_1^{(0)} \right) \right\|_2^2 \right)$ with $\left\| \sigma \left( W^{(0)} x + b_1^{(0)} \right) \right\|_2 \leq 2$ w.h.p. and $|h| \leq C''$ is bounded. ② uses Claim C.3 and (C.16). ③ uses $C' \phi_{1,j,\varepsilon}(x) = \phi_{1,j,\varepsilon}(\langle w_{1,j}^*, x \rangle)$, $\phi_{2,j}(x) = \phi_{2,j}(\langle w_{2,j}^*, x \rangle)$, and denote by $L_\Phi$ the Lipschitz continuity parameter of $\Phi$ (namely, $|\Phi(x) - \Phi(y)| \leq L_\Phi |x - y|$ for all $x, y \in \left[ -p_2 \mathfrak{C}_{\mathfrak{s}}(\phi, 1), p_2 \mathfrak{C}_{\mathfrak{s}}(\phi, 1) \right]$). ④ uses $L_\Phi \leq \mathfrak{C}_{\mathfrak{s}}(\Phi, p_2 \mathfrak{C}_{\mathfrak{s}}(\phi, 1))$ and our assumption $m_1 \geq M$. This proves Claim C.4. $\square$

**Step 2: From expectation to finite neurons**

Intuitively, we wish to apply concentration bound on Claim C.4 with respect to all neurons $i \in [m_2]$ on the second layer. Recall $a_i \sim \mathcal{N}(0, \varepsilon_a)$ is the weight of the $i$-th neuron at the output layer. Our main result of Step 2 is the following claim.

**Claim C.5.** *In the same notation as Claim C.4,*

$$\frac{1}{m_2} \sum_{i \in [m_2]} \left[ \frac{a_i^2}{\varepsilon_a^2} \mathbb{I}_{n_i(x) + b_{2,i}^{(0)} \geq 0} h \left( \sum_{j \in [p_2]} v_{1,j}^* \alpha_{i,j}, b_{2,i}^{(0)} \right) \left( \sum_{j \in [p_2]} v_{2,j}^* \phi_{2,j}(\langle w_{2,j}^*, x \rangle) \right) \right]$$

$$= \Phi \left( \sum_{j \in [p_2]} v_{1,j}^* \phi_{1,j}(\langle w_{1,j}^*, x \rangle) \right) \left( \sum_{j \in [p_2]} v_{2,j}^* \phi_{2,j}(\langle w_{2,j}^*, x \rangle) \right) \pm O \left( p_2^2 \mathfrak{C}_{\mathfrak{s}}(\Phi, p_2 \mathfrak{C}_{\mathfrak{s}}(\phi, 1)) \mathfrak{C}_{\mathfrak{s}}(\phi, 1) \varepsilon \right)$$

*Proof of Claim C.5.* Across different choices of $i$, the values of $n_i(x)$ and $\alpha_{i,j}$ can be correlated. This makes it not a trivial thing to apply concentration. In the remainder of this proof, let us try to modify the two quantities to make them independent across $i$.

**First modify $\alpha_{i,j}$.** Recall $\alpha_{i,j} = C'\rho_j = C'\sum_{l\in[S]}\rho_{j,l}$ where each $\rho_{j,l}$ is a function on $v_i^{(0)}[j,l], W^{(0)}[j,l]$, and $b_1^{(0)}[j,l]$. Now, using the $\widetilde{\rho}$ notion from Lemma 6.4, let us also define $\widetilde{\rho}_{j,l} = \widetilde{\rho}_j\left(v_i^{(0)}[j,l]\right)$ which is in the same distribution as $\rho_{j,l}$ except that it does not depend on $W^{(0)}$ or $b_1^{(0)}$. We can similarly let $\widetilde{\rho}_j = \sum_{l\in[S]}\widetilde{\rho}_{j,l}$. From Lemma 6.4, we know that with high probability over $W^{(0)}$:[24]

$$\widetilde{\rho}_j \sim \mathcal{N}\left(0, \frac{1}{C'^2 m_2}\right) \quad \text{and} \quad \mathcal{W}_2(\rho_{j,l}|_{W^{(0)},b^{(0)}}, \widetilde{\rho}_{j,l}) \leq \widetilde{O}\left(\frac{1}{C\sqrt{m_1 m_2}}\right)$$

$$\implies \mathcal{W}_2(\rho_j|_{W^{(0)},b^{(0)}}, \widetilde{\rho}_j) \leq \widetilde{O}\left(\frac{S}{C\sqrt{m_1 m_2}}\right) \qquad (\text{C.18})$$

According, we define $\widetilde{\alpha}_{i,j} = C'\widetilde{\rho}_j = C'\sum_{l\in[S]}\widetilde{\rho}_{j,l}$.

**Next modifty $n_i(x)$.** Recall $n_i(x) = \sum_{r\in[m_1]} v_{i,r}^{(0)}\sigma\left(\langle w_r^{(0)}, x\rangle + b_{1,r}^{(0)}\right)$ and accordingly we define

$$\widetilde{n}_i(x) := \frac{\sum_{r\in[m_1]} v_{i,r}^{(0)}\sigma\left(\langle w_r^{(0)}, x\rangle + b_{1,r}^{(0)}\right)}{\|u\|_2}\mathbb{E}[\|u\|_2]$$

where vector $u := \left(\sigma(\langle w_r^{(0)}, x\rangle + b_{1,r}^{(0)})\right)_{r\in[m_1]}$. By definition, we know

$$\widetilde{n}_i(x) \sim \mathcal{N}\left(0, \frac{1}{m_2}\mathbb{E}[\|u\|_2]^2\right)$$

is a Gaussian variable and is independent of $u$. As a consequence, the quantities $\widetilde{n}_i(x)$ are independent among different choices of $i \in [m_1]$. Using standard concentration on $\|u\|_2$ (see the line above (C.14)), we have for every $x$ with $\|x\|_2 = 1$,

$$\mathcal{W}_2(n_i(x), \widetilde{n}_i(x)) \leq \widetilde{O}\left(\frac{1}{\sqrt{m_1 m_2}}\right) \qquad (\text{C.19})$$

**Concentration.** Using Claim C.4 of Step 1 we have

$$\mathbb{E}\left[\mathbb{I}_{n_i(x)+b_{2,i}^{(0)}\geq 0}h\left(\sum_{j\in[p_2]}v_{1,j}^*\alpha_{i,j}, b_{2,i}^{(0)}\right)\left(\sum_{j\in[p_2]}v_{2,j}^*\phi_{2,j}(\langle w_{2,j}^*, x\rangle)\right)\right]$$

$$= \Phi\left(\sum_{j\in[p_2]}v_{1,j}^*\phi_{1,j}(\langle w_{1,j}^*, x\rangle)\right)\left(\sum_{j\in[p_2]}v_{2,j}^*\phi_{2,j}(\langle w_{2,j}^*, x\rangle)\right) \pm O\left(p_2^2\mathfrak{C}_\mathfrak{s}(\Phi, p_2\mathfrak{C}_\mathfrak{s}(\phi,1))\mathfrak{C}_\mathfrak{s}(\phi,1)\varepsilon\right)$$

Using the notions of $\widetilde{n}$ and $\widetilde{\alpha}$ and the Wasserstein distance bounds (C.18) and (C.19), it implies[25]

$$\mathbb{E}\left[\mathbb{I}_{\widetilde{n}_i(x)+b_{2,i}^{(0)}\geq 0}h\left(\sum_{j\in[p_2]}v_{1,j}^*\widetilde{\alpha}_{i,j}, b_{2,i}^{(0)}\right)\left(\sum_{j\in[p_2]}v_{2,j}^*\phi_{2,j}(\langle w_{2,j}^*, x\rangle)\right)\right]$$

$$= \Phi\left(\sum_{j\in[p_2]}v_{1,j}^*\phi_{1,j}(\langle w_{1,j}^*, x\rangle)\right)\left(\sum_{j\in[p_2]}v_{2,j}^*\phi_{2,j}(\langle w_{2,j}^*, x\rangle)\right) \pm O\left(p_2^2\mathfrak{C}_\mathfrak{s}(\Phi, p_2\mathfrak{C}_\mathfrak{s}(\phi,1))\mathfrak{C}_\mathfrak{s}(\phi,1)\varepsilon\right)$$

Using $\mathbb{E}_{a_i}[\frac{a_i^2}{\varepsilon_a^2}] = 1$ and applying standard concentration —and the independence of tuples $(a_i, \widetilde{n}_i(x), (\widetilde{\alpha}_{i,j})_{j\in[p_2]})$ with respect to different choices of $i$— we know with high probability

$$\frac{1}{m_2}\sum_{i\in[m_2]}\left[\frac{a_i^2}{\varepsilon_a^2}\mathbb{I}_{\widetilde{n}_i(x)+b_{2,i}^{(0)}\geq 0}h\left(\sum_{j\in[p_2]}v_{1,j}^*\widetilde{\alpha}_{i,j}, b_{2,i}^{(0)}\right)\left(\sum_{j\in[p_2]}v_{2,j}^*\phi_{2,j}(\langle w_{2,j}^*, x\rangle)\right)\right]$$

$$= \mathbb{E}\left[\mathbb{I}_{\widetilde{n}_i(x)+b_{2,i}^{(0)}\geq 0} h\left(\sum_{j\in[p_2]} v_{1,j}^* \widetilde{\alpha}_{i,j}, b_{2,i}^{(0)}\right)\left(\sum_{j\in[p_2]} v_{2,j}^* \phi_{2,j}(\langle w_{2,j}^*, x\rangle)\right)\right] \pm \widetilde{O}\left(\frac{C''C'''}{\sqrt{m_2}}\right)$$

Using again the Wasserstein distance bounds bounds (C.18) and (C.19), we can combine the above two equations to derive that w.h.p.

$$\frac{1}{m_2}\sum_{i\in[m_2]}\left[\frac{a_i^2}{\varepsilon_a^2}\mathbb{I}_{n_i(x)+b_{2,i}^{(0)}\geq 0} h\left(\sum_{j\in[p_2]} v_{1,j}^* \alpha_{i,j}, b_{2,i}^{(0)}\right)\left(\sum_{j\in[p_2]} v_{2,j}^* \phi_{2,j}(\langle w_{2,j}^*, x\rangle)\right)\right]$$
$$= \Phi\left(\sum_{j\in[p_2]} v_{1,j}^* \phi_{1,j}(\langle w_{1,j}^*, x\rangle)\right)\left(\sum_{j\in[p_2]} v_{2,j}^* \phi_{2,j}(\langle w_{2,j}^*, x\rangle)\right) \pm O\left(p_2^2\mathfrak{C}_{\mathfrak{s}}(\Phi, p_2\mathfrak{C}_{\mathfrak{s}}(\phi,1))\mathfrak{C}_{\mathfrak{s}}(\phi,1)\varepsilon\right)$$

$\square$

### Step 3: From finite neurons to the network

We now construct the network and prove Lemma 6.2. Recall we focus on constructing

$$a^* \cdot \Phi\left(\sum_{j\in[p_2]} v_{1,j}^* \phi_{1,j}(\langle w_{1,j}^*, x\rangle)\right)\left(\sum_{j\in[p_2]} v_{2,j}^* \phi_{2,j}(\langle w_{2,j}^*, x\rangle)\right) \quad.$$

We shall explain towards the end how to extend it to multiple terms and multiple outputs.

For each $\phi_{2,j}$, applying Lemma 6.3b, we can construct $h_{\phi,j}\colon \mathbb{R}^2 \to [-C, C]$ satisfying

$$\forall i' \in [m_1]\colon \quad \mathbb{E}\left[h_{\phi,j}\left(\langle w_{2,j}^*, w_{i'}^{(0)}\rangle, b_{1,i'}^{(0)}\right)\mathbb{I}_{\langle w_{i'}^{(0)}, x\rangle+b_{1,i'}^{(0)}\geq 0}\right] = \phi_{2,j}(\langle w_{2,j}^*, x\rangle) \pm \varepsilon \quad. \quad (C.20)$$

Now consider an arbitrary vector $v \in \mathbb{R}^{m_1}$ with $v_i \in \{-1, 1\}$.

- Define $V^* \in \mathbb{R}^{m_2 \times m_1}$ as

$$V^* = (C_0 \cdot C''/C)^{-1/2}\frac{a^*}{m_2}\left(a_i h\left(\sum_{j\in[p_2]} v_{1,j}^* \alpha_{i,j}, b_{2,i}^{(0)}\right)v^\top\right)_{i\in[m_2]}$$

  where the notations of $h\colon \mathbb{R}^2 \to [-C'', C'']$ and $\alpha_{i,j}$ come from Claim C.4. We have

$$\|V^*\|_{2,\infty} \leq \frac{\widetilde{O}(\sqrt{C''C})}{\sqrt{C_0}} \cdot \frac{\sqrt{m_1}}{m_2}$$

- Define $W^* \in \mathbb{R}^{m_1 \times d}$ as

$$W^* = (C_0 \cdot C''/C)^{1/2}\frac{2}{\varepsilon_a^2 m_1}\left(v_i \sum_{j\in[p_2]} v_{2,j}^* h_{\phi,j}\left(\langle w_{2,j}^*, w_i^{(0)}\rangle, b_{1,i}^{(0)}\right)e_d\right)_{i\in[m_1]} \quad.$$

  We have

$$\|W^*\|_{2,\infty} \leq \frac{2\sqrt{p_2}\sqrt{C_0 \cdot C''C}}{m_1},$$

Given these weights, suppose the signs of ReLU's are determined by the random initialization (i.e., by $W^{(0)}$ and $V^{(0)}$). We can consider the network output

$$g^{(0)}(x; W^*, V^*) := aD_{v,x}V^*D_{w,x}W^*x$$
$$= \sum_{i\in[m_2]} a_i\mathbb{I}_{n_i(x)+b_{2,i}^{(0)}\geq 0}\sum_{i'\in[m_1]} v_{i,i'}^*\langle w_i^*, x\rangle\mathbb{I}_{\langle w_{i'}^{(0)}, x\rangle+b_{1,i'}^{(0)}\geq 0}$$
$$= \frac{a^*}{m_2}\sum_{i\in[m_2]}\frac{a_i^2}{\varepsilon_a^2}h\left(\sum_{j\in[p_2]} v_{1,j}^* \alpha_{i,j}, b_{2,i}^{(0)}\right)\mathbb{I}_{n_i(x)+b_{2,i}^{(0)}\geq 0}$$

$$\times \sum_{j\in[p_2]} v_{2,j}^* \left( \frac{1}{m_1} \sum_{i'\in[m_1]} h_{\phi,j}\left( \langle w_{2,j}^*, w_{i'}^{(0)} \rangle, b_{1,i'}^{(0)} \right) \mathbb{I}_{\langle w_{i'}^{(0)}, x \rangle + b_{1,i'} \geq 0} \right) \tag{C.21}$$

Above, we have used $\langle e_d, x \rangle = x_d = 1/2$. Since $h_{\phi,j} \in [-C'', C'']$, we can apply concentration on (C.20) and get (recalling $m_1$ is sufficiently large) w.h.p

$$\frac{1}{m_1} \sum_{i'\in[m_1]} h_{\phi,j}\left( \langle w_{2,j}^*, w_{i'}^{(0)} \rangle, b_{1,i'}^{(0)} \right) \mathbb{I}_{\langle w_{i'}^{(0)}, x \rangle + b_{1,i'} \geq 0} = \phi_{2,j}(\langle w_{2,j}^*, x \rangle) \pm 2\varepsilon$$

Using Claim C.5, we have

$$\frac{1}{m_2} \sum_{i\in[m_2]} \left[ \frac{a_i^2}{\varepsilon_a^2} \mathbb{I}_{n_i(x)+b_{2,i}^{(0)} \geq 0} h\left( \sum_{j\in[p_2]} v_{1,j}^* \alpha_{i,j}, b_{2,i}^{(0)} \right) \left( \sum_{j\in[p_2]} v_{2,j}^* \phi_{2,j}(\langle w_{2,j}^*, x \rangle) \right) \right]$$

$$= \Phi\left( \sum_{j\in[p_2]} v_{1,j}^* \phi_{1,j}(\langle w_{1,j}^*, x \rangle) \right) \left( \sum_{j\in[p_2]} v_{2,j}^* \phi_{2,j}(\langle w_{2,j}^*, x \rangle) \right) \pm O\left( p_2^2 \mathfrak{C}_{\mathfrak{s}}(\Phi, p_2 \mathfrak{C}_{\mathfrak{s}}(\phi, 1)) \mathfrak{C}_{\mathfrak{s}}(\phi, 1) \varepsilon \right)$$

Putting this into (C.21), and using $a^* \in [-1, 1]$ and $h \in [-C'', C'']$, we know that with high probability

$$g^{(0)}(x; W^*, V^*) = a^* \cdot \Phi\left( \sum_{j\in[p_2]} v_{1,j}^* \phi_{1,j}(\langle w_{1,j}^*, x \rangle) \right) \left( \sum_{j\in[p_2]} v_{2,j}^* \phi_{2,j}(\langle w_{2,j}^*, x \rangle) \right)$$

$$\pm O\left( p_2^2 \mathfrak{C}_{\mathfrak{s}}(\Phi, p_2 \mathfrak{C}_{\mathfrak{s}}(\phi, 1)) \mathfrak{C}_{\mathfrak{s}}(\phi, 1) \varepsilon \right)$$

Finally, scaling down the value of $\varepsilon$ by factor $p_2^2 \mathfrak{C}_{\mathfrak{s}}(\Phi, p_2 \mathfrak{C}_{\mathfrak{s}}(\phi, 1)) \mathfrak{C}_{\mathfrak{s}}(\phi, 1)$ we complete the proof of Lemma 6.2 for a single output and for a single $\Phi$.

*Remark C.6.* The proof generalizes to fitting a combination of multiple functions

$$\sum_{i\in[p_1]} a_i^* \Phi_i\left( \sum_{j\in[p_2]} v_{1,i,j}^* \phi_{1,j}(\langle w_{1,j}^*, x \rangle) \right) \left( \sum_{j\in[p_2]} v_{2,i,j}^* \phi_{2,j}(\langle w_{2,j}^*, x \rangle) \right)$$

in the same way as the proof of Lemma B.1, if we choose the vector $v \in \{-1, 1\}^{m_1}$ uniformly at random and apply concentration. Note that we have to further scale down $\varepsilon$ by $\frac{1}{p_1}$ for the same reason as Lemma B.1, and the norm of $V^*$ shall grow by a factor of $p_1$.

*Remark C.7.* The proof generalizes to multiple outputs in the same way as Lemma B.1, using the fact that weights $a_{r,i}$ are independent across different outputs $r$. Note that we have to further scale down $\varepsilon$ by $\frac{1}{k}$ for the same reason as Lemma B.1, and the norm of $V^*$ shall grow by a factor of $k$.

Finally, applying the two remarks above, we have

$$\|V^*\|_{2,\infty} \leq \frac{\widetilde{O}(\sqrt{C''C})p_1 k}{\sqrt{C_0}} \cdot \frac{\sqrt{m_1}}{m_2} \quad \text{and} \quad \|W^*\|_{2,\infty} \leq \frac{2\sqrt{p_2}\sqrt{C_0 \cdot C''C}}{m_1},$$

We thus finish the proof of Lemma 6.2 with our choice of $C_0$. ∎

## C.2 Coupling

### C.2.1 Lemma 6.5: Coupling

**Lemma 6.5** (coupling, restated). *Suppose $\tau_v \in [0, 1]$, $\tau_w \in \left( \frac{1}{m_1^{3/2}}, \frac{1}{m_1^{1/2}} \right]$, $\sigma_w \in \left( \frac{1}{m_1^{3/2}}, \frac{\tau_w}{m_1^{1/4}} \right]$, $\sigma_v \in \left( 0, \frac{1}{m_2^{1/2}} \right]$ and $\eta > 0$. Given fixed unit vector $x$, and perturbation matrices $W', V', W'', V''$ (that may depend on the randomness of $W^{(0)}, b_1^{(0)}, V^{(0)}, b_2^{(0)}$ and $x$) satisfying*

$$\|W'\|_{2,4} \leq \tau_w, \|V'\|_F \leq \tau_v, \|W''\|_{2,4} \leq \tau_w, \|V''\|_F \leq \tau_v ,$$

*and random diagonal matrix $\Sigma$ with each diagonal entry i.i.d. drawn from $\{\pm 1\}$, then with high probability the following holds:*

1. *(Sparse sign change).* $\|D'_{w,x}\|_0 \leq \widetilde{O}(\tau_w^{4/5} m_1^{6/5})$, $\|D'_{v,x}\|_0 \leq \widetilde{O}\left(\sigma_v m_2^{3/2} + \tau_v^{2/3} m_2 + \tau_w^{2/3} m_1^{1/6} m_2\right)$.

2. *(Cross term vanish).*

$$g_r(x; W^{(0)} + W^\rho + W' + \eta\boldsymbol{\Sigma}W'', V^{(0)} + V^\rho + V' + \eta V''\boldsymbol{\Sigma})$$
$$= g_r\left(x; W^{(0)} + W^\rho + W', V^{(0)} + V^\rho + V'\right) + g_r^{(b,b)}(x; \eta\boldsymbol{\Sigma}W'', \eta V''\boldsymbol{\Sigma}) + g_r'(x)$$

*where* $\mathbb{E}_{\boldsymbol{\Sigma}}[g_r'(x)] = 0$ *and with high probability* $|g_r'(x)| \leq \eta\widetilde{O}\left(\frac{\sqrt{m_2}\tau_v}{\sqrt{m_1}} + m_2^{1/2}\tau_w\right)$.

**Part I, Sparsity**

For notation simplicity, we only do the proof when there is no bias term. The proof with bias term is analogous (but more notationally involved). Let us denote

$$z_0 := D_{w,x} W^{(0)} x$$
$$z_2 := (D_{w,x} + D'_{w,x})(W^{(0)} + W^\rho + W')x - D_{w,x} W^{(0)} x \ .$$

We let

- diagonal matrix $D_{w,x}$ denotes the sign of ReLU's at weights $W^{(0)}$,

- diagonal matrix $D_{w,x} + D''_{w,x}$ denotes the sign of ReLU's at weights $W^{(0)} + W^\rho$, and

- diagonal matrix $D_{w,x} + D'_{w,x}$ denotes the sign of ReLU's at weights $W^{(0)} + W^\rho + W'$.

**Sign change in $D''_{w,x}$.** Each coordinate of $W^{(0)}x \sim \mathcal{N}\left(0, \frac{1}{m_1}\right)$ and each coordinate of $W^\rho x \sim \mathcal{N}(0, \sigma_w^2)$. Thus, by standard property of Gaussian, for each $i$, we have $\mathbf{Pr}[|W_i^\rho x| \geq |W_i^{(0)}x|] \leq \widetilde{O}\left(\sigma_w\sqrt{m_1}\right)$. By concentration bound, with high probability, the number of sign changes of the ReLU activations in the first hidden layer caused by adding $W^\rho$ is no more than

$$\|D''_{w,x}\|_0 \leq \widetilde{O}(\sigma_w m_1^{3/2}) \ .$$

Moreover, for each coordinate $i$ with $[D''_{w,x}]_{i,i} \neq 0$, we must have $|(D''_{w,x} W^{(0)} x)_i| \leq |(W^\rho x)_i| \leq \widetilde{O}(\sigma_w)$ with high probability, and thus

$$\|D''_{w,x} W^{(0)} x\|_2 \leq \widetilde{O}\left(\sigma_w\sqrt{\sigma_w m_1^{3/2}}\right) = \widetilde{O}\left(\sigma_w^{3/2} m_1^{3/4}\right)$$

By our assumption $\sigma_w \leq \tau_w/m_1^{1/4}$, we have

$$\|D''_{w,x}\|_0 \leq \tau_w m_1^{5/4} \quad \text{and} \quad \|D''_{w,x} W^{(0)} x\|_2 \leq \tau_w^{3/2} m_1^{3/8} \tag{C.22}$$

**Sign change in $D'_{w,x} - D''_{w,x}$.** Let $s = \|D'_{w,x} - D''_{w,x}\|_0$ be the total number of sign changes of the ReLU activations in the first hidden layer caused by further adding $W'x$. Observe that, the total number of coordinates $i$ where $|((W^{(0)} + W^\rho)x_i| \leq s'' := \frac{2\tau_w}{s^{1/4}}$ is at most $\widetilde{O}\left(s'' m_1^{3/2}\right)$ with high probability. Now, if $s \geq \widetilde{\Omega}\left(s'' m_1^{3/2}\right)$, then $W'$ must have caused the sign change of $\frac{s}{2}$ coordinates each by absolute value at least $s''$. Since $\|W'\|_{2,4} \leq \tau_w$, this is impossible because $\frac{s}{2} \times (s'')^4 > \tau_w^4$. Therefore, we must have

$$s \leq \widetilde{O}(s'' m_1^{3/2}) = \widetilde{O}\left(\frac{\tau_w}{s^{1/4}} m_1^{3/2}\right)$$
$$\implies \|D'_{w,x} - D''_{w,x}\|_0 = s \leq \widetilde{O}\left(\tau_w^{4/5} m_1^{6/5}\right) \ . \tag{C.23}$$

Next, for each coordinate $i$ where $(D'_{w,x} - D''_{w,x})_{i,i} \neq 0$, we must have $|((W^{(0)} + W^\rho)x)_i| \leq |(W'x)_i|$, and since $(W'x)_i^4$ must sum up to at most $\tau_w$ for those $s$ coordinates, we have

$$\left\|\left(D'_{w,x} - D''_{w,x}\right)(W^{(0)} + W^\rho)x\right\|_2 \leq \sqrt{\sum_{i, (D'_{w,x} - D''_{w,x})_{i,i} \neq 0} (W'x)_i^2}$$

$$\leq \sqrt{\sqrt{s \cdot \sum_{i,(D'_{w,x}-D''_{w,x})_{i,i}\neq 0} (W'x)_i^4}} \leq O\left(s^{1/4}\tau_w\right) = \widetilde{O}\left(\tau_w^{6/5}m_1^{3/10}\right)$$

$$(C.24)$$

**Sum up: First Layer.** Combining (C.22) and (C.23) and using $\tau_w \leq m_1^{-1/4}$ from assumption, we have

$$\|D'_{w,x}\|_0 \leq \widetilde{O}\left(\tau_w^{4/5}m_1^{6/5} + \tau_w m_1^{5/4}\right) \leq \widetilde{O}\left(\tau_w^{4/5}m_1^{6/5}\right).$$

By the 1-Lipschitz continuity of ReLU, we know that w.h.p.

$$\|z_2\|_2 = \left\|(D_{w,x} + D'_{w,x})(W^{(0)} + W^\rho + W')x - D_{w,x}W^{(0)}x\right\|_2$$

$$= \left\|\sigma(W^{(0)} + W^\rho + W')x) - \sigma(W^{(0)}x)\right\|_2$$

$$\leq \|(W^\rho + W')x\|_2 \leq \|W'x\|_2 + \|W^\rho x\|_2 \leq \widetilde{O}\left(\tau_w m_1^{1/4} + \sigma_w m_1^{1/2}\right) \leq \widetilde{O}\left(\tau_w m_1^{1/4}\right)$$

where we have used our assumption $\sigma_w \leq \tau_w m_1^{-1/4}$.

**Second Layer Sign Change.** The sign change in the second layer is caused by input vector

$$\text{changing from} \quad V^{(0)}z_0 \quad \text{to} \quad V^{(0)}z_0 + V^{(0)}z_2 + V^\rho(z_0 + z_2) + V'(z_0 + z_2).$$

Here, using w.h.p. $\|z\|_2 \leq \widetilde{O}(1)$, we have

$$\|V^\rho(z_0 + z_2)\|_\infty \leq \widetilde{O}(\sigma_v) \cdot (\|z_0\|_2 + \|z_2\|_2) \leq \widetilde{O}(\sigma_v)$$

$$\|V^{(0)}z_2 + V'(z_0 + z_2)\|_2 \leq \widetilde{O}(\tau_v + \|z_2\|_2) \leq \widetilde{O}\left(\tau_v + \tau_w m_1^{1/4}\right)$$

In comparison (at random initialization) we have $V^{(0)}z_0 \sim \mathcal{N}\left(0, \frac{\|z_0\|_2^2}{m_2}I\right)$ with $\|z_0\|_2 = \widetilde{\Omega}(1)$. Using a careful two-step argument (see Claim C.8), we can bound

$$\|D'_{v,x}\|_0 \leq \widetilde{O}\left(\left(\tau_v + \tau_w m_1^{1/4}\right)^{2/3}m_2 + \sigma_v m_2^{3/2}\right).$$

$$\blacksquare$$

**Part II, Diagonal Cross Term**

Recall

$$g_r(x; W, V) = a_r(D_{v,x} + D'_{v,x})\left(V(D_{w,x} + D'_{w,x})(Wx + b_1) + b_2\right)$$

$$g_r^{(b)}(x; W, V) = a_r(D_{v,x} + D'_{v,x})V(D_{w,x} + D'_{w,x})(Wx + b_1)$$

$$g_r^{(b,b)}(x; W, V) = a_r(D_{v,x} + D'_{v,x})V(D_{w,x} + D'_{w,x})Wx$$

and one can carefully check that

$$g_r(x; W^{(0)} + W^\rho + W' + \eta\boldsymbol{\Sigma}W'', V^{(0)} + V^\rho + V' + \eta V''\boldsymbol{\Sigma})$$

$$= g_r\left(x; W^{(0)} + W^\rho + W', V^{(0)} + V^\rho + V'\right) + g_r^{(b,b)}(\eta\boldsymbol{\Sigma}W'', \eta V''\boldsymbol{\Sigma})$$

$$+ \underbrace{g_r^{(b)}(x; W^{(0)} + W^\rho + W', \eta V''\boldsymbol{\Sigma}) + g_r^{(b,b)}(x; \eta\boldsymbol{\Sigma}W'', V^{(0)} + V^\rho + V')}_{\text{error terms}}$$

We consider the last two error terms.

**First error term.** The first term is

$$g_r^{(b)}(x; W^{(0)} + W^\rho + W', \eta V''\boldsymbol{\Sigma})$$

$$= \eta a_r(D_{v,x} + D'_{v,x})V''\boldsymbol{\Sigma}(D_{w,x} + D'_{w,x})((W^{(0)} + W^\rho + W')x + b_1)$$

Clearly, it has zero expectation with respect to $\boldsymbol{\Sigma}$. With high probability, we have

$$\|(D_{w,x} + D'_{w,x})((W^{(0)} + W^\rho + W')x + b_1)\|_\infty$$

$$\leq \|(W^{(0)} + W^\rho + W')x + b_1\|_\infty \leq \widetilde{O}\left(\tau_w + \sigma_w + \frac{1}{m_1^{1/2}}\right) \leq \widetilde{O}\left(\frac{1}{m_1^{1/2}}\right) \qquad (C.25)$$

and we have $\|a_r(D_{v,x} + D'_{v,x})V''\|_2 \leq \widetilde{O}\left(\tau_v m_2^{1/2}\right)$. By Fact C.9, using the randomness of $\Sigma$, with high probability

$$|g_r^{(b)}(x; W^{(0)} + W^\rho + W', \eta V''\Sigma)| = \eta\widetilde{O}\left(\frac{\tau_v \sqrt{m_2}}{\sqrt{m_1}}\right)$$

**Second error term.** We write down the second error term

$$g_r^{(b,b)}(x; \eta\Sigma W'', V^{(0)} + V^\rho + V') = \eta a_r(D_{v,x} + D'_{v,x})(V^{(0)} + V^\rho + V')(D_{w,x} + D'_{w,x})\Sigma W''x$$
$$= \eta a_r(D_{v,x} + D'_{v,x})V'(D_{w,x} + D'_{w,x})\Sigma W''x$$
$$+ \eta a_r D_{v,x}(V^{(0)} + V^\rho)(D_{w,x} + D'_{w,x})\Sigma W''x$$
$$+ \eta a_r D'_{v,x}(V^{(0)} + V^\rho)(D_{w,x} + D'_{w,x})\Sigma W''x$$

Obviously all the three terms on the right hand side have zero expectation with respect to $\Sigma$.

- For the first term, since w.h.p. $\|a_r(D_{v,x} + D'_{v,x})V'(D_{w,x} + D'_{w,x})\|_2 = \widetilde{O}(\tau_v m_2^{1/2})$ and $\|W''x\|_\infty \leq \tau_w$, by Fact C.9, using the randomness of $\Sigma$ we know that w.h.p.

$$|a_r(D_{v,x} + D'_{v,x})V'(D_{w,x} + D'_{w,x})\Sigma W''x| \leq \widetilde{O}(\tau_v m_2^{1/2}\tau_w)$$

- For the second term, since $\|W''x\|_2 \leq \tau_w m_1^{1/4}$ and w.h.p.[26]

$$\|a_r D_{v,x}(V^{(0)} + V^\rho)(D_{w,x} + D'_{w,x})\|_\infty \leq \|a_r D_{v,x}(V^{(0)} + V^\rho)\|_\infty \leq \widetilde{O}(1)$$

by Fact C.9, using the randomness of $\Sigma$ we know that w.h.p.

$$|a_r D_{v,x}(V^{(0)} + V^\rho)(D_{w,x} + D'_{w,x})\Sigma W''x| = \widetilde{O}(\tau_w m_1^{1/4})$$

- For the third term, again by $\|W''x\|_\infty \leq \tau_w$ and Fact C.9, we have: w.h.p.

$$|a_r D'_{v,x}(V^{(0)} + V^\rho)\Sigma W''x| \leq \widetilde{O}\left(\|a_r D'_{v,x}(V^{(0)} + V^\rho)\|_2\tau_w\right)$$
$$\leq \widetilde{O}\left(m_2^{1/2}\tau_w\right).$$

$\blacksquare$

**Tool**

The following are some tools used in the above proofs.

A variant of the following claim has appeared in [4].

**Claim C.8.** *Suppose $V \in \mathbb{R}^{m_2 \times m_1}$ is a random matrix with entries drawn i.i.d. from $\mathcal{N}\left(0, \frac{1}{m_2}\right)$, For all unit vector $h \in \mathbb{R}^{m_1}$, and for all $g' \in \mathbb{R}^{m_2}$ that can be written as*

$$g' = g'_1 + g'_2 \text{ where } \|g'_1\| \leq 1 \text{ and } \|g'_2\|_\infty \leq \frac{1}{4\sqrt{m_2}}.$$

*Let $D'$ be the diagonal matrix where $(D')_{k,k} = \mathbb{I}_{(Vh+g')_k \geq 0} - \mathbb{I}_{(Vh)_k \geq 0}$. Then, letting $x = D'(Vh + g')$, we have*

$$\|x\|_0 \leq O(m_2\|g'_1\|^{2/3} + m_2^{3/2}\|g'_2\|_\infty) \quad \text{and}$$
$$\|x\|_1 \leq O\left(m_2^{1/2}\|g'_1\|^{4/3} + m_2^{3/2}\|g'_2\|_\infty^2\right) \ .$$

*Proof of Claim C.8.* We first observe $g = Vh$ follows from $\mathcal{N}\left(0, \frac{\mathbf{I}}{m_2}\right)$ regardless of the choice of $h$. Therefore, in the remainder of the proof, we just focus on the randomness of $g$.

We also observe that $(D')_{j,j}$ is non-zero for some diagonal $j \in [m_2]$ only if

$$|(g'_1 + g'_2)_j| > |(g)_j| \ . \tag{C.26}$$

Let $\xi \leq \frac{1}{2\sqrt{m_2}}$ be a parameter to be chosen later. We shall make sure that $\|g_2'\|_\infty \leq \xi/2$.

- We denote by $S_1 \subseteq [m_2]$ the index sets where $j$ satisfies $|(g)_j| \leq \xi$. Since we know $(g)_j \sim \mathcal{N}(0, 1/m_2)$, we have $\mathbf{Pr}[|(g)_j| \leq \xi] \leq O\left(\xi\sqrt{m_2}\right)$ for each $j \in [m_2]$. Using Chernoff bound for all $j \in [m_2]$, we have with high probability

$$|S_1| = |\{i \in [m_2] : |(g)_j| \leq \xi\}| \leq O(\xi m_2^{3/2}) \ .$$

  Now, for each $j \in S_1$ such that $x_j \neq 0$, we must have $|x_j| = |(g + g_1' + g_2')_j| \leq |(g_1')_j| + 2\xi$ so we can calculate the $\ell_2$ norm of $x$ on $S_1$:

$$\sum_{i \in S_1} |x_j| \leq \sum_{i \in S_1}(|(g_1')_i| + 2\xi) \leq 2\xi|S_1| + \sqrt{|S_1|}\|g_1'\| \leq O(\|g_1'\|^2\sqrt{\xi}m_2^{3/4} + \xi^2 m_2^{3/2}) \ .$$

- We denote by $S_2 \subseteq [m_2] \setminus S_1$ the index set of all $j \in [m_2] \setminus S_1$ where $x_j \neq 0$. Using (C.26), we have for each $j \in S_2$:

$$|(g_1')_j| \geq |(g)_j| - |(g_2')_j| \geq \xi - \|g_2'\|_\infty \geq \xi/2$$

  This means $|S_2| \leq \frac{4\|g_1'\|^2}{\xi^2}$ . Now, for each $j \in S_2$ where $x_j \neq 0$, we know that the signs of $(g + g_1' + g_2')_j$ and $(g)_j$ are opposite. Therefore, we must have

$$|x_j| = |(g + g_1' + g_2')_j| \leq |(g_1' + g_2')_j| \leq |(g_1')_j| + \xi/2 \leq 2|(g_1')_j|$$

  and therefore

$$\sum_{j \in S_2} |x_j| \leq 2 \sum_{j \in S_2} |(g_1')_j| \leq 2\sqrt{|S_2|}\|g_1'\| \leq 4\frac{\|g_1'\|^2}{\xi}$$

From above, we have $\|x\|_0 \leq |S_1| + |S_2| \leq O(\xi m_2^{3/2} + \frac{\|g_1'\|^2}{\xi^2})$. Choosing $\xi = \max\left\{2\|g_2'\|_\infty, \Theta(\frac{\|g_1'\|^{2/3}}{m_2^{1/2}})\right\}$ we have the desired result on sparsity.

Combining the two cases, we have

$$\|x\|_1 \leq O\left(\frac{\|g_1'\|^2}{\xi} + \|g_1'\|^2\sqrt{\xi}m_2^{3/4} + \xi^2 m_2^{3/2}\right) \leq O\left(\frac{\|g_1'\|^2}{\xi} + \xi^2 m_2^{3/2}\right) \ .$$

Choosing $\xi = \max\left\{2\|g_2'\|_\infty, \Theta(\frac{\|g_1'\|^{2/3}}{m_2^{1/2}})\right\}$, we have the desired bound on Euclidean norm. $\qquad\square$

**Fact C.9.** *If $\Sigma$ is a diagonal matrix with diagonal entries randomly drawn from $\{-1, 1\}$. Then, given vectors $x, y$, with high probability*

$$|x^\top \Sigma y| \leq \widetilde{O}(\|x\|_2 \cdot \|y\|_\infty)$$

### C.2.2 Corollary 6.6: Existence After Coupling

Corollary 6.6 is a corollary to Lemma 6.2 with $g^{(0)}$ replaced with $g^{(b,b)}$. Recall that $g^{(b,b)}$ is different from $g^{(0)}$ only by the diagonal signs, namely,

$$g_r^{(0)}(x; W^*, V^*) = a_r D_{v,x} V^* D_{w,x} W^* x$$
$$g_r^{(b,b)}(x; W^*, V^*) = a_r (D_{v,x} + D_{v,x}') V^* (D_{w,x} + D_{w,x}') W^* x$$

where $D_{v,x} + D_{v,x}'$ and $D_{w,x} + D_{w,x}'$ are the diagonal sign matrices determined at $W^{(0)} + W' + W^\rho$, $V^{(0)} + V' + V^\rho$.

**Corollary 6.6** (existence after coupling)**.** *In the same setting as Lemma 6.2, perturbation matrices $W', V'$ (that may depend on the randomness of the initialization and $\mathcal{D}$) with*

$$\|W'\|_{2,4} \leq \tau_w, \|V'\|_F \leq \tau_v \ .$$

*Using parameter choices from Table 1, w.h.p. there exist $W^*$ and $V^*$ (independent of the randomness of $W^\rho, V^\rho$) satisfying*

$$\|W^*\|_{2,\infty} = \max_i \|w_i^*\|_2 \leq \frac{C_0}{m_1}, \quad \|V^*\|_{2,\infty} = \max_i \|v_i^*\|_2 \leq \frac{\sqrt{m_1}}{m_2}$$

$$\mathbb{E}_{(x,y)\sim\mathcal{D}}\left[\sum_{r=1}^{k}\left|f_r^*(x) - g_r^{(b,b)}(x; W^*, V^*)\right|\right] \leq \varepsilon,$$

$$\mathbb{E}_{(x,y)\sim\mathcal{D}}\left[L(G^{(b,b)}(x; W^*, V^*), y)\right] \leq \mathsf{OPT} + \varepsilon.$$

*Proof of Corollary 6.6.* The idea to prove Corollary 6.6 is simple. First construct $W^*$ and $V^*$ from Lemma 6.2, and then show that $g^{(0)}$ and $g^{(b,b)}$ are close using Lemma 6.5. By Lemma 6.5 and our parameter choices Table 1, we know

$$\|D'_{w,x}\|_0 \leq \widetilde{O}(\tau_w^{4/5} m_1^{6/5}) \ll O(m_1)$$

$$\|D'_{v,x}\|_0 \leq \widetilde{O}\left(\sigma_v m_2^{3/2} + \tau_v^{2/3} m_2 + \tau_w^{2/3} m_1^{1/6} m_2\right) \leq \widetilde{O}((\varepsilon/C_0)^{\Theta(1)} m_2)$$

Now, recall that Lemma 6.2 says $\|W^*\|_{2,\infty} \leq \tau_{w,\infty}$ and $\|V^*\|_{2,\infty} \leq \tau_{v,\infty}$ for $\tau_{w,\infty}\tau_{v,\infty} = \frac{C_0}{\sqrt{m_1 m_2}}$. Therefore,

$$|a_r D_{v,x} V^* D'_{w,x} W^* x| = \left|\sum_{j\in[m_2]} a_{r,j}(D_{v,x})_{j,j}\langle v_j^*, D'_{w,x} W^* x\rangle\right| \leq \widetilde{O}(m_2\tau_{v,\infty})\|D'_{w,x}W^*x\|_2$$

$$\leq \widetilde{O}(m_2\tau_{v,\infty})\sqrt{\|D'_{w,x}\|_0}\tau_{w,\infty} \ll \varepsilon$$

$$|a_r D'_{v,x} V^* D_{w,x} W^* x| = \left|\sum_{j\in[m_2]} a_{r,j}(D'_{v,x})_{j,j}\langle v_j^*, D_{w,x} W^* x\rangle\right| \leq \widetilde{O}(\|D'_{v,x}\|_0\tau_{v,\infty})\|D_{w,x}W^*x\|_2$$

$$\leq \widetilde{O}(\|D'_{v,x}\|_0\tau_{v,\infty}) \cdot O(\sqrt{m_1}\tau_{w,\infty}) \ll \varepsilon$$

In other words,

$$|g^{(b,b)}(x; W^*, V^*) - g^{(0)}(x; W^*, V^*)| \leq 2\varepsilon \ . \qquad \square$$

### C.2.3 Lemma 6.9: Smoothed Real vs Pseudo Networks

Recall

$$P_{\rho,\eta} := f_r(x; W + W^\rho + \eta\boldsymbol{\Sigma}W'', V + V^\rho + \eta V''\boldsymbol{\Sigma})$$
$$= a_r D_{v,x,\rho,\eta}\Big((V + V^\rho + \eta V''\boldsymbol{\Sigma})D_{w,x,\rho,\eta}\left((W + W^\rho + \eta\boldsymbol{\Sigma}W'')x + b_1\right) + b_2\Big)$$
$$P'_{\rho,\eta} := g_r(x; W + W^\rho + \eta\boldsymbol{\Sigma}W'', V + V^\rho + \eta V''\boldsymbol{\Sigma})$$
$$= a_r D_{v,x,\rho}\Big((V + V^\rho + \eta V''\boldsymbol{\Sigma})D_{w,x,\rho}\left((W + W^\rho + \eta\boldsymbol{\Sigma}W'')x + b_1\right) + b_2\Big).$$

**Lemma 6.9** (smoothed real vs pseudo). *There exists $\eta_0 = \frac{1}{\mathsf{poly}(m_1,m_2)}$ such that, for every $\eta \leq \eta_0$, for every fixed $x$ with $\|x\|_2 = 1$, for every $W', V', W'', V''$ that may depend on the randomness of the initialization and $x$, with*

$$\|W'\|_{2,4} \leq \tau_w, \quad \|V'\|_{2,2} \leq \tau_v, \quad \|W''\|_{2,\infty} \leq \tau_{w,\infty}, \quad \|V''\|_{2,\infty} \leq \tau_{v,\infty}$$

*we have with high probability:*

$$\mathbb{E}_{W^\rho, V^\rho}\left[\frac{|P_{\rho,\eta} - P'_{\rho,\eta}|}{\eta^2}\right] = \widetilde{O}\left(m_1\frac{\tau_{w,\infty}^2}{\sigma_w} + \frac{m_2\tau_{w,\infty}^2}{\sigma_v} + \frac{m_2}{m_1}\frac{\tau_{v,\infty}^2}{\sigma_v}\right) + O_p(\eta).$$

*where $O_p$ hides polynomial factor of $m_1, m_2$.*

*Proof of Lemma 6.9.* Since $P_{\rho,\eta}$ and $P'_{\rho,\eta}$ only differ in the sign pattern, we try to bound the (expected) output difference $P_{\rho,\eta} - P'_{\rho,\eta}$ by analyzing these sign changes. We use the same proof structure as Lemma 6.5, that is to first bound the sign changes in the first layer, and then the second

layer. One can carefully verify

$$P_{\rho,\eta} - P'_{\rho,\eta} = \underbrace{a_r D_{v,x,\rho}(V + V^\rho + \eta V''\boldsymbol{\Sigma})\big(D_{w,x,\rho,\eta} - D_{w,x,\rho}\big)((W + W^\rho + \eta\boldsymbol{\Sigma}W'')x + b_1)}_{\clubsuit}$$

$$+ \underbrace{a_r\big(D_{v,x,\rho,\eta} - D_{v,x,\rho}\big)\Big((V + V^\rho + \eta V''\boldsymbol{\Sigma})D_{w,x,\rho,\eta}((W + W^\rho + \eta\boldsymbol{\Sigma}W'')x + b_1) + b_2\Big)}_{\spadesuit} \quad .$$

We call $\clubsuit$ the output difference caused by sign change of the first layer; and $\spadesuit$ that caused by the sign change of the second layer.

**Sign Change of First Layer.**   We write

$$z = D_{w,x,\rho}\left((W + W^\rho + \eta\boldsymbol{\Sigma}W'')x + b_1\right)$$
$$z + z' = D_{w,x,\rho,\eta}\left((W + W^\rho + \eta\boldsymbol{\Sigma}W'')x + b_1\right) \quad .$$

The first observation here is that, since $\|\eta\boldsymbol{\Sigma}W''x\|_\infty \le \eta\tau_{w,\infty}$, when a coordinate $i$ has sign change (i.e. has $z'_i \ne 0$), it has value at most $|z'_i| \le \eta\tau_{w,\infty}$. In other words

$$\|z'\|_\infty \le \eta\tau_{w,\infty} \quad .$$

Since $\|\eta\boldsymbol{\Sigma}W''x\|_\infty \le \eta\tau_{w,\infty}$, and since each coordinate of $W^\rho x$ is i.i.d. from $\mathcal{N}(0, \sigma_w^2)$, we know

$$\forall i \in [m_1]: \quad \mathbf{Pr}_{W^\rho}[z'_i \ne 0] \le \widetilde{O}\left(\frac{\eta\tau_{w,\infty}}{\sigma_w}\right). \tag{C.27}$$

One consequence of (C.27) is $\mathbf{Pr}[\|z'\|_0 \ge 2] \le O_p(\eta^2)$. When $\|z'\|_0 \ge 2$, the contribution to $\clubsuit$ is $O_p(\eta)$. Multiplying them together, the total contribution to $\clubsuit$ *in expectation* is at most $O_p(\eta^3)$.

Thus, we only need to consider the case $\|z\|_0 = 1$. Let $i$ be this coordinate so that $z'_i \ne 0$. This happens with probability at most $O(\eta\tau_{w,\infty}/\sigma_w)$ for each $i \in [m_1]$. The contribution of $z'$ to $\clubsuit$ is

$$\clubsuit = a_r D_{v,x,\rho}(V + V^\rho + \eta V'')z'$$

and let us deal with the three terms separately:

- For the term $a_r D_{v,x,\rho}\eta V''z'$, it is of absolute value at most $O_p(\eta^2)$. Since $\|z_0\|_0 = 1$ happens with probability $O_p(\eta)$, the total contribution to the expected value of $\clubsuit$ is only $O_p(\eta^3)$.

- For the term $a_r D_{v,x,\rho}(V + V^\rho)z'$, we first observe that with high probability $\|a_r D_{v,x,\rho}(V + V^\rho)\|_\infty \le \widetilde{O}(\frac{\|a_r\|_2}{\sqrt{m_2}}) \le \widetilde{O}(1)$ (for a proof see Footnote 26). Therefore, given $\|z'\|_0 = 1$ and $\|z'\|_\infty \le \eta\tau_{w,\infty}$, we have that $\|a_r D_{v,x,\rho}(V + V^\rho)z'\| \le \widetilde{O}(\eta\tau_{w,\infty})$. Since this happens with probability at most $O(\eta\tau_{w,\infty}/\sigma_w) \times m_1$ —recall there are $m_1$ many possible $i \in [m_1]$— the total contribution to the expected value of $\clubsuit$ is $\widetilde{O}\left(\eta^2 m_1 \frac{\tau_w^2}{\sigma_w}\right) + O_p(\eta^3)$.

In sum, we have with high probability

$$\mathbb{E}_{W^\rho, V^\rho}[|\clubsuit|] \le \widetilde{O}\left(\eta^2 m_1 \frac{\tau_w^2}{\sigma_w}\right) + O_p(\eta^3)$$

**Sign Change of Second Layer.**   Recall that the sign of the ReLU of the second layer is changed from $D_{w,x,\rho}$ to $D_{w,x,\rho,\eta}$. Let us compare the vector inputs of these two matrices before ReLU is applied, that is

$$\delta = ((V + V^\rho + \eta V'')D_{w,x,\rho,\eta}((W + W^\rho + \eta W'')x + b_1) + b_2)$$
$$- ((V + V^\rho)D_{w,x,\rho}((W + W^\rho)x + b_1) + b_2) \quad .$$

This difference $\delta$ has the following four terms:

1. $\eta(V + V^\rho)D_{w,x,\rho}\boldsymbol{\Sigma}W''x$.

   With $\|W''x\|_\infty \le \tau_{w,\infty}$ and $\|V\|_2 \le \widetilde{O}(1)$, by Fact C.9 we know that w.h.p.

   $$\|\eta(V + V^\rho)D_{w,x,\rho}\boldsymbol{\Sigma}W''x\|_\infty \le \eta\|(V + V^\rho)D_{w,x,\rho}\|_2 \cdot \|W''x\|_\infty \cdot \widetilde{O}(1) \le \widetilde{O}(\eta\tau_{w,\infty}) \quad .$$

2. $(V + V^\rho + \eta V''\boldsymbol{\Sigma})z'$.

   This is non-zero with probability $O_p(\eta)$, and when it is non-zero, its Euclidean norm is $O_p(\eta)$.

3. $\eta V'' \mathbf{\Sigma} z$.

Since $\|z\|_\infty \le \widetilde{O}(\tau_w + \sigma_w + m_1^{-1/2}) = O(m_1^{-1/2})$ owing to (C.25), by Fact C.9, we know that w.h.p.

$$\|\eta V'' \mathbf{\Sigma} z\|_\infty \le \widetilde{O}(\eta) \cdot \max_i \|V_i''\|_2 \cdot \|z\|_\infty \le \widetilde{O}(\eta \tau_{v,\infty} m_1^{-1/2}) \ .$$

4. $\eta^2 V'' \mathbf{\Sigma} D_{w,x,\rho} \mathbf{\Sigma} W'' x$ is at most $O_p(\eta^2)$ in Euclidean norm.

In sum,

$$\|\delta\|_2 \le \left\{ \begin{array}{ll} O_p(\eta), & \text{w.p. } \le O_p(\eta); \\ \widetilde{O}(\frac{\eta \tau_{v,\infty}}{\sqrt{m_1}} + \eta \tau_{w,\infty}) + O_p(\eta^2), & \text{otherwise.} \end{array} \right.$$

Since originally each coordinate of $V^\rho z$ follows from $\mathcal{N}(0, \sigma_v \|z\|_2^2)$ with $\|z\|_2 = \widetilde{\Omega}(1)$, using a similar argument as (C.27)[27] we can bound the contribution to the *expected* value of ♠ by:

$$\mathbb{E}_{W^\rho, V^\rho}[|♠|] \le m_2 \times \widetilde{O}\left( \eta \left( \frac{1}{\sqrt{m_1}} \tau_{v,\infty} + \tau_{w,\infty} \right) \times \eta \frac{(\frac{1}{\sqrt{m_1}} \tau_{v,\infty} + \tau_{w,\infty})}{\sigma_v} \right) + O_p(\eta^3)$$

$$= \widetilde{O}\left( \eta^2 \frac{m_2 \tau_{v,\infty}^2}{\sigma_v m_1} + \eta^2 \frac{m_2 \tau_{w,\infty}^2}{\sigma_v} \right) + O_p(\eta^3) \ .$$

$\square$

### C.2.4 Lemma 6.11: Stronger Coupling

We will need the following coupling lemma when $\mathbf{\Sigma}$ is used (for Algorithm 1).

**Lemma 6.11** (stronger coupling). *Given a fixed $x$, with high probability over the random initialization and over a random diagonal matrix $\mathbf{\Sigma}$ with diagonal entries i.i.d. generated from $\{-1, 1\}$, it satisfies that for every $W', V'$ (that can depend on the initialization and $x$ but not $\mathbf{\Sigma}$) with $\|V'\|_2 \le \tau_v, \|W'\|_{2,4} \le \tau_w$ for $\tau_v \in [0, 1]$ and $\tau_w \in \left[ \frac{1}{m_1^{3/4}}, \frac{1}{m_1^{9/16}} \right]$, we have*

$$f_r(x; W^{(0)} + \mathbf{\Sigma} W', V^{(0)} + V'\mathbf{\Sigma}) = a_r D_{v,x}(V^{(0)} D_{w,x}(W^{(0)}x + b_1) + b_2) + a_r D_{v,x} V' D_{w,x} W' x$$

$$\pm \widetilde{O}\left( \tau_w^{8/5} m_1^{9/10} + \tau_w^{16/5} m_1^{9/5} \sqrt{m_2} + \frac{\sqrt{m_2}}{\sqrt{m_1}} \tau_v \right) \ .$$

*Under parameter choices Table 1, the last error term is at most $\varepsilon/k$.*

*Proof of Lemma 6.11.* For notation simplicity, let us do the proof without the bias term $b_1$ and $b_2$. The proof with them are analogous.

We use $D_{w,x}^{(0)}$ and $D_{v,x}^{(0)}$ to denote the sign matrices at random initialization $W^{(0)}, V^{(0)}$, and we let $D_{w,x}^{(0)} + D_{w,x}'$ and $D_{v,x}^{(0)} + D_{v,x}'$ be the sign matrices at $W^{(0)} + \mathbf{\Sigma} W', V^{(0)} + V'\mathbf{\Sigma}$. Define

$$z = D_{w,x}^{(0)} W^{(0)} x$$
$$z_1 = D_{w,x}^{(0)} \mathbf{\Sigma} W' x$$
$$z_2 = D_{w,x}'(W^{(0)} x + \mathbf{\Sigma} W' x).$$

Since w.h.p. each coordinate of $z$ has $\|z\|_\infty = \widetilde{O}(m_1^{-1/2})$, using Fact C.9 (so using the randomness of $\mathbf{\Sigma}$), we know with high probability

$$\|V'\mathbf{\Sigma} z\|_2^2 = \sum_{i \in [m_2]} \langle v_i', \mathbf{\Sigma} z \rangle^2 \le \sum_{i \in [m_2]} \widetilde{O}(\|v_i'\|_2^2 \cdot \|z\|_\infty^2) \le \widetilde{O}(m_1^{-1}) \sum_{i \in [m_1]} \|v_i'\|_2^2 = \widetilde{O}(\tau_v^2 m_1^{-1})$$

(C.28)

Thus we have: $\|V'\boldsymbol{\Sigma} z\|_2 \le \widetilde{O}(\tau_v m_1^{-1/2})$.

On the other hand, applying Lemma 6.5 with $W'' = 0$ and $V'' = 0$, we have $\|z_2\|_0 \le s = \widetilde{O}(\tau_w^{4/5} m_1^{6/5})$. Therefore, using the same derivation as (C.24), we can bound its Euclidean norm

$$\|z_2\|_2 = \|D'_{w,x}(W^{(0)} + \boldsymbol{\Sigma} W')x\|_2 \le \sqrt{\sum_{i,(D'_{w,x})_{i,i} \ne 0} (\boldsymbol{\Sigma} W'x)_i^2}$$

$$\le \sqrt{\sqrt{s \cdot \sum_{i,(D'_{w,x})_{i,i} \ne 0} (W'x)_i^4}} \le O\left(s^{1/4}\tau_w\right) = \widetilde{O}\left(\tau_w^{6/5} m_1^{3/10}\right)$$

Using the same derivation as (C.28), we also have w.h.p

$$\|V'\boldsymbol{\Sigma} z_2\|_2 \le \widetilde{O}\big(\|V'\|_F \|z_2\|_\infty\big) \le \widetilde{O}\big(\|V'\|_F \|z_2\|_2\big) \le \widetilde{O}\left(\tau_v \tau_w^{6/5} m_1^{3/10}\right) \ .$$

Now, recall using Cauchy-Shwartz and the 1-Lipschitz continuity of the ReLU function, we have for every $(x_i, y_i)_{i \in [m_2]}$, with high probability (over $a$):

$$\sum_{i \in [m_2]} a_{i,r}(\sigma(x_i + y_i) - \sigma(x_i)) \le \widetilde{O}(\sqrt{m_2}) \|(\sigma(x_i + y_i) - \sigma(x_i))_{i \in [m_2]}\|_2 \le \widetilde{O}(\sqrt{m_2}) \|y\|_2.$$

Therefore, we can bound that

$$f_r(x; W^{(0)} + \boldsymbol{\Sigma} W', V^{(0)} + V'\boldsymbol{\Sigma})$$

$$= \sum_{i \in [m_2]} a_{i,r} \sigma\left(\langle v_i^{(0)} + \boldsymbol{\Sigma} v_i', z + z_1 + z_2\rangle\right)$$

$$= \underbrace{\sum_{i \in [m_2]} a_{i,r} \sigma\left(\langle v_i^{(0)}, z + z_1 + z_2\rangle + \langle \boldsymbol{\Sigma} v_i', z_1\rangle\right)}_{①} \pm \widetilde{O}\left(\frac{\sqrt{m_2}}{\sqrt{m_1}}\tau_v + \sqrt{m_2}\tau_v \tau_w^{6/5} m_1^{3/10}\right) \quad \text{(C.29)}$$

To bound ①, we consider the difference between

$$① = a_r(D_{v,x}^{(0)} + D_{v,x}'') \left(V^{(0)}(z + z_1 + z_2) + V'D_{w,x}^{(0)}W'x\right)$$

$$② = a_r D_{v,x}^{(0)} \left(V^{(0)}(z + z_1 + z_2) + V'D_{w,x}^{(0)}W'x\right)$$

where $D_{v,x}''$ is the diagonal sign change matrix due to moving input from $V^{(0)}z$ to $V^{(0)}(z + z_1 + z_2) + V'D_{w,x}^{(0)}W'x$. This difference has the following three terms.

- $V^{(0)}z_1 = V^{(0)}D_{w,x}^{(0)}\boldsymbol{\Sigma} W'x$. Since $\|W'x\|_2 \le \tau_w m_1^{1/4}$ and $\max_i \|V_i^{(0)}\|_\infty \le \widetilde{O}(\frac{1}{\sqrt{m_2}})$, by Fact C.9 (thus using the randomness of $\boldsymbol{\Sigma}$), we know that w.h.p. $\|V^{(0)}z_1\|_\infty \le \widetilde{O}\left(\tau_w \frac{m_1^{1/4}}{\sqrt{m_2}}\right)$.

- $V^{(0)}z_2$. Using the sparsity of $z_2$ we know w.h.p. $\|V^{(0)}z_2\|_\infty \le \widetilde{O}(\|z_2\|_2 \sqrt{s} m_2^{-1/2}) \le \widetilde{O}\left(\frac{\tau_w^{8/5} m_1^{9/10}}{\sqrt{m_2}}\right)$

- $\|V'D_{w,x}^{(0)}W'x\|_2 \le \|V'\|_F \cdot \|W'x\|_2 \le \tau_w m_1^{1/4} \tau_v$.

Together, using $\|a_r\|_\infty \le \widetilde{O}(1)$ and invoking Claim C.8, we can bound it by:

$$|① - ②| \le \widetilde{O}\left(\left(\tau_w \frac{m_1^{1/4}}{\sqrt{m_2}} + \frac{\tau_w^{8/5} m_1^{9/10}}{\sqrt{m_2}}\right)^2 m_2^{3/2} + (\tau_w m_1^{1/4} \tau_v)^{4/3} m_2^{1/2}\right) \quad \text{(C.30)}$$

(When invoking Claim C.8, we need $\tau_w \le m_1^{-9/16}$ and $\tau_w m_1^{1/4} \tau_v \le 1$.)

Finally, from ② to our desired goal

$$③ = a_r D_{v,x}^{(0)} V^{(0)} D_{w,x}^{(0)} W^{(0)} x + a_r D_{v,x}^{(0)} V' D_{w,x}^{(0)} W'x = a_r D_{v,x}^{(0)}\left(V^{(0)}z + V'D_{w,x}^{(0)}W'x\right)$$

there are still two terms:

- Since w.h.p. $\|a_r D_{v,x}^{(0)} V^{(0)}\|_\infty = \widetilde{O}(1)$ and $z_2$ is $s$ sparse, we know that w.h.p.

$$|a_r D_{v,x}^{(0)} V^{(0)} z_2| \le \widetilde{O}(\|z_2\|_2 \sqrt{s}) \le \widetilde{O}\left(\tau_w^{8/5} m_1^{9/10}\right)$$

- Since $\|W'x\|_2 \le \tau_w m_1^{1/4}$ and w.h.p. $\|a_r D_{v,x}^{(0)} V^{(0)} D_{w,x}^{(0)}\|_\infty \le \widetilde{O}(1)$ (see Footnote 26), by Fact C.9,

$$|a_r D_{v,x}^{(0)} V^{(0)} z_1| = |a_r D_{v,x}^{(0)} V^{(0)} D_{w,x}^{(0)} \boldsymbol{\Sigma} W'x| \le \widetilde{O}(\|a_r D_{v,x}^{(0)} V^{(0)} D_{w,x}^{(0)}\|_\infty \cdot \|W'x\|_2) \le \widetilde{O}\left(\tau_w m_1^{1/4}\right)$$

In other words

$$|② - ③| \le \widetilde{O}\left(\tau_w^{8/5} m_1^{9/10} + \tau_w m_1^{1/4}\right) \tag{C.31}$$

Putting together (C.29), (C.30), (C.31), one can carefully verify that

$$
\begin{aligned}
f_r(x; W^{(0)} + \boldsymbol{\Sigma} W', V^{(0)} + V' \boldsymbol{\Sigma}) &= ① \pm \widetilde{O}\left(\frac{\sqrt{m_2}}{\sqrt{m_1}} \tau_v + \sqrt{m_2} \tau_v \tau_w^{6/5} m_1^{3/10}\right) \\
&= a_r D_{v,x}^{(0)} V^{(0)} D_{w,x}^{(0)} W^{(0)} x + a_r D_{v,x}^{(0)} V' D_{w,x}^{(0)} W'x \\
&\quad \pm \widetilde{O}\left(\tau_w^{8/5} m_1^{9/10} + \tau_w m_1^{1/4}\right) \\
&\quad \pm \widetilde{O}\left(\left(\tau_w \frac{m_1^{1/4}}{\sqrt{m_2}} + \frac{\tau_w^{8/5} m_1^{9/10}}{\sqrt{m_2}}\right)^2 m_2^{3/2} + (\tau_w m_1^{1/4} \tau_v)^{4/3} m_2^{1/2}\right) \\
&\quad \pm \widetilde{O}\left(\frac{\sqrt{m_2}}{\sqrt{m_1}} \tau_v + \sqrt{m_2} \tau_v \tau_w^{6/5} m_1^{3/10}\right) \\
&= a_r D_{v,x}^{(0)} V^{(0)} D_{w,x}^{(0)} W^{(0)} x + a_r D_{v,x}^{(0)} V' D_{w,x}^{(0)} W'x \\
&\quad \pm \widetilde{O}\left(\tau_w^{8/5} m_1^{9/10} + \tau_w^{16/5} m_1^{9/5} \sqrt{m_2} + \frac{\sqrt{m_2}}{\sqrt{m_1}} \tau_v\right)
\end{aligned}
$$

Above, we have used our parameter choices $\tau_v \in [0,1]$ and $\tau_w \in [\frac{1}{m_1^{3/4}}, \frac{1}{m_1^{9/16}}]$. $\qquad\square$

## C.3 Optimization

Recall in the first variant of SGD,

$$L'(\lambda_t, W_t, V_t) = \mathbb{E}_{W^\rho, V^\rho, (x,y) \sim \mathcal{Z}}\left[L\left(\lambda_t F\left(x; W^{(0)} + W^\rho + W_t, V^{(0)} + V^\rho + V_t\right), y\right)\right] + R(\sqrt{\lambda_t} W_t, \sqrt{\lambda_t} V_t)$$

### C.3.1 Lemma 6.7: Descent Direction

**Lemma 6.7** (descent direction)**.** *For every $\varepsilon_0 \in (0,1)$ and $\varepsilon = \frac{\varepsilon_0}{k p_1 p_2^2 \mathfrak{C}_\mathfrak{s}(\Phi, p_2 \mathfrak{C}_\mathfrak{s}(\phi,1)) \mathfrak{C}_\mathfrak{s}(\phi,1)^2}$, for every constant $\gamma \in (0, 1/4]$, consider the parameter choices in Table 1, and consider any $\lambda_t, W_t, V_t$ (that may depend on the randomness of $W^{(0)}, b^{(0)}, V^{(0)}, b^{(1)}$ and $\mathcal{Z}$) with*

$$\lambda_t \in \left((\varepsilon/\log(m_1 m_2))^{\Theta(1)}, 1\right] \quad \text{and} \quad L'(\lambda_t, W_t, V_t) \in [(1+\gamma)\mathsf{OPT} + \Omega(\varepsilon_0/\gamma), \widetilde{O}(1)]$$

*With high probability over the random initialization, there exists $W^*, V^*$ with $\|W^*\|_F, \|V^*\|_F \le 1$ such that for every $\eta \in \left[0, \frac{1}{\mathsf{poly}(m_1, m_2)}\right]$:*

$$\min\left\{\mathbb{E}_{\boldsymbol{\Sigma}}\left[L'\left(\lambda_t, W_t + \sqrt{\eta} \boldsymbol{\Sigma} W^*, V_t + \sqrt{\eta} V^* \boldsymbol{\Sigma}\right)\right], L'\left((1-\eta)\lambda_t, W_t, V_t\right)\right\}$$
$$\le (1 - \eta\gamma/4)(L'(\lambda_t, W_t, V_t)),$$

*where $\boldsymbol{\Sigma} \in \mathbb{R}^{m_1 \times m_1}$ is a diagonal matrix with each diagonal entry i.i.d. uniformly drawn from $\{\pm 1\}$.*

*Proof of Lemma 6.7.* For each output $r \in [k]$,

- Define the "pseudo function" for every $W', V'$ as

$$g_r(x; W', V') = a_r D_{v,x,\rho,t}[(V^{(0)} + V^\rho + V') D_{w,x,\rho,t}[(W^{(0)} + W^\rho + W')x + b_1] + b_2]$$

<div align="center">62</div>

where $D_{v,x,\rho,t}$ and $D_{w,x,\rho,t}$ are the diagonal sign matrices at weights $W^{(0)}+W^\rho+W_t, V^{(0)}+V^\rho+V_t$.

- Recall the real network as
$$f_r(x; W', V') = a_r D_{v,x,\rho,V'}[(V^{(0)} + V^\rho + V')D_{w,x,\rho,W'}[(W^{(0)} + W^\rho + W')x + b_1] + b_2]$$
where $D_{v,x,\rho,V'}$ and $D_{w,x,\rho,W'}$ are the diagonal sign matrices at weights $W^{(0)} + W^\rho + W', V^{(0)} + V^\rho + V'$.

- $G(x; W', V') = (g_1, \ldots, g_k)$ and $F(x; W', V') = (f_1, \ldots, f_k)$.

As a sanity check, we have $G(x; W_t, V_t) = F(x; W_t, V_t)$. (Here we slightly override the notation and use $f_r(x; W', V')$ to denote $f_r(x; W^{(0)} + W^\rho + W', V^{(0)} + V^\rho + V')$.)

By our regularizer parameters $\lambda_w, \lambda_v$ in Table 1, as long as $L'(\lambda_t, W_t, V_t) \leq \widetilde{O}(1)$, we know

$$R(\sqrt{\lambda_t}W_t, \sqrt{\lambda_t}V_t) \leq \widetilde{O}(1) \implies \|\sqrt{\lambda_t}W_t\|_{2,4} \leq \widetilde{O}(\tau'_w) \text{ and } \|\sqrt{\lambda_t}V_t\|_{2,2} \leq \widetilde{O}(\tau'_v)$$
$$\implies \|W_t\|_{2,4} \leq \tau_w \text{ and } \|V_t\|_{2,2} \leq \tau_v \tag{C.32}$$

Applying Corollary 6.6 (but scaling up the target $F^*$ by $\frac{1}{\lambda_t}$), we know that there exists $W^*, V^*$ with (here we have scaled up $W^*$ and scaled down $V^*$ both by $m_1^{0.005}$)

$$\|\sqrt{\lambda_t}W^*\|_{2,\infty} \leq \frac{C_0}{m_1^{1-0.005}}, \quad \|\sqrt{\lambda_t}V^*\|_{2,\infty} \leq \frac{m_1^{1/2-0.005}}{m_2} \quad \text{and} \tag{C.33}$$

$$\|\frac{1}{\lambda_t}F^*(x) - G^*(x)\|_2 \leq \varepsilon \quad \text{where} \quad G^*(x) := \left(a_r^\top D_{v,x,\rho,t}V^* D_{w,x,\rho,t}W^* x\right)_{r \in [k]}. \tag{C.34}$$

By our parameter choices in Table 1, this implies

$$\lambda_w \|\sqrt{\lambda_t}W^*\|_{2,4}^4 \leq \varepsilon_0 \quad \text{and} \quad \lambda_v \|\sqrt{\lambda_t}V^*\|_F^2 \leq \varepsilon_0$$
$$\|W^*\|_F \ll 1 \quad \text{and} \quad \|V^*\|_F \ll 1 .$$

Let us study an update direction

$$\widehat{W} = W_t + \sqrt{\eta}\Sigma W^*, \widehat{V} = V_t + \sqrt{\eta}V^*\Sigma.$$

**Change in Regularizer.** We first consider the change of the regularizer. We know that

$$\mathbb{E}_\Sigma\left[\|V_t + \sqrt{\eta}V^*\Sigma\|_F^2\right] = \|V_t\|_F^2 + \eta\|V^*\|_F^2 .$$

On the other hand,

$$\mathbb{E}\left[\|W_t + \sqrt{\eta}\Sigma W^*\|_{2,4}^4\right] = \sum_{i \in [m_1]} \mathbb{E}\left[\|w_{t,i} + \sqrt{\eta}\Sigma w_i^*\|_2^4\right]$$

For each term $i \in [m_1]$, we can bound

$$\|w_{t,i} + \sqrt{\eta}\Sigma w_i^*\|_2^2 = \|w_{t,i}\|_2^2 + \eta\|w_i^*\|_2^2 + 2\sqrt{\eta}\langle w_{t,i}, w_i^*\rangle(\Sigma)_{i,i}$$

and therefore

$$\mathbb{E}\left[\|w_{t,i} + \sqrt{\eta}\Sigma w_i^*\|_2^4\right] = \|w_{t,i}\|_2^4 + 4\eta\langle w_{t,i}, w_i^*\rangle^2 + \eta^2\|w_i^*\|_2^4 + 2\eta\|w_{t,i}\|_2^2\|w_i^*\|_2^2$$
$$\leq \|w_{t,i}\|_2^4 + 6\eta\|w_{t,i}\|_2^2\|w_i^*\|_2^2 + O_p(\eta^2) .$$

(Recall we use $O_p(\cdot)$ to hide polynomial factors in $m_1$ and $m_2$.) By Cauchy-Schwarz,

$$\sum_{i \in [m_1]} \|w_{t,i}\|_2^2\|w_i^*\|_2^2 \leq \sqrt{\left(\sum_{i \in [m_1]} \|w_{t,i}\|_2^4\right)\left(\sum_{i \in [m_1]} \|w_i^*\|_2^4\right)} \leq \|W_t\|_{2,4}^2\|W^*\|_{2,4}^2$$

and therefore

$$\mathbb{E}\left[\|W_t + \sqrt{\eta}\Sigma W^*\|_{2,4}^4\right] \leq \|W_t\|_2^4 + 6\eta\|W_t\|_{2,4}^2\|W^*\|_{2,4}^2 + O_p(\eta^2)$$

By $\lambda_v\|\sqrt{\lambda_t}V^*\|_F^2 \leq \varepsilon_0$, $\lambda_w\|\sqrt{\lambda_t}W^*\|_{2,4}^4 \leq \varepsilon_0$, and $\lambda_w\|\sqrt{\lambda_t}W_t\|_{2,4}^4 \leq R(\sqrt{\lambda_t}W_t, \sqrt{\lambda_t}V_t)$, we know that

$$\mathbb{E}[R(\sqrt{\lambda_t}\widehat{W}, \sqrt{\lambda_t}\widehat{V})] \leq R(\sqrt{\lambda_t}W_t, \sqrt{\lambda_t}V_t) + 4\eta\varepsilon_0 + 6\eta\sqrt{\varepsilon_0} \cdot \sqrt{R(\sqrt{\lambda_t}W_t, \sqrt{\lambda_t}V_t)}$$

$$\leq R(\sqrt{\lambda_t}W_t, \sqrt{\lambda_t}V_t) + 10\eta\varepsilon_0 + \frac{1}{4}\eta R(\sqrt{\lambda_t}W_t, \sqrt{\lambda_t}V_t) \ . \tag{C.35}$$

**Change in Objective.** We now consider the change in the objective value. Recall from (C.33) the construction of good network $W^*, V^*$ satisfies $\tau_{v,\infty} \leq \frac{1}{m_1^{999/2000}}$ and $\tau_{w,\infty} \leq \frac{1}{m_1^{999/1000}}$. For polynomially small $\eta$, by Lemma 6.9 (replacing its $\eta$ with $\sqrt{\eta}$), we have:

$$\mathbb{E}_{W^\rho, V^\rho} \left| g_r(x; \widehat{W}, \widehat{V}) - f_r(x; \widehat{W}, \widehat{V}) \right| \leq O(\varepsilon\eta) + O_p(\eta^{1.5}). \tag{C.36}$$

First we focus on $G(x; \widehat{W}, \widehat{V})$. Since $\|W_t\|_{2,4} \leq \tau_w$ and $\|V_t\|_{2,2} \leq \tau_v$ from (C.32), we can apply Lemma 6.5 to get

$$\begin{aligned}
G(x; \widehat{W}, \widehat{V}) &= G(x; W_t, V_t) + \sqrt{\eta}G'(x) + \eta G^*(x) \\
&= F(x; W_t, V_t) + \sqrt{\eta}G'(x) + \eta G^*(x)
\end{aligned} \tag{C.37}$$

where $G'(x)$ is from Lemma 6.5 and satisfies $\mathbb{E}_{\boldsymbol{\Sigma}}[G'(x)] = 0$ and w.h.p. $\|G'(x)\|_2 \leq \varepsilon$; and $G^*(x)$ is from (C.34).

Combining (C.36) and (C.37), we know that for every fixed $x, y$ in the support of distribution $\mathcal{Z}$:

$$\begin{aligned}
&\mathbb{E}_{W^\rho, V^\rho, \boldsymbol{\Sigma}}[L(\lambda_t F(x; \widehat{W}, \widehat{V}), y)] \\
&\leq \mathbb{E}_{W^\rho, V^\rho, \boldsymbol{\Sigma}}[L(\lambda_t G(x; \widehat{W}, \widehat{V}), y)] + O(\eta\varepsilon) + O_p(\eta^{1.5}). \\
&\overset{\text{①}}{\leq} \mathbb{E}_{W^\rho, V^\rho} \left[ L(\lambda_t \mathbb{E}_{\boldsymbol{\Sigma}}[G(x; \widehat{W}, \widehat{V}), y]) \right] + \mathbb{E}_{W^\rho, V^\rho, \boldsymbol{\Sigma}} \left\| \sqrt{\eta}G'(x) \right\|^2 + O(\eta\varepsilon) + O_p(\eta^{1.5}). \\
&\overset{\text{②}}{\leq} \mathbb{E}_{W^\rho, V^\rho} L \left( \lambda_t G(x; W_t, V_t) + \lambda_t \eta G^*(x), y \right) + O(\eta\varepsilon) + O_p(\eta^{1.5}). \\
&\overset{\text{③}}{\leq} \mathbb{E}_{W^\rho, V^\rho} L \left( \lambda_t G(x; W_t, V_t) + \eta F^*(x), y \right) + O(\eta\varepsilon) + O_p(\eta^{1.5}). \\
&= \mathbb{E}_{W^\rho, V^\rho} L \left( \lambda_t F(x; W_t, V_t) + \eta F^*(x), y \right) + O(\eta\varepsilon) + O_p(\eta^{1.5}). 
\end{aligned} \tag{C.38}$$

Above, ① uses the 1-Lipschitz smoothness of $L$ which implies

$$\mathbb{E}[L(v)] \leq L(\mathbb{E}[v]) + \mathbb{E}[\|v - \mathbb{E}[v]\|^2]$$

and $\mathbb{E}_{\boldsymbol{\Sigma}}[G(\widehat{W}, \widehat{V}, x)] = G(W_t, V_t, x) + \eta G^*(x)$. Inequality ② also uses $\mathbb{E}_{\boldsymbol{\Sigma}}[G(x; \widehat{W}, \widehat{V})] = G(x; W_t, V_t) + \eta G^*(x)$. Inequality ③ uses (C.34) and the 1-Lipschitz continuity of $L$.

Next, by convexity of the loss function, we have

$$\begin{aligned}
L \left( \lambda_t F(x; W_t, V_t) + \eta F^*(x), y \right) &= L \left( (1-\eta)(1-\eta)^{-1}\lambda_t F(x; W_t, V_t) + \eta F^*(x), y \right) \\
&\leq (1-\eta) \left( L((1-\eta)^{-1}\lambda_t F(x; W_t, V_t), y) \right) + \eta L(F^*(x), y)
\end{aligned} \tag{C.39}$$

For sufficiently small $\eta$, we know that

$$L((1-\eta)^{-1}\lambda_t F(x; W_t, V_t), y) + L((1-\eta)\lambda_t F(x; W_t, V_t), y) \leq 2L \left( \lambda_t F(x; W_t, V_t), y \right) + O_p(\eta^2)$$

Putting this into (C.39), we have

$$\begin{aligned}
&L \left( \lambda_t F(x; W_t, V_t) + \eta F^*(x), y \right) \\
&\leq (1-\eta) \left( 2L \left( \lambda_t F(x; W_t, V_t), y \right) - L((1-\eta)\lambda_t F(x; W_t, V_t), y) \right) + \eta L(F^*(x), y) + O_p(\eta^2)
\end{aligned} \tag{C.40}$$

**Putting All Together.** Let us denote

$$\begin{aligned}
c_1 &= \mathbb{E}_{W^\rho, V^\rho, \boldsymbol{\Sigma}, (x,y) \sim \mathcal{Z}}[L(\lambda_t F(x; \widehat{W}, \widehat{V}), y)] \\
c_2 &= \mathbb{E}_{W^\rho, V^\rho, (x,y) \sim \mathcal{Z}}[L((1-\eta)\lambda_t F(x; W_t, V_t), y)] \\
c_3 &= \mathbb{E}_{W^\rho, V^\rho, (x,y) \sim \mathcal{Z}}[L(\lambda_t F(x; W_t, V_t), y)] \\
c_1' &= \mathbb{E}_{\boldsymbol{\Sigma}} \left[ L' \left( \lambda_t, \widehat{W}, \widehat{V} \right) \right] = c_1 + \mathbb{E}_{\boldsymbol{\Sigma}}[R(\sqrt{\lambda_t}\widehat{W}, \sqrt{\lambda_t}\widehat{V})] \\
c_2' &= L'((1-\eta)\lambda_t, W_t, V_t) = c_2 + R(\sqrt{(1-\eta)\lambda_t}W_t, \sqrt{(1-\eta)\lambda_t}V_t) \\
c_3' &= L'(\lambda_t, W_t, V_t) = c_3 + R(\sqrt{\lambda_t}W_t, \sqrt{\lambda_t}V_t)
\end{aligned}$$

The objective growth inequalities (C.38) and (C.40) together imply
$$c_1 \le (1 - \eta)(2c_3 - c_2) + \eta(\mathsf{OPT} + O(\varepsilon)) + O_p(\eta^{1.5}) \qquad \text{(C.41)}$$
The regularizer growth inequality (C.35) implies
$$c_1' - c_1 \le (1 + \frac{\eta\gamma}{4})(c_3' - c_3) + O(\eta\varepsilon_0/\gamma)$$

and therefore
$$
\begin{aligned}
c_1' - c_1 - ((1 - \eta)(2(c_3' - c_3) - (c_2' - c_2))) &\le c_1' - c_1 - ((1 - \eta)(2(c_3' - c_3) - (1 - \eta)(c_3' - c_3))) \\
&\le c_1' - c_1 - (1 - \eta^2)(c_3' - c_3) \\
&\le \frac{\eta\gamma}{4}c_3' + O(\eta\varepsilon_0/\gamma) + O(\eta^2) \ . \qquad \text{(C.42)}
\end{aligned}
$$

Putting (C.41) and (C.42) together we have
$$c_1' \le (1 - \eta)(2c_3' - c_2') + \frac{\eta\gamma}{4}c_3' + \eta(\mathsf{OPT} + O(\varepsilon_0/\gamma)) + O_p(\eta^{1.5})$$

Multiplying $\frac{1}{2(1-\eta)}$ on both sides, we have:
$$\frac{1}{2}(1 - \eta)^{-1}c_1' + \frac{1}{2}c_2' \le c_3' + \frac{\eta\gamma}{8}c_3' + \eta\frac{1}{2}\mathsf{OPT} + O(\eta\varepsilon_0/\gamma) + O_p(\eta^{1.5})$$

Therefore,
$$\left(\frac{1}{2}(1 - \eta)^{-1} + \frac{1}{2}\right)\min\{c_1', c_2'\} \le \left(1 + \frac{\eta\gamma}{8}\right)c_3' + \eta\frac{1}{2}\mathsf{OPT} + O(\eta\varepsilon_0/\gamma) + O_p(\eta^{1.5})$$

and this implies that
$$\min\{c_1', c_2'\} \le \left(1 - \eta\frac{1}{2} + \frac{\eta\gamma}{8}\right)c_3' + \eta\frac{1}{2}\mathsf{OPT} + O(\eta\varepsilon_0/\gamma) + O_p(\eta^{1.5})$$

Therefore, as long as $c_3' \ge (1 + \gamma)\mathsf{OPT} + \Omega(\varepsilon_0/\gamma)$ and $\gamma \in [0, 1]$, we have:
$$\min\{c_1', c_2'\} \le (1 - \eta\gamma/4)c_3'$$

This completes the proof. $\qquad\qquad\qquad\qquad\qquad\qquad\qquad\qquad\qquad\qquad\qquad\square$

### C.3.2 Lemma 6.8: Convergence

**Lemma 6.8** (convergence). *In the setting of Theorem 3, with probability at least $99/100$, Algorithm 2 (the first SGD variant) converges in $TT_w = \mathsf{poly}(m_1, m_2)$ iterations to a point*
$$L'(\lambda_T, W_T, V_T) \le (1 + \gamma)\mathsf{OPT} + \varepsilon_0.$$

*Proof of Lemma 6.8.* For the first variant of SGD, note that there are $T = \Theta(\eta^{-1}\log\frac{\log(m_1m_2)}{\varepsilon_0})$ rounds of weight decay, which implies that $\lambda_t \ge (\varepsilon/\log(m_1m_2))^{O(1)}$ is always satisfied (because $\gamma$ is a constant). By Lemma 6.7, we know that as long as $L' \in [(1+\gamma)\mathsf{OPT} + \Omega(\varepsilon_0/\gamma), \widetilde{O}(1)]$, then there exists $\|W^*\|_F, \|V^*\|_F \le 1$ such that either
$$\mathbb{E}_{\mathbf{\Sigma}}\left[L'\left(\lambda_{t-1}, W_t + \sqrt{\eta}\mathbf{\Sigma}W^*, V_t + \sqrt{\eta}V^*\mathbf{\Sigma}\right)\right] \le (1 - \eta\gamma/4)(L'(\lambda_{t-1}, W_t, V_t))$$
or
$$L'((1 - \eta)\lambda_{t-1}, W_t, V_t) \le (1 - \eta\gamma/4)(L'(\lambda_{t-1}, W_t, V_t))$$

In the first case, recall $L'$ is $B = \mathsf{poly}(m_1, m_2)$ second-order smooth,[28] by Fact A.8, it satisfies ($\lambda_{t-1}$ is fixed and the Hessian is with respect to $W$ and $V$):
$$\lambda_{\min}\left(\nabla^2 L'(\lambda_{t-1}, W_t, V_t)\right) < -1/(m_1m_2)^8 \ .$$
On the other hand, for every $t \ge 1$, since $W_t$ is the output of noisy SGD, by the escape saddle point theorem of [19] (stated in Lemma A.9), we know with probability at least $1 - p$ it satisfies $\lambda_{\min}\left(\nabla^2 L'(\lambda_{t-1}, W_t, V_t)\right) > -1/(m_1m_2)^8$ . Choosing $p = \frac{1}{1000T}$, we know with probability at

least 0.999, this holds for all rounds $t = 1, 2, \dots, T$. In other words, for all rounds $t = 1, 2, \dots, T$, the first case cannot happen and therefore as long as $L' \geq (1 + \gamma)\mathsf{OPT} + \Omega(\varepsilon_0/\gamma)$,

$$L'((1 - \eta)\lambda_{t-1}, W_t, V_t) \leq (1 - \eta\gamma/4)(L'(\lambda_{t-1}, W_t, V_t)).$$

On the other hand, for each round $t = 0, 1, \dots, T - 1$, as long as $L' \leq \widetilde{O}(1)$, by Lemma A.9, it holds that

$$L'(\lambda_t, W_{t+1}, V_{t+1}) \leq L'(\lambda_t, W_t, V_t) + (m_1 m_2)^{-1} .$$

Since initially, $L'(\lambda_1, W_0, V_0) \leq \widetilde{O}(1)$ w.h.p., we have w.h.p. $L' \leq \widetilde{O}(1)$ throughout the process. Since $\gamma$ is a constant, after $T = \Theta(\eta^{-1} \log \frac{\log m}{\varepsilon_0})$ rounds of weight decay, we have $L' \leq (1 + \gamma)\mathsf{OPT} + O(\varepsilon_0/\gamma)$. Since $\gamma$ is a constant, re-scaling $\varepsilon_0$ down by a constant factor finishes the proof. $\qquad\square$

*Remark* C.10. For the second variant of the SGD, note that

$$\boldsymbol{\Sigma}_1 W_t + \sqrt{\eta}\boldsymbol{\Sigma}_1\boldsymbol{\Sigma} W^*, \quad V_t\boldsymbol{\Sigma}_1 + \sqrt{\eta}V^*\boldsymbol{\Sigma}_1\boldsymbol{\Sigma}$$

satisfies that $\boldsymbol{\Sigma}_1\boldsymbol{\Sigma}$ is still a diagonal matrix with each diagonal entry i.i.d. $\{\pm 1\}$. Thus, the convergence results from Lemma 6.7 and Lemma 6.8 still apply, if we replace $L'$ with

$$L''(\lambda_t, W_t, V_t) = \mathbb{E}_{W^\rho, V^\rho, \boldsymbol{\Sigma}, (x,y)\sim\mathcal{Z}} \left[ L\left(\lambda_t F\left(x; W^{(0)} + W^\rho + \boldsymbol{\Sigma}W_t, V^{(0)} + V^\rho + V_t\boldsymbol{\Sigma}\right), y\right) \right]$$
$$+ R(\sqrt{\lambda_t}W_t, \sqrt{\lambda_t}V_t).$$

## C.4  Generalization

### C.4.1  Lemma 6.10: Generalization For $L_R = L_1$

We derive a very crude Rademacher complexity bound for our three-layer neural network. We have not tried to tighten the polynomial dependency in $m_1$ and $m_2$.

**Lemma 6.10** (generalization for $L_R = L_1$). *For every $\tau'_v, \tau'_w \geq 0$, every $\sigma_v \in (0, 1/\sqrt{m_2}]$, w.h.p. for every $r \in [k]$ and every $N \geq 1$, the empirical Rademacher complexity is bounded by*

$$\frac{1}{N}\mathbb{E}_{\xi\in\{\pm 1\}^N}\left[\sup_{\|V'\|_F \leq \tau'_v, \|W'\|_{2,4} \leq \tau'_w} \sum_{i\in[N]} \xi_i f_r(x_i; W^{(0)} + W^\rho + W', V^{(0)} + V^\rho + V')\right]$$
$$\leq \widetilde{O}\left(\frac{\tau'_w m_1\sqrt{m_2} + \tau'_v m_2}{\sqrt{N}} + \frac{\tau'_v\sqrt{m_1 m_2 \tau'_w(1/\sqrt{m_1} + \tau'_w)}}{N^{1/4}}\right) .$$

*Proof.* Let $W = W^{(0)} + W^\rho$ and $V = V^{(0)} + V^\rho$ for notation simplicity. Recall the input to the $j$-th neuron on the second layer is

$$n_j(x; W + W', V + V') = \sum_{i\in[m_1]} (v_{j,i} + v'_{j,i})\sigma\left(\langle w_i + w'_i, x\rangle + b^{(0)}_{1,i}\right) + b^{(0)}_{2,j}$$

For analysis purpose, let us truncate $V'$ by zeroing out all of its large coordinates. Namely, $V'' \in \mathbb{R}^{m_2 \times m_1}$ is defined so that $V''_{i,j} = V'_{i,j}$ if $|V'_{i,j}| \leq \delta$ and $V''_{i,j} = 0$ otherwise. At most $(\tau'_v)^2/\delta^2$ coordinates will be zeroed out because $\|V'\|_F \leq \tau'_v$. Since for each $x$ in the training set, we have with high probability $|\sigma(\langle w_i + w'_i, x\rangle + b^{(0)}_{1,i})| \leq \widetilde{O}(\frac{1}{\sqrt{m_1}} + \|w'_i\|_2) \leq \widetilde{O}(\frac{1}{\sqrt{m_1}} + \tau'_w)$, and since $\|a_r\|_\infty \leq \widetilde{O}(1)$, it satisfies

$$|f_r(x; W + W', V + V') - f_r(x; W + W', V + V'')| \leq \widetilde{O}(\frac{1}{\sqrt{m_1}} + \tau'_w) \times \frac{(\tau'_v)^2}{\delta} . \qquad \text{(C.43)}$$

We now bound the Rademacher complexity of $f_r(x_i; W + W', V + V'')$ in the following simple steps.

- $\{x \mapsto \langle w'_i, x\rangle \mid \|w'_i\|_2 \leq \tau'_w\}$ has Rademacher complexity $O(\frac{\tau'_w}{\sqrt{N}})$ by Proposition A.12a.

- $\{x \mapsto \langle w_i + w_i', x \rangle + b_i \mid \|w_i'\|_2 \leq \tau_w'\}$ has Rademacher complexity $O(\frac{\tau_w'}{\sqrt{N}})$ because singleton class has zero complexity and adding it does not affect complexity by Proposition A.12c.

- $\{x \mapsto n_j(x; W + W', V + V'') \mid \|W'\|_{2,\infty} \leq \tau_w' \wedge \|v_j''\|_1 \leq \sqrt{m_1}\tau_v' \wedge \|v_j''\|_\infty \leq \delta\}$ has Rademacher complexity $O(\frac{\tau_w'}{\sqrt{N}}) \cdot \widetilde{O}(\frac{m_1}{\sqrt{m_2}} + \delta m_1) + \widetilde{O}(\frac{\tau_v'}{\sqrt{N}})$. This is because w.h.p. $\|v_j\|_1 \leq \widetilde{O}(\frac{m_1}{\sqrt{m_2}})$ so we can apply Proposition A.12e, and because $\|v_j''\|_1 \leq \sqrt{m_1}\tau_v'$ and $\|v_j''\|_\infty \leq \delta$ so we can apply Proposition A.12f by choosing $f_i^{(0)}(x) = \langle w_i, x \rangle + b_i$ which satisfies $|f_i^{(0)}(x)| \leq \widetilde{O}(\frac{1}{\sqrt{m_1}})$ w.h.p.

- $\{x \mapsto f_r(x; W + W', V + V'') \mid \|W'\|_{2,\infty} \leq \tau_w' \wedge \forall j \in [m_2], \|v_j''\|_1 \leq \sqrt{m_1}\tau_v'\}$ has Rademacher complexity $\left( O(\frac{\tau_w'}{\sqrt{N}}) \cdot \widetilde{O}(\frac{m_1}{\sqrt{m_2}} + \delta m_1) + \widetilde{O}(\frac{\tau_v'}{\sqrt{N}}) \right) \cdot \widetilde{O}(m_2)$ because w.h.p. $\|a_r\|_1 \leq \widetilde{O}(m_2)$ and Proposition A.12e.

Finally, noticing that $\|W'\|_{2,4} \leq \tau_w'$ implies $\|W'\|_{2,\infty} \leq \tau_w'$ and $\|V''\|_F \leq \tau_v'$ implies $\|v_j''\|_1 \leq \sqrt{m_1}\|v_j''\|_2 \leq \sqrt{m_1}\tau_v'$, we finish the proof that the Rademacher complexity of $f_r(x_i; W + W', V + V'')$ is at most

$$\widetilde{O}\left( \frac{\tau_w' m_1 \sqrt{m_2} + \tau_v' m_2}{\sqrt{N}} + \frac{\tau_w' m_1 m_2 \delta}{\sqrt{N}} \right) \ .$$

Combining this with (C.43), and tuning the best choice of $\delta$ gives the desired result. $\qquad \square$

### C.4.2 Lemma 6.12: Generalization For $L_R = L_2$

**Lemma 6.12** (generalization for $L_R = L_2$). *For every $\tau_v' \in [0, 1]$, $\tau_w' \in \left[ \frac{1}{m_1^{3/4}}, \frac{1}{m_1^{9/16}} \right]$, every $\sigma_w \in [0, 1/\sqrt{m_1}]$ and $\sigma_v \in [0, 1/\sqrt{m_2}]$, w.h.p. for every $r \in [k]$ and every $N \geq 1$, we have by our choice of parameters in Lemma 6.7, the empirical Rademacher complexity is bounded by*

$$\frac{1}{N}\mathbb{E}_{\xi \in \{\pm 1\}^N}\left[ \sup_{\|V'\|_F \leq \tau_v', \|W'\|_{2,4} \leq \tau_w'} \left| \sum_{i \in [N]} \xi_i \mathbb{E}_{\mathbf{\Sigma}}[f_r(x_i; W^{(0)} + W^\rho + \mathbf{\Sigma}W', V^{(0)} + V^\rho + V'\mathbf{\Sigma})] \right| \right]$$

$$\leq \widetilde{O}\left( \frac{\tau_w' \tau_v' m_1^{1/4} \sqrt{m_2}}{\sqrt{N}} + \left( (\tau_w')^{8/5} m_1^{9/10} + (\tau_w')^{16/5} m_1^{9/5} \sqrt{m_2} + \frac{\sqrt{m_2}}{\sqrt{m_1}} \tau_v' \right) \right).$$

*Under parameter choices in Table 1, this is at most $\widetilde{O}\left( \frac{\tau_w' \tau_v' m_1^{1/4} \sqrt{m_2}}{\sqrt{N}} \right) + \varepsilon/k$.*

*Proof of Lemma 6.12.* Let $W = W^{(0)} + W^\rho$ and $V = V^{(0)} + V^\rho$ for notation simplicity. Applying Lemma 6.11 (with $\tau_w$ chosen as $\tau_w'$), we know that by our choice of parameters,

$$f_r(x; W + \mathbf{\Sigma}W', V + V'\mathbf{\Sigma}) = a_r D_{v,x}(V D_{w,x}(Wx + b_1) + b_2) + a_r D_{v,x} V' D_{w,x} W' x \pm B$$

for $B = \widetilde{O}\left( (\tau_w')^{8/5} m_1^{9/10} + (\tau_w')^{16/5} m_1^{9/5} \sqrt{m_2} + \frac{\sqrt{m_2}}{\sqrt{m_1}} \tau_v' \right)$.

We bound the Rademacher complexity of the right hand side. It consists of three terms and the Rademacher complexity is the summation of the three (recall Proposition A.12c).

The first term does not depend on $W'$ or $V'$ so has Rademacher complexity zero.

The third term has Rademacher complexity at most $B$.

The second term corresponds to the function class

$$\mathcal{F} = \{x \mapsto a_r D_{v,x} V' D_{w,x} W' x \mid \|V'\|_F \leq \tau_v', \|W'\|_{2,4} \leq \tau_w'\}$$

We calculate its Rademacher complexity as follows.

$$\sup_{\|W'\|_{2,4} \leq \tau_w', \|V'\|_F \leq \tau_v'} \left| \sum_{j \in [N]} \xi_j a_r D_{v,x_j} V' D_{w,x_j} W' x_j \right|$$

$$= \sup_{\|W'\|_{2,4}\leq\tau_w', \|V'\|_F\leq\tau_v'} \left| \mathbf{Tr}\Big( \sum_{j\in[N]} \xi_j x_j a_r D_{v,x_j} V' D_{w,x_j} W' \Big) \right|$$

$$\leq \sup_{\|W'\|_{2,4}\leq\tau_w', \|V'\|_F\leq\tau_v'} \left\| \sum_{j\in[N]} \xi_j x_j a_r D_{v,x_j} V' D_{w,x_j} \right\|_F \|W'\|_F$$

$$\leq \tau_w' m_1^{1/4} \sup_{\|V'\|_F\leq\tau_v'} \left\| \sum_{j\in[N]} \xi_j x_j a_r D_{v,x_j} V' D_{w,x_j} \right\|_F$$

Let us bound the last term entry by entry. Let $[D_{w,x_j}]_q$ denote the $q$-th column of $D_{w,x_j}$, $[V']_q$ the $q$-th column of $V'$.

$$\left\| \sum_{j\in[N]} \xi_j x_j a_r D_{v,x_j} V' D_{w,x_j} \right\|_F = \sqrt{ \sum_{p\in[d],q\in[m_1]} \left( \sum_{j\in[N]} \xi_j x_{j,p} a_r D_{v,x_j} V' [D_{w,x_j}]_q \right)^2 }$$

$$= \sqrt{ \sum_{p\in[d],q\in[m_1]} \left( \sum_{j\in[N]} \xi_j [D_{w,x_j}]_{q,q} x_{j,p} a_r D_{v,x_j} [V']_q \right)^2 }$$

$$\leq \sqrt{ \sum_{p\in[d],q\in[m_1]} \left\| \sum_{j\in[N]} \xi_j [D_{w,x_j}]_{q,q} x_{j,p} a_r D_{v,x_j} \right\|^2 \|[V']_q\|^2 }$$

For random $\xi_j$, we know that w.h.p. over the randomness of $\xi_j$ (notice that we can do so because $D_{w,x_j}$ only depends on $W^{(0)}$ but not on $W'$, so we can take randomness argument on $\xi$),

$$\left\| \sum_{j\in[N]} \xi_j [D_{w,x_j}]_{q,q} x_{j,p} a_r D_{v,x_j} \right\|^2 \leq \widetilde{O}\left( \|a_r\|_2^2 \sum_{j\in[N]} x_{j,p}^2 \right),$$

Thus,

$$\mathbb{E}_\xi \sup_{\|V'\|_F\leq\tau_v'} \sqrt{ \sum_{p\in[d],q\in[m_1]} \left\| \sum_{j\in[N]} \xi_j [D_{w,x_j}]_{q,q} x_{j,p} a_r D_{v,x_j} \right\|^2 \|[V']_q\|^2 }$$

$$\leq \widetilde{O}\left( \sup_{\|V'\|_F\leq\tau_v'} \|a_r\|_2 \sqrt{ \sum_{p\in[d],q\in[m_1]} \sum_{j\in[N]} x_{j,p}^2 \|[V']_q\|^2 } \right)$$

$$\leq \widetilde{O}\left( \|a_r\|_2 \sup_{\|V'\|_F\leq\tau_v'} \sqrt{ \sum_{q\in[m_1]} \sum_{j\in[N]} \|[V']_q\|^2 } \right)$$

$$\leq \widetilde{O}\left( \tau_v' \sqrt{m_2 N} \right).$$

This implies $\widehat{\mathfrak{R}}(\mathcal{X};\mathcal{F}) \leq \widetilde{O}\big(\frac{\tau_w'\tau_v' m_1^{1/4}\sqrt{m_2}}{\sqrt{N}}\big)$ and finishes the proof. $\qquad\square$

## C.5 Final Theorems

### C.5.1 Theorem 3: First SGD Variant

**Theorem 3.** *Consider Algorithm 2. For every* constant *$\gamma \in (0, 1/4]$, every $\varepsilon_0 \in (0, 1/100]$, every $\varepsilon = \frac{\varepsilon_0}{kp_1 p_2^2 \mathfrak{C}_\mathfrak{s}(\Phi, p_2\mathfrak{C}_\mathfrak{s}(\phi,1))\mathfrak{C}_\mathfrak{s}(\phi,1)^2}$, there exists*

$$M = \mathsf{poly}\left( \mathfrak{C}_\varepsilon(\Phi, \sqrt{p_2}\mathfrak{C}_\varepsilon(\phi,1)), \frac{1}{\varepsilon} \right)$$

*such that for every $m_2 = m_1 = m \geq M$, and properly set $\lambda_w, \lambda_v, \sigma_w, \sigma_v$ in Table 1, as long as*

$$N \geq \widetilde{\Omega}(Mm^{3/2})$$

*there is a choice $\eta = 1/\mathsf{poly}(m_1, m_2)$ and $T = \mathsf{poly}(m_1, m_2)$ such that with probability $\geq 99/100$,*

$$\mathbb{E}_{(x,y)\sim\mathcal{D}}L(\lambda_T F(x; W_T^{(out)}, V_T^{(out)}), y) \leq (1+\gamma)\mathsf{OPT} + \varepsilon_0.$$

*Proof of Theorem 3.* For notation simplicity let $L_F(z; \lambda, W, V) := L(\lambda F(x; W, V), y)$ for $z = (x, y)$. For the first SGD variant, recall from Lemma 6.8 that

$$\mathbb{E}_{z\sim\mathcal{Z}, W^\rho, V^\rho} L_F(z; \lambda_T, W^{(0)} + W_T + W^\rho, V^{(0)} + V_T + V^\rho) + R(\sqrt{\lambda_T}W_T, \sqrt{\lambda_T}V_T) \leq (1+\gamma)\mathsf{OPT} + \varepsilon_0.$$

Since $L_F(z; \lambda_T, W^{(0)} + W_T + W^\rho, V^{(0)} + V_T + V^\rho) \in [0, \widetilde{O}(1)]$ w.h.p., by randomly sampling $\widetilde{O}(1/\varepsilon_0^2)$ many $W^{\rho,j}, V^{\rho,j}$, we know w.h.p. there exists one $j^*$ with

$$\mathbb{E}_{z\in\mathcal{Z}} L_F(z; \lambda_T, W^{(0)} + W^{\rho,j^*} + W_T, V^{(0)} + V^{\rho,j^*} + V_T) \leq (1+\gamma)\mathsf{OPT} + 2\varepsilon_0$$

Now, recall that

$$\|\sqrt{\lambda_T}W_T\|_{2,4} \leq \tau_w' \text{ and } \|\sqrt{\lambda_T}V_T\|_F \leq \tau_v'$$

due to our regularizer (see (C.32)). By simple spectral norm bound, we also know w.h.p. for every $(x, y) \sim \mathcal{D}$ and $j$,[29]

$$\left|L_F(z; \lambda_T, W^{(0)} + W^{\rho,j} + W_T, V^{(0)} + V^{\rho,j} + V_T)\right| \leq \widetilde{O}(\sqrt{km_2}) \ .$$

Therefore, we can plug in the Rademacher complexity from Lemma 6.10 and $b = \widetilde{O}(\sqrt{km_2})$ into standard generalization statement Corollary A.11. Using our choices of $\tau_w'$ and $\tau_v'$ from Table 1 as well as $m_1 = m_2$, this bound implies as long as $N \geq \widetilde{O}(M(m_2)^{3/2})$, w.h.p. for every pair $W^{\rho,j}, V^{\rho,j}$, it holds

$$\mathbb{E}_{z\in\mathcal{D}} L_F(z; \lambda_T, W^{(0)} + W^{\rho,j} + W_T, V^{(0)} + V^{\rho,j} + V_T)$$
$$\leq \mathbb{E}_{z\in\mathcal{Z}} L_F(z; \lambda_T, W^{(0)} + W^{\rho,j} + W_T, V^{(0)} + V^{\rho,j} + V_T) + \varepsilon_0$$

Together, we have

$$\mathbb{E}_{z\in\mathcal{D}} L_F(z; \lambda_T, W^{(0)} + W^{\rho,j^*} + W_T, V^{(0)} + V^{\rho,j^*} + V_T) \leq (1+\gamma)\mathsf{OPT} + 3\varepsilon_0$$

$\square$

### C.5.2 Theorem 2: Second SGD Variant

**Theorem 2.** *Consider Algorithm 1. For every $\mathsf{constant}$ $\gamma \in (0, 1/4]$, every $\varepsilon_0 \in (0, 1/100]$, every $\varepsilon = \frac{\varepsilon_0}{kp_1 p_2^2 \mathfrak{C}_{\mathfrak{s}}(\Phi, p_2 \mathfrak{C}_{\mathfrak{s}}(\phi, 1))\mathfrak{C}_{\mathfrak{s}}(\phi, 1)^2}$, there exists*

$$M = \mathsf{poly}\left(\mathfrak{C}_\varepsilon(\Phi, \sqrt{p_2}\mathfrak{C}_\varepsilon(\phi, 1)), \frac{1}{\varepsilon}\right)$$

*such that for every $m_2 = m_1 = m \geq M$, and properly set $\lambda_w, \lambda_v, \sigma_w, \sigma_v$ in Table 1, as long as*

$$N \geq \widetilde{\Omega}\left(\left(\frac{\mathfrak{C}_\varepsilon(\Phi, \sqrt{p_2}\mathfrak{C}_\varepsilon(\phi, 1)) \cdot \mathfrak{C}_\varepsilon(\phi, 1) \cdot \sqrt{p_2}p_1 k^2}{\varepsilon_0}\right)^2\right)$$

*there is a choice $\eta = 1/\mathsf{poly}(m_1, m_2)$ and $T = \mathsf{poly}(m_1, m_2)$ such that with probability $\geq 99/100$,*

$$\mathbb{E}_{(x,y)\sim\mathcal{D}}L(\lambda_T F(x; W_T^{(out)}, V_T^{(out)}), y) \leq (1+\gamma)\mathsf{OPT} + \varepsilon_0.$$

*Proof of Theorem 2.* For notation simplicity let $L_F(z; \lambda, W, V) := L(\lambda F(x; W, V), y)$ for $z = (x, y)$. Recall from Remark C.10 that Lemma 6.8 still works in this setting, so we have

$$\mathbb{E}_{W^\rho, V^\rho, \boldsymbol{\Sigma}, z \sim \mathcal{Z}} \left[ L_F \left( z; \lambda_T, W^{(0)} + W^\rho + \boldsymbol{\Sigma} W_T, V^{(0)} + V^\rho + V_T \boldsymbol{\Sigma} \right) \right]$$
$$+ R(\sqrt{\lambda_T} W_T, \sqrt{\lambda_T} V_T) \le (1 + \gamma) \mathsf{OPT} + \varepsilon_0 \ .$$

For the same reason as the proof of Theorem 3, we know w.h.p. among $\widetilde{O}(1/\varepsilon_0^2)$ choices of $j$,

$$\min_j \left\{ \mathbb{E}_{\boldsymbol{\Sigma}, z \in \mathcal{Z}} L_F(z; \lambda_T, W^{(0)} + W^{\rho,j} + \boldsymbol{\Sigma} W_T, V^{(0)} + V^{\rho,j} + V_T \boldsymbol{\Sigma}) \right\} \le (1 + \gamma) \mathsf{OPT} + 2\varepsilon_0$$
(C.44)

Without loss of generality, in the remainder of the proof we assume $\mathsf{OPT} \le O(\varepsilon_0)$. This can be done because is $\varepsilon_0$ is too small we can increase it to $\varepsilon_0 = \Theta(\mathsf{OPT})$. By our regularizer parameters $\lambda_w, \lambda_v$ in Table 1, we know

$$R(\sqrt{\lambda_T} W_T, \sqrt{\lambda_T} V_T) \le (1 + \gamma) \mathsf{OPT} + \varepsilon_0 \le O(\varepsilon_0)$$
$$\implies \|\sqrt{\lambda_T} W_T\|_{2,4} \le O(\tau_w' \varepsilon_0^{1/4}) \text{ and } \|\sqrt{\lambda_T} V_T\|_{2,2} \le O(\tau_v' \varepsilon_0^{1/2})$$
(C.45)

By Lemma 6.11 (but viewing $V^{(0)} + V^\rho$ as $V^{(0)}$ and viewing $W^{(0)} + W^\rho$ as $W^{(0)}$), we know for every $(x, y)$, w.h.p. over $W^{(0)}, V^{(0)}, W^\rho, V^\rho, \boldsymbol{\Sigma}$

$$f_r(x; W^{(0)} + W^\rho + \boldsymbol{\Sigma} W_T, V^{(0)} + V^\rho + V_T \boldsymbol{\Sigma})$$
$$= a_r D_{v,x,\rho} \left( (V^{(0)} + V^\rho) D_{w,x,\rho} ((W^{(0)} + V^\rho)x + b_1) + b_2 \right) + a_r D_{v,x,\rho} V_T D_{w,x,\rho} W_T x \pm \varepsilon$$
$$= f_r(x; W^{(0)} + W^\rho, V^{(0)} + V^\rho) + g_r^{(b,b)}(x; W_T, V_T) \pm \varepsilon/k$$

where $D_{v,x,\rho}$ and $D_{w,x,\rho}$ are the diagonal sign indicator matrices at weights $W^{(0)} + W^\rho, V^{(0)} + V^\rho$, and we denote by $g_r^{(b,b)}(x; W', V') := a_r D_{v,x,\rho} V' D_{w,x,\rho} W' x$ the output of the pseudo network. This immediately implies for every $(x, y)$ and every $j$, w.h.p. over $W^{(0)}, V^{(0)}, W^{\rho,j}, V^{\rho,j}, \boldsymbol{\Sigma}$

$$f_r(x; W^{(0)} + W^{\rho,j} + \boldsymbol{\Sigma} W_T, V^{(0)} + V^{\rho,j} + V_T \boldsymbol{\Sigma})$$
$$= f_r(x; W^{(0)} + W^{\rho,j}, V^{(0)} + V^{\rho,j}) + g_r^{(b,b)}(x; W_T, V_T) \pm \varepsilon/k$$
(C.46)

Using the 1-Lipschitz continuity of $L$ together with (C.45) and (C.46), it is not hard to derive that for every $j$, with high probability over $(x, y) \sim \mathcal{D}, W^{(0)}, V^{(0)}, W^{\rho,j}, V^{\rho,j}$[30]

$$\left| \mathbb{E}_{\boldsymbol{\Sigma}} L_F(z; \lambda_T, W^{(0)} + W^{\rho,j} + \boldsymbol{\Sigma} W_T, V^{(0)} + V^{\rho,j} + V_T \boldsymbol{\Sigma}) \right| \le \widetilde{O}(C_0) \ .$$

Therefore, we can plug in the Rademacher complexity from Lemma 6.12 with $b = \widetilde{O}(C_0)$ into standard generalization statement Corollary A.11.[31] Using our choices of $\tau_w'$ and $\tau_v'$ from Table 1 as well as $m_1 = m_2$, the Rademacher complexity is dominated by

$$\lambda_T \frac{m_2^{1/2} m_1^{1/4} \|W_T\|_{2,4} \|V_T\|_{2,2}}{\sqrt{N}} \le \frac{\varepsilon_0^{3/4} m_2^{1/2} m_1^{1/4} \tau_v' \tau_w'}{\sqrt{N}} \le \frac{C_0}{\sqrt{N}}$$

In other words, as long as $N \ge (kC_0/\varepsilon_0)^2$, the Rademacher complexity of a single output is at most $\frac{\varepsilon_0}{k}$, so the generalization error is at most $\varepsilon_0$ by Corollary A.11. Or, in symbols, for every $j$, w.h.p. over $W^{(0)}, V^{(0)}, W^{\rho,j}, V^{\rho,j}$,

$$\mathbb{E}_{\boldsymbol{\Sigma}, z \in \mathcal{D}} L_F(z; \lambda_T, W^{(0)} + W^{\rho,j} + \boldsymbol{\Sigma} W_T, V^{(0)} + V^{\rho,j} + V_T \boldsymbol{\Sigma})$$
$$\le \mathbb{E}_{\boldsymbol{\Sigma}, z \in \mathcal{Z}} L_F(z; \lambda_T, W^{(0)} + W^{\rho,j} + \boldsymbol{\Sigma} W_T, V^{(0)} + V^{\rho,j} + V_T \boldsymbol{\Sigma}) + \varepsilon_0$$
(C.47)

Putting this into (C.44), we have

$$\min_j \left\{ \mathbb{E}_{\boldsymbol{\Sigma}, z \in \mathcal{D}} L_F(z; \lambda_T, W^{(0)} + W^{\rho,j} + \boldsymbol{\Sigma} W_T, V^{(0)} + V^{\rho,j} + V_T \boldsymbol{\Sigma}) \right\} \le (1 + \gamma) \mathsf{OPT} + 3\varepsilon_0$$
(C.48)

Next, let us take expectation over $z \sim \mathcal{D}$ for (C.46) (strictly speaking, this needs one to carefully deal with the tail bound and apply the 1-Lipschitz continuity of $L$). We have for every $j$, w.h.p over $W^{(0)}, V^{(0)}, W^{\rho,j}, V^{\rho,j}, \Sigma$

$$\mathbb{E}_{(x,y)\sim\mathcal{D}} \left[ L\left( \lambda_T F\left( x; W^{(0)} + W^{\rho,j} + \Sigma W_T, V^{(0)} + V^{\rho,j} + V_T \Sigma \right), y \right) \right]$$

$$= \mathbb{E}_{(x,y)\sim\mathcal{D}} \left[ L\left( \lambda_T F\left( x; W^{(0)} + W^{\rho,j}, V^{(0)} + V^{\rho,j} \right) + \lambda_T G^{(b,b)}\left( x; W_T, V_T \right), y \right) \right] \pm 2\varepsilon$$

Since the right hand side (except the $\pm 2\varepsilon$ term) does not depend on the randomness of $\Sigma$, we know that the left hand side with respect to a random sample $\widehat{\Sigma}$ must stay close to its expectation with respect to $\Sigma$. Or, in symbols, for every $j$, w.h.p over $W^{(0)}, V^{(0)}, W^{\rho,j}, V^{\rho,j}$

$$\mathbb{E}_{(x,y)\sim\mathcal{D}} \left[ L\left( \lambda_T F\left( x; W^{(0)} + W^{\rho,j} + \widehat{\Sigma} W_T, V^{(0)} + V^{\rho,j} + V_T \widehat{\Sigma} \right), y \right) \right]$$

$$= \mathbb{E}_{\Sigma,(x,y)\sim\mathcal{D}} \left[ L\left( \lambda_T F\left( x; W^{(0)} + W^{\rho,j} + \Sigma W_T, V^{(0)} + V^{\rho,j} + V_T \Sigma \right), y \right) \right] \pm 4\varepsilon \ . \quad \text{(C.49)}$$

For similar reason, replacing $\mathcal{D}$ with $\mathcal{Z}$, we have

$$\mathbb{E}_{(x,y)\sim\mathcal{Z}} \left[ L\left( \lambda_T F\left( x; W^{(0)} + W^{\rho,j} + \widehat{\Sigma} W_T, V^{(0)} + V^{\rho,j} + V_T \widehat{\Sigma} \right), y \right) \right]$$

$$= \mathbb{E}_{\Sigma,(x,y)\sim\mathcal{Z}} \left[ L\left( \lambda_T F\left( x; W^{(0)} + W^{\rho,j} + \Sigma W_T, V^{(0)} + V^{\rho,j} + V_T \Sigma \right), y \right) \right] \pm 4\varepsilon \ . \quad \text{(C.50)}$$

These imply two things.

- Putting (C.50) into (C.44), we have

$$\mathbb{E}_{z\in\mathcal{Z}} L_F(z; \lambda_T, W^{(0)} + W^{\rho,j^*} + \widehat{\Sigma} W_T, V^{(0)} + V^{\rho,j^*} + V_T \widehat{\Sigma}) \quad \text{(C.51)}$$

$$= \min_j \left\{ \mathbb{E}_{z\in\mathcal{Z}} L_F(z; \lambda_T, W^{(0)} + W^{\rho,j} + \widehat{\Sigma} W_T, V^{(0)} + V^{\rho,j} + V_T \widehat{\Sigma}) \right\} \leq (1+\gamma)\mathsf{OPT} + 3\varepsilon_0 \ .$$

- Putting (C.50) and (C.49) into (C.47), we have

$$\mathbb{E}_{z\in\mathcal{D}} L_F(z; \lambda_T, W^{(0)} + W^{\rho,j} + \widehat{\Sigma} W_T, V^{(0)} + V^{\rho,j} + V_T \widehat{\Sigma})$$

$$\leq \mathbb{E}_{z\in\mathcal{Z}} L_F(z; \lambda_T, W^{(0)} + W^{\rho,j} + \widehat{\Sigma} W_T, V^{(0)} + V^{\rho,j} + V_T \widehat{\Sigma}) + 2\varepsilon_0 \quad \text{(C.52)}$$

Combining (C.51) and (C.52), we immediately have

$$\mathbb{E}_{(x,y)\sim\mathcal{D}} \left[ L\left( \lambda_T F\left( x; W^{(0)} + W^{\rho,j^*} + \widehat{\Sigma} W_T, V^{(0)} + V^{\rho,j^*} + V_T \widehat{\Sigma} \right), y \right) \right] \leq (1+\gamma)\mathsf{OPT} + 5\varepsilon_0$$

as desired. Scaling down $\varepsilon_0$ by constant finishes the proof. $\qquad\square$