[Reviews · NeurIPS 2019]

Reviewer 1



The algorithm shows that SGD plus overparameterization can learn any neural network with a well-behaved Taylor expansion. This is interesting! The paper also makes some questionable claims: "To the best of our knowledge, this is the first result showing that training only hidden layers of neural networks can provably learn two (or even three) layer neural networks with non-trivial activation functions." The work of Daniely already shows how to train all the hidden layers of (any depth) network to learn classes using a (different) measure of complexity. It is true that his method is similar to the 'train last layer' approach, but in this submission something similar happens in the two layer case. The authors seem to say that something more sophisticated happens when training a 3 layer network, but I could not understand this section. It is not clear to me what the relationship to NTK is. Also, in the paper the authors on line 126 indicate that sample complexity is polynomially related to network size (within poly(1/\eps)). But then later it says the sample complexity is polylog in the network size. Can you please clarify? The authors do not use normal SGD: they use SGD plus noise. Why can't standard SGD work?

Reviewer 2



The paper tackles the questions of how neural networks are able to generalize despite having much more parameters than training samples and how stochastic gradient descent is able to provide a good network despite the highly nonconvex optimization. The authors do this by showing how neural networks are able to perform almost as well as the best performer in a broad concept class. Overall, the theorems are powerful since they do not pose as stringent constraints as other works: no assumption is made on the data distribution and the target concept class is broad (involving nonlinear functions). Further, the paper provides theoretical insight on three-layer networks and show their power over two-layer networks. Thus, the insights are significant and provide new insights to the research field. Finally, the paper is clearly written, providing a clear discussion of how this work relates to others in the field and providing some intuition into the technical results.

Reviewer 3



I thank the authors for their response. I understand that generalization is not the major contribution in this paper -- thanks for the note. I also appreciate the plot showing the numerical values of the weight norms for varying width. It is reassuring to know that these quantities do vary inversely with width for this setting. I think adding these sorts of plots to the appendix of the paper (with a bit more detailed experimentation and discussion) would be useful for the paper. Keeping these in mind, I have increased the score. ========== Summary: ========== This paper presents an analysis of the training dynamics and generalization behavior of two/three layered neural networks under a loss function that is Lipschitz, convex and non-negative in terms of the network output. Crucially, the activations are allowed to be non-smooth. For the two-layer result, the training algorithm is stochastic gradient descent and for the three-layer networks, the algorithm is a noisy version of SGD that has explicit weight decay. Furthermore, in the two-layer result, the top layer weights are assumed to be carefully initialized to sufficiently small values. The results in all these cases, are all of the following form: - when the true target can be represented similarly by a two/three layer network - when the learner network is sufficiently overparameterized, - when the learning rate is set to be a particular quantity based on the other hyperparamters (upto some constants), then, the algorithm converges to a solution with test loss that is independent of the learner's parameter count (upto logarithmic factors). Positives: ========= 1. This is one of the few works in this space that considers (i) a non-smooth activation function and (ii) considers three layer networks and (iii) the result applies to a large class of underlying data distributions. These are significantly more realistic but harder setups. As highlighted by the paper, the analysis in this paper also takes into account the non-convex interactions between the weight matrices across the multiple layers (unlike earlier works which typically circumvent these interactions by considering favorable setups) and this is done while also admitting significant sign changes on the activations (i.e., how many activations suffer sign change after training). 2. The results require very long, extensive proofs and I appreciate how the authors have presented the main ideas/challenges concisely. Overall, I found the paper well-written (despite the density of notations). Main concern: ======== 3. This paper has two contributions: one that analyzes convergence, and the other that analyzes the generalization of the final solution. My concerns lie only with the latter. The paper claims that its analysis sheds insight into why neural networks generalize well, by (a) arguing that information is "evenly distributed among neurons" (this is inferred by examining the ratio between the ||W||_{2,4} norm and the ||W||_{F} norm) and by (b) showing that the generalization bound is parameter-count-independent (ignoring a log factor). Delving a bit deeper into the main presented result, I'd like to focus on the actual generalization bound applied to arrive at these results, namely, (a) Lemma 6.9 for three layer networks, and (b) Lemma C.5 in two-layer networks. These are (fairly simple) applications of Rademacher complexity bounds on certain-norm bounded classes of deep networks. The parameter-count-independence in these bounds crucially depend on the choice of the learning rate (and also other hyperparameters), which is required to be a value that is inversely proportional to the width. If one were to apply these bounds in other situations, they would be far from parameter count dependent (see point 3.2 (b) below). Overall, it seems like the the parameter-count-independence in these generalization bounds may rather be an artifact of forcing thinner networks to learn with larger learning rates. Of course, I can't say this for sure. I understand these the learning rates need to be set carefully to make certain claims about the optimization, and I'm completely fine with making these kinds of assumptions as far as the optimization theory is concerned. But my scepticism arises only we venture into the generalization theory. To discuss this more concretely, I've three sub-questions/comments below: 3.1 Let's use W' to denote the first layer's update matrix (final-initial matrix), and V' to denote the second layer's update matrix. Now, in the three layer network, the given bound grows as ||W'||_{2,4}*(width of first layer * \sqrt{width of second layer})/\sqrt{training set size} and also as ||V'||_F* width of second layer /\sqrt{training set size} For the two layer case the bound grows as: ||W'||_{2,\infty}*(width of first layer)/\sqrt{training set size}. In the analysis of this paper, the norms are shown to be favorably bounded during optimization, in a way that it cancels out the other terms in these bounds, making them parameter-count-independent. However, in practice, this is not really the case, at least, if one were to fix the learning rate to be independent of the parameter count (although as I note in 3.2, even if we vary the learning rate as required by this paper, it doesn't seem to help). For example, the analysis in this paper upper bounds ||W'||_{2,\infty} as 1/width, which would imply an upper bound on distance from initialization, ||W'||_F as 1/\sqrt{width}. But in practice -- when the learning rate is fixed -- distance from initialization is typically a constant (or shows very slight decrease) beyond a particular level of overparameterization (see Fig 2 left in "Towards Understanding the Role of Over-Parametrization in Generalization of Neural Networks" https://arxiv.org/pdf/1805.12076.pdf or Fig 1 left in "Generalization in Deep Networks: The Role of Distance from Initialization" https://arxiv.org/pdf/1901.01672.pdf). If we were to apply the same Rademacher complexity analysis from this paper, for the more realistic situation where learning rate is fixed, these bounds would become parameter-count-dependent. And this is what makes the generalization bound in the paper less insightful to me, and also a bit fishy. 3.2 Now what happens if in practice we were to vary the learning rate as required by the theoretical analysis here? I'd be curious about the following experiments: - (a) Train networks of varying widths exactly as defined in the setups considered in the paper. Then, numerically compute the generalization bounds in Lemma 6.9 or C.5, by plugging in the **empirically** observed norm bounds (NOT by plugging in the theoretically derived upper bounds on these norms). Do these numerically computed bounds stay constant with parameter count? - (b) Train standard neural networks of varying widths, with **standard** initialization, and with learning rate inversely proportional to width. Again, how do these bounds vary with width? -- I tried experiment (b) on a synthetic dataset, where I varied the width of a 3 layer network across [50, 100, 200, 400, 800, 1600] and the learning rate proportionally across [0.128, 0.064, 0.032, 0.016, 0.008, 0.004] and I always observed that these norms showed only a mild decrease -- at best the norms decrease by a factor of 2, even though the width increases by as much as a factor of 2^5 from 50 to 1600. 3.3 In the paper, the learning rates have to be tuned carefully as they need to satisfy certain upper bounds and lower bounds depending on the width. How crucial is the lower bound on the learning rate to the proof of convergence? (a) I presume the lower bound would be important for showing finite-time learnability. If we were to not worry about the time complexity, is it possible to modify the discussion on optimization to apply for arbitrarily small learning rates? Based on my preliminary understanding, the lower bound does seem to be crucial for aspects besides the time complexity, and hence cannot be gotten rid of. (b) If it were possible to get rid of the lower bound, one should then be able to derive generalization bounds across different widths for some sufficiently small fixed, **width-independent** learning rates. One could then study generalization bounds in this setting; and if these bounds are shown to be width-independent, they'd be a lot more interesting to me. It's not clear if the current analysis is amenable to this, but I'd like to know the authors' intuition about this as they should be more familiar with the intricate details. Minor suggestions ==== - For the sake of transparency, please state in the introduction that the SGD considered in the three layer case is explicitly regularized and stochasticized. I went through some parts of the "pre-appendix" and I've some small suggestions for improving the discussion here: - The phrase "a good subnetwork that approximates the target well" is a bit confusing. What exactly makes the subnetwork good? That it approximates the target well? That it has small Frobenius norms? - The discussion below Lemma 6.6 is not clear enough -- I didn't completely understand what exactly this Lemma wants to show (as in what does Line 628 exactly mean) or how it implies convergence Lemma 6.7. - The outline of (the outline of) the proof in Lines 451-459 of the three layer network can have some more details in it. - It'd be good to have references to the section headings in the actual appendix wherever appropriate. For example, specify the section headings which have the proofs for Lemmas 6.x. Line 674: the following the Line 590: the naively approach

[Author Response · NeurIPS 2019]

We thank all the reviewers for the time reading our paper! We will fix all the minor issues, and below we only address
the main concerns. We restate those questions below.

- 3 • **R2:** The two-layer part of this submission might be similar to Daniely's?
No. Daniely's result trains only the *last layer* of (any depth) network, and the changes of hidden layers are negligible
(can be set zero). Daniely proves generalization using conjugate kernel, but it's unclear what *explicit functions* can
be learned and what's the *explicit sample complexity*. In contrast, we consider target functions consisting of 2-layer
networks, and show how they can be PAC-learned and what's the sample complexity.

- 8 • **R2:** Why the 3-layer result is *not based on NTK* (neural tangent kernel)?
Recall $W = W_0 + W'$ and $V = V_0 + V'$ where $W_0, V_0$ are initializations. In NTK, by ignoring higher-order terms,
the network is a linear function in $(W', V')$, so entries of $W'$ will *never* be multiplied with $V'$. In contrast, see
Lemma 6.10, we track $DV'DW'$ so $V'$ and $W'$ are multiplied together. This is non-linear so is not NTK. In fact, as
we explained in the paper, our concept class (learnable by three-layer networks) is not captured by the NTK of a
three layer network. We shall make it more clear in the revision. (We thank **R5** for carefully reading our paper and
acknowledge "unlike prior work" we have considered "non-convex interactions between weight matrices.")

- 15 • **R2:** Does the sample complexity really depend polylog in the network size?
Sorry for the confusion. What we mean is sample complexity *grows* polylog in $m$. Indeed, the sample complexity
shares some poly terms with $M_0$, but as $m \geq M_0$, sample complexity only *grows* polylog in $m$.

- 18 • **R2+R4:** Why can't standard SGD (without noise) work?
There are many reasons (see footnotes on Page 7). For instance, *all* existing escape-saddle point papers need noise.
Since we rely on such existing work, we need noise for theoretical purpose.

- 21 • **R4:** Why increasing $m$ supports more target functions? What is $R$?
Sorry for the confusion and it is actually simpler than you may have thought. For any target function, there is
a corresponding $M_0$ such that our theorem applies whenever $m \geq M_0$. Here, $M_0$ depends on the complexity
notion introduced on line 122. In other words, if we increase $M_0$, there will be more functions to be supported by
this threshold $M_0$. Finally, $R$ only plays some role in our 3-layer theorem, where it allows the complexity to be
composed with another complexity function.

- 27 • **R4:** Modify references to e.g. Table 1 in line 239. Great suggestion and we will do that!

- 28 • **R5** raises concerns about the significance of the generalization part of this paper.
Although the generalization lemmas *on their own* are simple, they are not our main contribution (and constitute
5% of this paper). **Instead**, our main contribution is to make the convergence theorems and generalization lemmas
*compatible*: SGD can find solutions with small norms so that generalization lemmas apply (almost independent of
$m$). This cannot be done by combining any "convergence theorem" and any "generalization lemma". For instance,
the prior work Allen-Zhu et al [2] proves convergence, but it is not compatible with our generalization lemmas.

- 34 • **R5** has concerns about the practical relevance of our generalization lemmas.
  - 35 – **R5**: What will happen if learning rate is independent of the parameter count?
  36 Under our (wlog.) choice of initialization $W, V \sim \mathcal{N}(0, 1/m)$ and output layer $\mathcal{N}(0, 1)$, it is ***not* a good idea**
  37 to use constant learning rate. For instance, in our 3-layer experiment below, any learning rate $lr \in [0.01, 0.1]$
  38 works for $m = 50$, but any learning rate $lr \in (0.01, \infty)$ gives NaN error for $m \geq 5000$. In general, learning
  39 rate *depends on initialization*: if hidden weights are scaled up and output layer is scaled down, then the learning
  40 rate will increase.
  - 41 – **R5:** If learning rate decreases as $m$, does $\|W'\|$ and $\|V'\|$ decrease experimentally?
  42 **Yes.** For 2-layer network on MNIST with $lr = 400/m$ and target accuracy 95%, norm $\|W'\|_F/\|W_0\|_F$
  43 decreases, see Li-Liang [30, Fig 5 of page 26 of NeurIPS camera ready]. Below in left figure, we show $\|W'\|_{2,\infty}$
  44 also decreases. As another example with 3-layer networks, say inputs are random $\|x\|_2 = 1$ and the true label
  45 $y = x_1 \cdot x_2 + x_3 \cdot x_4$. Using our standard initialization, then $\|W'\|_F, \|W'\|_{2,4}, \|V'\|_F$ all decrease as $m$ increases
  46 (for $lr = 1/m$, test errors drop below 0.003 within 30 epochs), see right figure. We have plugged them into the
  47 generalization formula and they do give very meaningful bounds on sample complexity even for large $m$.



[Meta-Review · NeurIPS 2019]

This paper presents analysis of what 3-layer neural nets with non-smooth activation functions can learn. Extending such analysis to three layers is very challenging and all reviewers find this result interesting. Therefore, I recommend acceptance.